# Learning on the Edge: Online Learning with Stochastic Feedback Graphs

**Emmanuel Esposito**[*]
Dept. of Computer Science
Università degli Studi di Milano, Italy
& Istituto Italiano di Tecnologia, Italy
`emmanuel@emmanuelesposito.it`

**Federico Fusco**[*]
Dept. of Computer, Control
and Management Engineering
Sapienza Università di Roma, Italy
`fuscof@diag.uniroma1.it`

**Dirk van der Hoeven**[*]
Dept. of Computer Science
Università degli Studi di Milano, Italy
`dirk@dirkvanderhoeven.com`

**Nicolò Cesa-Bianchi**
Dept. of Computer Science
Università degli Studi di Milano, Italy
`nicolo.cesa-bianchi@unimi.it`

## Abstract

The framework of feedback graphs is a generalization of sequential decision-making with bandit or full information feedback. In this work, we study an extension where the directed feedback graph is stochastic, following a distribution similar to the classical Erdős-Rényi model. Specifically, in each round every edge in the graph is either realized or not with a distinct probability for each edge. We prove nearly optimal regret bounds of order $\min\{\min_\varepsilon \sqrt{(\alpha_\varepsilon/\varepsilon)T}, \min_\varepsilon(\delta_\varepsilon/\varepsilon)^{1/3}T^{2/3}\}$ (ignoring logarithmic factors), where $\alpha_\varepsilon$ and $\delta_\varepsilon$ are graph-theoretic quantities measured on the support of the stochastic feedback graph $\mathcal{G}$ with edge probabilities thresholded at $\varepsilon$. Our result, which holds without any preliminary knowledge about $\mathcal{G}$, requires the learner to observe only the realized out-neighborhood of the chosen action. When the learner is allowed to observe the realization of the entire graph (but only the losses in the out-neighborhood of the chosen action), we derive a more efficient algorithm featuring a dependence on weighted versions of the independence and weak domination numbers that exhibits improved bounds for some special cases.

## 1 Introduction

In this work we study an online learning framework for decision-making with partial feedback. In each decision round, the learner chooses an action in a fixed set and is charged a loss. In our setting, the loss of any action in all decision rounds is preliminarily chosen by an adversary, but the feedback received by the learner at the end of each round $t$ is stochastic. More specifically, the loss of each action $i$ (including $I_t$, the one selected by the learner at round $t$) is independently observed with a certain probability $p(I_t, i)$, where the probabilities $p(i, j)$ for all pairs $i, j$ are fixed but unknown.

This feedback model can be viewed as a stochastic version of the feedback graph model for online learning [Mannor and Shamir, 2011], where the feedback received by the learner at the end of each round is determined by a directed graph defined over the set of actions. In this model, the learner deterministically observes the losses of all the actions in the

---

[*]Equal contribution.

36th Conference on Neural Information Processing Systems (NeurIPS 2022).

out-neighborhood of the action selected in that round. In certain applications, however, deterministic feedback is not realistic. Consider for instance a sensor network for monitoring the environment, where the learner can decide which sensor to probe in order to maximize some performance measure. Each probed sensor may also receive readings of other sensors, but whether a sensor successfully transmits to another sensor depends on a number of environmental factors, which include the position of the two sensors, but also their internal state (e.g., battery levels) and the weather conditions. Due to the variability of some of these factors, the possibility of reading from another sensor can be naturally modeled as a stochastic event.

Online learning with adversarial losses and stochastic feedback graphs has been studied before, but under fairly restrictive assumptions on the probabilities $p(i, j)$. Let $\mathcal{G}$ be a stochastic feedback graph, represented by its probability matrix $p(i, j)$ for $i, j \in V$ where $V$ is the action set. When $p(i, j) = \varepsilon$ for all distinct $i, j \in V$ and for some $\varepsilon > 0$, then $\mathcal{G}$ follows the Erdős-Rényi random graph model. Under the assumption that $\varepsilon$ is known and $p(i, i) = 1$ for all $i \in V$ (all self-loops occur w.p. 1), Alon et al. [2017] show that the optimal regret after $T$ rounds is of order $\sqrt{T/\varepsilon}$, up to logarithmic factors. This result has been extended by Kocák et al. [2016a], who prove a regret bound of order $\sqrt{\sum_t (1/\varepsilon_t)}$ when the parameter $\varepsilon_t$ of the random graph is unknown and allowed to change over time. However, their result holds only under rather strong assumptions on the sequence $\varepsilon_t$ for $t \geq 1$. In a recent work, Ghari and Shen [2022] show a regret bound of order $(\alpha/\varepsilon)\sqrt{KT}$, ignoring logarithmic factors, when each (unknown) probability $p(i, j)$ in $\mathcal{G}$ is either zero or at least $\varepsilon$ for some known $\varepsilon > 0$, and all self-loops $(i, i)$ have probability $p(i, i) \geq \varepsilon$. Here $\alpha$ is the independence number (computed ignoring edge orientations) of the support graph $\text{supp}(\mathcal{G})$; i.e., the directed graph with adjacency matrix $A(i, j) = \mathbb{I}_{\{p(i,j)>0\}}$. Their bound holds under the assumption that $\text{supp}(\mathcal{G})$ is preliminarily known to the learner.

Our analysis does away with a crucial assumption that was key to prove all previous results. Namely, we do not assume any special property of the matrix $\mathcal{G}$, and we do not require the learner to have any preliminary knowledge of this matrix. The fact that positive edge probabilities are not bounded away from zero implies that the learner must choose a threshold $\varepsilon \in (0, 1]$ below which the edges are deemed to be too rare to be exploitable for learning. If $\varepsilon$ is too small, then waiting for rare edges slows down learning. On the other hand, if $\varepsilon$ is too large, then the feedback becomes sparse and the regret increases.

To formalize the intuition of rare edges, we introduce the notion of thresholded graph $\text{supp}(\mathcal{G}_\varepsilon)$ for any $\varepsilon > 0$. This is the directed graph with adjacency matrix $A(i, j) = \mathbb{I}_{\{p(i,j) \geq \varepsilon\}}$. As the thresholded graph is a deterministic feedback graph $G$, we can refer to Alon et al. [2015] for a characterization of minimax regret $R_T$ based on whether $G$ is not observable ($R_T$ of order $T$), weakly observable ($R_T$ of order $\delta^{1/3}T^{2/3}$), or strongly observable ($R_T$ of order $\sqrt{\alpha T}$).[1] Here $\alpha$ and $\delta$ are, respectively, the independence and the weak domination number of $G$; see Section 2 for definitions. Let $\alpha_\varepsilon$ and $\delta_\varepsilon$ respectively denote the independence number and the weak domination number of $\text{supp}(\mathcal{G}_\varepsilon)$. As $\alpha_\varepsilon$ and $\delta_\varepsilon$ both grow when $\varepsilon$ gets larger, the ratios $\alpha_\varepsilon/\varepsilon$ and $\delta_\varepsilon/\varepsilon$ capture the trade-off involved in choosing $\varepsilon$. We define the optimal values for $\varepsilon$ as follows:

$$\varepsilon_s^* = \underset{\varepsilon \in (0,1]}{\arg\min} \left\{ \frac{\alpha_\varepsilon}{\varepsilon} \; : \; \text{supp}(\mathcal{G}_\varepsilon) \text{ is strongly observable} \right\}, \tag{1}$$

$$\varepsilon_w^* = \underset{\varepsilon \in (0,1]}{\arg\min} \left\{ \frac{\delta_\varepsilon}{\varepsilon} \; : \; \text{supp}(\mathcal{G}_\varepsilon) \text{ is observable} \right\}. \tag{2}$$

We adopt the convention that the minimum of an empty set is infinity and the relative $\arg\min$ is set to 0. The $\arg\min$ are well defined: there are at most $K^2$ values of $\varepsilon$ for which the support of $\mathcal{G}_\varepsilon$ varies, and the minimum is attained in one of these values. For simplicity, we let $\alpha^* = \alpha_{\varepsilon_s^*}$ and $\delta^* = \delta_{\varepsilon_w^*}$. Our first result can be informally stated as follows.

**Theorem 1** (Informal). *Consider the problem of online learning with an unknown stochastic feedback graph $\mathcal{G}$ on $T$ time steps. If $\text{supp}(\mathcal{G}_\varepsilon)$ is not observable for $\varepsilon = \tilde{\Theta}(K^3/T)$, then any learning algorithm suffers regret linear in $T$. Otherwise, there exists an algorithm whose*

---

[1] All these rates ignore logarithmic factors.

*regret satisfies (ignoring polylog factors in $K$ and $T$)*

$$R_T \leq \min \left\{ \sqrt{\frac{\alpha^*}{\varepsilon_s^*}T}, \left(\frac{\delta^*}{\varepsilon_w^*}\right)^{1/3} T^{2/3} \right\}.$$

*This bound is tight (up to polylog factors).*

This result shows that, without any preliminary knowledge of $\mathcal{G}$, we can obtain a bound that optimally trades off between the strongly observable rate $\sqrt{(\alpha^*/\varepsilon_s^*)T}$, for the best threshold $\varepsilon$ for which $\text{supp}(\mathcal{G}_\varepsilon)$ is strongly observable, and the (weakly) observable rate $(\delta^*/\varepsilon_w^*)^{1/3}T^{2/3}$, for the best threshold $\varepsilon$ for which $\text{supp}(\mathcal{G}_\varepsilon)$ is (weakly) observable. Note that this result improves on Ghari and Shen [2022] bound $(\alpha_\varepsilon/\varepsilon)\sqrt{KT}$, who additionally assume that $\text{supp}(\mathcal{G}_\varepsilon)$ and $\varepsilon$ (a lower bound on the self-loop probabilities) are both preliminarily available to the learner. On the other hand, the algorithm achieving the bound of Theorem 1 need not receive any information (neither prior nor during the learning process) besides the stochastic feedback.

We obtain positive results in Theorem 1 via an elaborate reduction to online learning with deterministic feedback graphs. Our algorithm works in two phases: first, it learns the edge probabilities in a round-robin procedure, then it commits to a carefully chosen estimate of the feedback graph and feeds it to an algorithm for online learning with deterministic feedback graphs. There are two main technical challenges the algorithm faces: on the one hand, it needs to switch from the first to the second phase at the right time in order to achieve the optimal regret. On the other hand, in order for the reduction to work, it needs to simulate the behaviour of a deterministic feedback graph using only feedback from a stochastic feedback graph (with unknown edge probabilities). We complement the positive results in Theorem 1 with matching lower bounds that are obtained by a suitable modification of the hard instances in Alon et al. [2015, 2017] so as to consider stochastic feedback graphs.

Our last result is an algorithm that, at the cost of an additional assumption on the feedback (i.e., the learner additionally observes the realization of the entire feedback graph at the end of each round), has regret which is never worse and may be considerably better than the regret of the algorithm in Theorem 1. While the bounds in Theorem 1 are tight up to log factors, we show that the factors $\alpha^*/\varepsilon_s^*$ and $\delta^*/\varepsilon_w^*$ can be improved for specific feedback graphs. Specifically, we design weighted versions of the independence and weak domination numbers, where the weights of a given node depend on the probabilities of seeing the loss of that node. On the technical side, we design a new importance-weighted estimator which uses a particular version of upper confidence bound estimates of the edge probabilities $p(i, j)$, rather than the true edge probabilities, which are unknown. We show that the cost of using this estimator is of the same order as the regret bound achievable had we known $p(i, j)$. Additionally, the algorithm that obtains these improved bounds is more efficient than the algorithm of Theorem 1. The improvement in efficiency comes from the following idea: we start with an optimistic algorithm that assumes that the support of $\mathcal{G}$ is strongly observable and only switches to the assumption that the support of $\mathcal{G}$ is (weakly) observable when it estimates that the regret under this second assumption is smaller. The algorithm learns which regime is better by keeping track of a bound on the regret of the optimistic algorithm while simultaneously estimating the regret in the (weakly) observable case, which it can do efficiently.

**Additional related work.** The problem of adversarial online learning with feedback graphs was introduced by Mannor and Shamir [2011], in the special case where all nodes in the feedback graph have self-loops. The results of Alon et al. [2015] (also based on prior work by Alon et al. [2013], Kocák et al. [2014]) have been recently slightly improved by Chen et al. [2021], with tighter constants in the regret bound. Variants of the adversarial setting have been studied by Feng and Loh [2018], Arora et al. [2019], Rangi and Franceschetti [2019] and Van der Hoeven et al. [2021], who study online learning with feedback graphs and switching costs and online multiclass classification with feedback graphs, respectively. There is also a considerable amount of work in the stochastic setting [Liu et al., 2018, Cortes et al., 2019, Li et al., 2020]. Finally, Rouyer et al. [2022] and Ito et al. [2022] independently designed different best-of-both-worlds learning algorithms achieving nearly optimal (up to polylogarithmic factors in $T$) regret bounds in the adversarial and stochastic settings.

Following Mannor and Shamir [2011], we can consider a more general scenario where the feedback graph is not fixed but changes over time, resulting in a sequence $G_1, \ldots, G_T$ of feedback graphs. Cohen et al. [2016] study a setting where the graphs are adversarially chosen and only the local structure of the feedback graph is observed. They show that, if the losses are generated by an adversary and all nodes always have a self-loop, one cannot do better than $\sqrt{KT}$ regret, and we might as well simply employ a standard bandit algorithm. Furthermore, removing the guarantee on the self-loops induces an $\Omega(T)$ regret. In Section 3, we are in a similar situation, as we also observe only local information about the feedback graph and the losses are generated by an adversary. However, we show that if the graphs are stochastically generated with a strongly observable support for some threshold $\varepsilon$, there is a $\sqrt{\alpha T / \varepsilon}$ regret bound. As a consequence, for $\varepsilon$ not too small, observing only the local information about the feedback graphs is in fact sufficient to obtain better results than in the bandit setting. Similarly, if there are no self-loops in the support but the support is weakly observable, then our regret bounds are sublinear rather than linear in $T$. Alon et al. [2013, 2017] and Kocák et al. [2014] also consider adversarially generated sequences $G_1, G_2, \ldots$ of deterministic feedback graphs. In the case of directed feedback graphs, Alon et al. [2013] investigate a model in which $G_t$ is revealed to the learner at the beginning of each round $t$. Alon et al. [2017] and Kocák et al. [2014] extend this analysis to the case when $G_t$ is strongly observable and made available only at the end of each round $t$. In comparison, in our setting the graphs (or the local information about the graph) revealed to the learner (at the end of each round) may not even be observable, let alone strongly observable. Despite this seemingly challenging setting for previous works, we nevertheless obtain sublinear regret bounds. Finally, Kocák et al. [2016b] study a feedback model where the losses of other actions in the out-neighborhood of the action played are observed with an edge-dependent noise. In their setting, the feedback graphs $G_t$ are weighted and revealed at the beginning of each round. They introduce edge weights $s_t(i, j) \in [0, 1]$ that determine the feedback according to the following additive noise model: $s_t(I_t, j)\ell_t(j) + (1 - s_t(I_t, j))\xi_t(j)$, where $\xi_t(j)$ is a zero-mean bounded random variable. Hence, if $s_t(i, j) = 1$, then $I_t = i$ allows to observe the loss of action $j$ without any noise. If $s_t(i, j) = 0$, then only noise is observed. Note that they assume $s_t(i, i) = 1$ for each $i$, implying strong observability. Although similar in spirit to our feedback model, our results do not directly compare with theirs.

Further work also takes into account a time-varying probability for the revelation of side-observations [Kocák et al., 2016a]. While the idea of a general probabilistic feedback graph has been already considered in the stochastic setting [Li et al., 2020, Cortes et al., 2020], the recent work by Ghari and Shen [2022] seems to be the first one in the adversarial setting that generalizes from the Erdős-Rényi model to a more flexible distribution where they allow "edge-specific" probabilities. We remark, however, that the assumptions of Ghari and Shen [2022] exclude some important instances of feedback graphs. For example, we cannot hope to employ their algorithm for efficiently solving the revealing action problem (see for example [Alon et al., 2015]). In a spirit similar to ours, Resler and Mansour [2019] studied a version of the problem where the topology of the graph is fixed and known a priori, but the feedback received by the learner is perturbed when traversing edges.

## 2   Problem Setting

A feedback graph over a set $V = [K]$ of actions is any directed graph $G = (V, E)$, possibly with self-loops. For any vertex $i \in V$, we use $N_G^{\text{in}}(i) = \{j \in V : (j, i) \in E\}$ to denote the in-neighborhood of $i$ and $N_G^{\text{out}}(i) = \{j \in V : (i, j) \in E\}$ to denote its out-neighborhood (we may omit the subscript when the graph is clear from the context). The independence number $\alpha(G)$ of a feedback graph $G$ is the cardinality of the largest subset $S$ of $V$ such that, for all distinct $i, j \in S$, it holds that $(i, j)$ and $(j, i)$ are not in $E$. We also use the following terminology for directed graphs $G = (V, E)$ [Alon et al., 2015]. Any $i \in V$ is: observable if $N_G^{\text{in}}(i) \neq \emptyset$, strongly observable if $i \in N_G^{\text{in}}(i)$ or $V \setminus \{i\} \subseteq N_G^{\text{in}}(i)$, and weakly observable if it is observable but not strongly. The graph $G$ is: observable if all $i \in V$ are observable, strongly observable if all $i \in V$ are strongly observable, and weakly observable if it is observable but not strongly. The weak domination number $\delta(G)$ of $G$ is the cardinality of the smallest subset $S$ of $V$ such that for all weakly observable $i \in V \setminus S$ there exists $j \in S$ such that $(j, i) \in E$.

In the online learning problem with a stochastic feedback graph, an oblivious adversary privately chooses a stochastic feedback graph $\mathcal{G}$ and a sequence $\ell_1, \ell_2, \ldots$ of loss functions $\ell_t \colon V \to [0, 1]$. At each round $t = 1, 2, \ldots$, the learner selects an action $I_t \in V$ to play and, independently, the adversary draws a feedback graph $G_t$ from $\mathcal{G}$ (denoted by $G_t \sim \mathcal{G}$). The learner then incurs loss $\ell_t(I_t)$ and observes the feedback $\left\{ (i, \ell_t(i)) : i \in N_{G_t}^{\mathrm{out}}(I_t) \right\}$. In some cases we consider a richer feedback, where at the end of each round $t$ the learner also observes the realized graph $G_t$. The learner's performance is measured using the standard notion of regret,

$$R_T = \max_{k \in V} \mathbb{E} \left[ \sum_{t=1}^{T} \left( \ell_t(I_t) - \ell_t(k) \right) \right]$$

where $I_1, \ldots, I_T$ are the actions played by the learner, and the expectation is computed over both the sequence $G_1, \ldots, G_T$ of feedback graphs drawn i.i.d. from $\mathcal{G}$ and the learner's internal randomization.

Fix any stochastic feedback graph $\mathcal{G} = \{p(i, j) : i, j \in V\}$. We sometimes use $e$ to denote a pair $(i, j)$, in which case we write $p_e$ to denote the probability $p(i, j)$. When $G_t = (V, E_t)$ is drawn from $\mathcal{G}$, each pair $(i, j) \in V \times V$ independently becomes an edge (i.e., $(i, j) \in E_t$) with probability $p(i, j)$. For any $\varepsilon > 0$, we define the thresholded version $\mathcal{G}_\varepsilon$ of $\mathcal{G}$ represented by $\{p'(i, j) : i, j \in V\}$, where $p'(i, j) = p(i, j)\mathbb{I}_{\{p(i,j) \geq \varepsilon\}}$. We also define the support feedback graph of $\mathcal{G}$ as the graph $\mathrm{supp}(\mathcal{G}) = (V, E)$ having $E = \{(i, j) \in V \times V : p(i, j) > 0\}$. To keep the notation tidy, we write $\alpha(\mathcal{G})$ instead of $\alpha(\mathrm{supp}(\mathcal{G}))$ and similarly for $\delta$.

## 3 Block Decomposition Approach

In this section, we present an algorithm for online learning with stochastic feedback graphs via a reduction to online learning with deterministic feedback graphs. Our algorithm EDGECATCHER (Algorithm 3) has an initial exploration phase followed by a commit phase. In the exploration phase, the edge probabilities are learned online in a round-robin fashion. A carefully designed stopping criterion then triggers the commit phase, where we feed the support of the estimated stochastic feedback graph to an algorithm for online learning with (deterministic) feedback graphs.

### 3.1 Estimating the Edge Probabilities

As a first step we design a routine, ROUNDROBIN (Algorithm 1), that sequentially estimates the stochastic feedback graph until a certain stopping criterion is met. The stopping criterion depends on a nonnegative function $\Phi$ that takes in input a stochastic feedback graph $\mathcal{G}$ together with a time horizon. Let $\hat{\tau} \leq T/K$ be the index of the last iteration of the outer for loop in Algorithm 1. We want to make sure that, for all $\tau \leq \hat{\tau}$, the stochastic feedback graphs $\hat{\mathcal{G}}_\tau$ are valid estimates of the underlying $\mathcal{G}$ up to a $\Theta(\varepsilon_\tau)$ precision. To formalize this notion of approximation, we introduce the following definition.

**Definition 1** ($\varepsilon$-good approximation)**.** *A stochastic feedback graph $\hat{\mathcal{G}} = \{\hat{p}_e : e \in V^2\}$ is an $\varepsilon$-good approximation of $\mathcal{G} = \{p_e : e \in V^2\}$ for some $\varepsilon \in (0, 1]$, if the following holds:*

1. *All the edges $e \in \mathrm{supp}(\mathcal{G})$ with $p_e \geq 2\varepsilon$ belong to $\mathrm{supp}(\hat{\mathcal{G}})$;*
2. *For all edges $e \in \mathrm{supp}(\hat{\mathcal{G}})$ with $p_e \geq \varepsilon/2$ it holds that $|\hat{p}_e - p_e| \leq p_e/2$;*
3. *No edge $e \in V^2$ with $p_e < \varepsilon/2$ belongs to $\mathrm{supp}(\hat{\mathcal{G}})$.*

We can now state the following theorem; we defer the proof in Appendix B. The proof follows from an application of the multiplicative Chernoff bound on edge probabilities.

**Theorem 2.** *If* ROUNDROBIN *(Algorithm 1) is run on the stochastic feedback graph $\mathcal{G}$, then, with probability at least $1 - 1/T$, the estimate $\hat{\mathcal{G}}_\tau$ is an $\varepsilon_\tau$-good approximation of $\mathcal{G}$ simultaneously for all $\tau \leq \hat{\tau}$, where $\hat{\tau} \leq T/K$ is the index of the last iteration of the outer for loop in Algorithm 1.*

---

**Algorithm 1:** ROUNDROBIN

---

**Environment**: stochastic feedback graph $\mathcal{G}$, sequence of losses $\ell_1, \ell_2, \dots, \ell_T$;
**Input:** time horizon $T$, stopping function $\Phi$, actions $V = \{1, 2, \dots, K\}$;
$n_e \leftarrow 0$, for all $e \in V^2$;
**for** each $\tau = 1, 2, \dots, \lfloor T/K \rfloor$ **do**
    **for** each $i = 1, 2, \dots K$ **do**
        Play action $i$ and observe $N_{G_t}^{\text{out}}(i)$ from $G_t \sim \mathcal{G}$;        `// t is the time step`
        $n_e \leftarrow n_e + 1$ for all $e \in N_{G_t}^{\text{out}}(i)$;
    $\hat{p}_e^\tau \leftarrow n_e/\tau$ for all edges $e \in V^2$;
    $\varepsilon_\tau \leftarrow 60 \ln(KT)/\tau$;
    $\hat{\mathcal{G}}_\tau \leftarrow \big(V, \{e \in V^2 : \hat{p}_e^\tau \geq \varepsilon_\tau\}\big)$ with weights $\hat{p}_e^\tau$;    `// estimated feedback graph`
    **if** $\Phi(\hat{\mathcal{G}}_\tau, T) \leq \tau K$ **then**
        **output** $\hat{\mathcal{G}}_\tau, \varepsilon_\tau$;
**output** $\hat{\mathcal{G}}_\tau, \varepsilon_\tau$;

---

## 3.2 Block Decomposition: Reduction to Deterministic Feedback Graph

As a second step, we present BLOCKREDUCTION (Algorithm 2) which reduces the problem of online learning with stochastic feedback graph to the corresponding problem with deterministic feedback graph. Surprisingly enough, in order for this reduction to work, we do not need the exact edge probabilities: an $\varepsilon$-good approximation is sufficient for this purpose.

The intuition behind BLOCKREDUCTION is simple: given that each edge $e$ in $\text{supp}(\mathcal{G}_\varepsilon)$ appears in $G_t$ with probability $p_e \geq \varepsilon$ at each time step $t$, if we wait for $\Theta\big((1/\varepsilon)\ln(T)\big)$ time steps it will appear at least once with high probability. Applying a union bound over all edges, we can argue that considering $\Delta = \Theta\big((1/\varepsilon)\ln(KT)\big)$ realizations of the stochastic feedback graph, then all the edges in $\text{supp}(\mathcal{G}_\varepsilon)$ are realized at least once with high probability.

Imagine now to play a certain action $a$ consistently during a block $B_\tau$ of $\Delta$ time steps. We want to reconstruct the average loss suffered by $a'$ in $B_\tau$:

$$c_\tau(a') = \sum_{t \in B_\tau} \frac{\ell_t(a')}{\Delta} \ , \tag{3}$$

and we want to do it for all $a'$ in the out-neighborhood of $a$. Let $\Delta_{(a,a')}^\tau$ be the number of times that the loss of $a'$ is observed by the learner; i.e., the number of times that $(a, a')$ is realized in the $\Delta$ time steps. With this notation, we can define the natural estimator $\hat{c}_\tau(a')$:

$$\hat{c}_\tau(a') = \sum_{t \in B_\tau} \ell_t(a') \frac{\mathbb{I}_{\{(a,a') \in E_t\}}}{\Delta_{(a,a')}^\tau} \ . \tag{4}$$

Conditioning on the event $\mathcal{E}_{(a,a')}^\tau$ that the edge $(a, a')$ in $\hat{G}$ is observed at least once in block $B_\tau$, we show in Lemma 2 in Appendix B that $\hat{c}_\tau(a')$ is an unbiased estimator of $c_\tau(a')$.

Therefore, we can construct unbiased estimators of the average of the losses on the blocks as if the stochastic feedback graph were deterministic. This allows us to reduce the original problem to that of online learning with deterministic feedback graph on the meta-instance given by the blocks. The details of BLOCKREDUCTION are reported in Algorithm 2, while the theoretical properties are summarized in the next result, whose proof can be found in Appendix B.

**Theorem 3.** *Consider the problem of online learning with stochastic feedback graph $\mathcal{G}$, and let $\hat{\mathcal{G}}$ be an $\varepsilon$-good approximation of $\mathcal{G}$. Let $\mathcal{A}$ be an algorithm for online learning with arbitrary deterministic feedback graph $G$ with regret bound $R_N^{\mathcal{A}}(G)$ over any sequence of $N$ losses in $[0, 1]$. Then, the regret of* BLOCKREDUCTION *(Algorithm 2) run with input $(T, \varepsilon/2, \hat{\mathcal{G}}, \mathcal{A})$ is at most $\Delta R_N^{\mathcal{A}}\big(\text{supp}(\hat{\mathcal{G}})\big) + \Delta$, where $N = \lfloor T/\Delta \rfloor$ and $\Delta = \lceil \frac{4}{\varepsilon} \ln(KT) \rceil$.*

---
**Algorithm 2:** BLOCKREDUCTION
---
**Environment**: stochastic feedback graph $\mathcal{G}$, sequence of losses $\ell_1, \ell_2, \ldots, \ell_T$;

**Input**: time horizon $T$, threshold $\varepsilon$, estimate $\hat{\mathcal{G}}$ of $\mathcal{G}$, learning algorithm $\mathcal{A}$;

$\Delta \leftarrow \lceil \frac{2}{\varepsilon} \ln(KT) \rceil$, $N \leftarrow \lfloor T/\Delta \rfloor$, $\hat{G} \leftarrow \mathrm{supp}(\hat{\mathcal{G}})$;

**Initialize** $\mathcal{A}$ with time horizon $N$ and graph $\hat{G}$;

$B_\tau \leftarrow \{(\tau-1)\Delta + 1, \ldots, \tau\Delta\}$, for all $\tau = 1, \ldots, N$;

**for** each round $\tau = 1, 2, \ldots, N$ **do**

    Draw action $a_\tau$ from the probability distribution over actions output by $\mathcal{A}$;

    **for** each round $t \in B_\tau$ **do**

        Play action $a_\tau$ and observe the revealed feedback;         // $G_t \sim \mathcal{G}$

    For all $a' \in N_{\hat{G}}^{\mathrm{out}}(a_\tau)$, compute $\hat{c}_\tau(a')$ as in (4), and feed them to $\mathcal{A}$;

Play arbitrarily the remaining $T - \Delta N$ rounds;

---

For online learning with deterministic feedback graphs we use the variants of EXP3.G contained in Alon et al. [2015]. Together with Theorem 3, this gives the following corollary; the details of the proof are in Appendix B.

**Corollary 1.** *Consider the problem of online learning with stochastic feedback graph $\mathcal{G}$, and let $\hat{\mathcal{G}}$ be an $\varepsilon$-good approximation of $\mathcal{G}$ for $\varepsilon \geq 1/T$ and with support $\hat{G}$.*

- *If $\hat{G}$ is strongly observable with independence number $\alpha$, then the regret of BLOCKRE-DUCTION run with parameter $\varepsilon/2$ using EXP3.G for strongly observable graphs as base algorithm $\mathcal{A}$ satisfies: $R_T \leq 4C_s \sqrt{(\alpha/\varepsilon)T} \big(\ln(KT)\big)^{3/2}$, where $C_s > 0$ is a constant in the regret bound of $\mathcal{A}$.*
- *If $\hat{G}$ is (weakly) observable with weak domination number $\delta$, then the regret of BLOCKRE-DUCTION run with parameter $\varepsilon/2$ using EXP3.G for weakly observable graphs as base algorithm $\mathcal{A}$ satisfies: $R_T \leq 4C_w (\delta/\varepsilon)^{1/3} \big(\ln(KT)\big)^{2/3} T^{2/3}$, where $C_w > 0$ is a constant in the regret bound of $\mathcal{A}$.*

Note that we can explicitly compute valid constants $C_s = 12 + 2\sqrt{2}$ and $C_w = 8$ directly from the bounds in Alon et al. [2015].

## 3.3 Explore then Commit to a Graph

We are now ready to combine the two routines we presented, ROUNDROBIN and BLOCKRE-DUCTION, in our final learning algorithm, EDGECATCHER. EDGECATCHER has two phases: in the first phase, ROUNDROBIN is used to quickly obtain an $\varepsilon$-good approximation $\hat{\mathcal{G}}$ of the underlying feedback graph $\mathcal{G}$, for a suitable $\varepsilon$. In the second phase, the algorithm commits to $\hat{\mathcal{G}}$ and feeds it to BLOCKREDUCTION. The crucial point is when to commit to a certain (estimated) stochastic feedback graph. If we commit too early, we might not observe a denser support graph, which implies missing out on a richer feedback. If we wait for too long, then the exploration phase ends up dominating the regret. To balance this trade-off, we use the stopping function $\Phi$. This function takes as input a probabilistic feedback graph together with a time horizon and outputs the regret bound that BLOCKREDUCTION would guarantee on this pair. It is defined as

$$\Phi(\mathcal{G}, T) = \min \left\{ 4C_s \sqrt{\frac{\alpha^*}{\varepsilon_s^*} T} \big(\ln(KT)\big)^{3/2}, \; 4C_w \left(\frac{\delta^*}{\varepsilon_w^*} \big(\ln(KT)\big)^2\right)^{1/3} T^{2/3} \right\} \tag{5}$$

for the specific choice of EXP3.G as the learning algorithm $\mathcal{A}$ adopted by BLOCKREDUCTION. Note that the dependence of $\Phi$ on the feedback graph $\mathcal{G}$ is contained in the topological parameters $\alpha^*$ and $\delta^*$ and the corresponding thresholds $\varepsilon_s^*$ and $\varepsilon_w^*$, defined in Equations (1) and (2); see Appendix A for more details on their computation. If we apply $\Phi$ to a stochastic feedback graph that is observable w.p. zero, its value is conventionally set to infinity. Observe that, otherwise, the min is achieved for a specific $\varepsilon^*$ and a specific $\mathcal{G}^* = \mathcal{G}_{\varepsilon^*}$. In Appendix B, we provide a sequence of lemmas (Lemmas 3 and 4 in particular) showing

---

**Algorithm 3:** EDGECATCHER

---

**Environment**: stochastic feedback graph $\mathcal{G}$, sequence of losses $\ell_1, \ell_2, \ldots, \ell_T$;

**Input**: time horizon $T$ and actions $V = \{1, 2, \ldots, K\}$;

Let $\Phi$ defined as in Equation (5);

Run ROUNDROBIN$(T, \Phi, V)$ and obtain $\hat{\mathcal{G}}$ and $\hat{\varepsilon}$;

Compute $\hat{\varepsilon}_s^*$ and $\hat{\varepsilon}_w^*$ for graph $\hat{\mathcal{G}}$ as in Equations (1) and (2);

Let $\hat{\varepsilon}^*$ be the best threshold as in Equation (5);

**if** $\hat{\varepsilon}^* = \hat{\varepsilon}_s^*$ **then** Let $\mathcal{A}$ be EXP3.G for strongly observable feedback graph;

        **else** Let $\mathcal{A}$ be EXP3.G for weakly observable feedback graph;

Let $T' = T - \hat{\tau}K$ be the remaining time steps;          `// ` $\hat{\tau}$ ` as in ` ROUNDROBIN

Run BLOCKREDUCTION$(T', \hat{\varepsilon}^*/2, \hat{\mathcal{G}}_{\hat{\varepsilon}^*}, \mathcal{A})$;

---

that, if ROUNDROBIN outputs an $\varepsilon$-good approximation of the graph, then the regret is bounded by a multiple of the stopping criterion evaluated at $\mathcal{G}$. Combined with Theorem 2, which tells us that ROUNDROBIN does in fact output an $\varepsilon$-good approximation of the graph with high probability, this proves our main result for this section.

**Theorem 4.** *Consider the problem of online learning with stochastic feedback graph $\mathcal{G}$ on $T$ time steps. If* $\mathrm{supp}(\mathcal{G}_{\varepsilon(K,T)})$ *is observable for* $\varepsilon(K,T) = CK^3(\ln(KT))^2/T$ *for a given constant $C > 0$, then there exists an algorithm whose regret $R_T$ satisfies (ignoring polylog factors in $K$ and $T$)* $R_T \leq \min\left\{\sqrt{(\alpha^*/\varepsilon_s^*)T}, \left(\delta^*/\varepsilon_w^*\right)^{1/3}T^{2/3}\right\}.$

## 4   Lower Bounds

In this section, we provide lower bounds that match the regret bound guaranteed by EDGECATCHER, shown in Theorem 4, up to polylogarithmic factors in $K$ and $T$. These lower bounds are valid even if the learner is allowed to observe the realization of the entire feedback graph at every time step, and knows a priori the "correct" threshold $\varepsilon$ to work with. Theorem 5 summarizes the lower bounds whose proofs can be found in Appendix C.

**Theorem 5** (Informal)**.** *Let $\mathcal{A}$ be a possibly randomized algorithm for the online learning problem with stochastic feedback graphs. Consider any directed graph $G = (V, E)$ with $|V| \geq 2$ and any $\varepsilon \in (0, 1]$. There exists a stochastic feedback graph $\mathcal{G}$ with $\mathrm{supp}(\mathcal{G}) = G$ and, for a sufficiently large time horizon $T$, there is a sequence $\ell_1, \ldots, \ell_T$ of loss functions on which the expected regret of $\mathcal{A}$ with respect to the stochastic generation of $G_1, \ldots, G_T \sim \mathcal{G}$ is*

- $\Omega(\sqrt{(\alpha(\mathcal{G}_\varepsilon)/\varepsilon)T})$ *if $G$ is strongly observable;*
- $\tilde{\Omega}((\delta(\mathcal{G}_\varepsilon)/\varepsilon)^{1/3}T^{2/3})$ *if $G$ is weakly observable;*
- $\Omega(T)$ *if $G$ is not observable.*

The lower bound in the non-observable case is the same as Alon et al. [2015, Theorem 6] with a deterministic feedback graph. The remaining lower bounds are nontrivial adaptations of the corresponding bounds for the deterministic case by Alon et al. [2015, 2017]. The main technical hurdle is due to the stochastic nature of the feedback graph, which needs to be taken into account in the proofs. The rationale behind the constructions used for proving the lower bounds is as follows: since each edge is realized only with probability $\varepsilon$, any algorithm requires $1/\varepsilon$ rounds in expectation in order to observe the loss of an action in the out-neighborhood of the played action, whereas one round would suffice with a deterministic feedback graph. Note that Theorem 5 implies that, if $\mathrm{supp}(\mathcal{G}_{\varepsilon(K,T)})$ is not observable for $\varepsilon(K,T)$ as in Theorem 4, then there is no hope to achieve sublinear regret, as the lower bounds for both strongly and weakly observable supports are linear in $T$ for all $\varepsilon \leq \varepsilon(K,T)$.

## 5   Refined Graph-Theoretic Parameters

Although the results from Section 3 are worst-case optimal up to log factors, we may find that the factors $\sqrt{\alpha(G_\varepsilon)/\varepsilon}$ and $(\delta(G_\varepsilon)/\varepsilon)^{1/3}$ for strongly and weakly observable $\mathrm{supp}(\mathcal{G}_\varepsilon) = G_\varepsilon$,

respectively, may be improved upon in certain cases. In particular, we show that, under additional assumptions on the feedback that we receive, we can obtain better regret bounds. To understand our results, we need some initial definitions. The weighted independence number for a graph $H = (V, E)$ and positive vertex weights $w(i)$ for $i \in V$ is defined as

$$\alpha_{\mathsf{w}}(H, w) = \max_{S \in \mathcal{I}(H)} \sum_{i \in S} w(i) \ ,$$

where $\mathcal{I}(H)$ denotes the family of independent sets in $H$. We consider two different weight assignments computed in terms of any stochastic feedback graph $\mathcal{G}$ with edge probabilities $p(i, j)$ and $\mathrm{supp}(\mathcal{G}) = G$. They are defined as $w_{\mathcal{G}}^-(i) = \left(\min_{j \in N_G^{\mathrm{in}}(i)} p(j, i)\right)^{-1}$ and $w_{\mathcal{G}}^+(i) = \left(\min_{j \in N_G^{\mathrm{out}}(i)} p(i, j)\right)^{-1}$. Then, the two corresponding weighted independence numbers are $\alpha_{\mathsf{w}}^-(\mathcal{G}) = \alpha_{\mathsf{w}}(G, w_{\mathcal{G}}^-)$ and $\alpha_{\mathsf{w}}^+(\mathcal{G}) = \alpha_{\mathsf{w}}(G, w_{\mathcal{G}}^+)$. The parameter of interest for the results in this section is $\alpha_{\mathsf{w}}(\mathcal{G}) = \alpha_{\mathsf{w}}^-(\mathcal{G}) + \alpha_{\mathsf{w}}^+(\mathcal{G})$. For more details on the weighted independence number see Appendix E. We also use the following definitions of weighted weak domination number $\delta_{\mathsf{w}}$ for a graph $H$ and positive vertex weights $w$, and self-observability parameter $\sigma$:

$$\delta_{\mathsf{w}}(H, w) = \min_{D \in \mathcal{D}(H)} \sum_{i \in D} w(i) \ , \qquad \sigma(\mathcal{G}) = \sum_{i : i \in N_G^{\mathrm{in}}(i)} (p(i, i))^{-1} \ ,$$

where $\mathcal{D}(H)$ denotes the family of weakly dominating sets in $H$. In this section, we focus on the weighted weak domination number $\delta_{\mathsf{w}}(\mathcal{G}) = \delta_{\mathsf{w}}(G, w_{\mathcal{G}}^+)$. To gain some intuition on the graph-theoretic parameters introduced above, consider the graph with only self-loops, also used in Example 1 below. If all $p(i, i) = \varepsilon$, the learner needs to pull a single arm $1/\varepsilon$ times for one observation in expectation, and $K/\varepsilon$ times to see the losses of all arms once. However, when the edge probabilities are different we need to pull arms for $\sum_{i=1}^K 1/p(i, i)$ times. The weighted independence number, weighted weak domination and self-observability generalize this intuition and tell us how many observations the learner needs to see all losses at least once in expectation. We now state the main result of this section.

**Theorem 6** (Informal)**.** *There exists an algorithm with per-round running time of $O(K^4)$ and whose regret is bounded (ignoring logarithmic factors) by*

$$\min\left\{T, \min_{\varepsilon}\left\{\sqrt{\alpha_{\mathsf{w}}(\mathcal{G}_\varepsilon)T} \ : \ \mathrm{supp}(\mathcal{G}_\varepsilon) \text{ is strongly observable}\right\},\right.$$
$$\left.\min_{\varepsilon}\left\{(\delta_{\mathsf{w}}(\mathcal{G}_\varepsilon))^{1/3} T^{2/3} + \sqrt{\sigma(\mathcal{G}_\varepsilon)T} \ : \ \mathrm{supp}(\mathcal{G}_\varepsilon) \text{ is observable}\right\}\right\} \ ,$$

The regret bound in Theorem 6 follows from Theorem 11 in Appendix D. The running time bound is determined by approximating $\delta_{\mathsf{w}}$ for all $K^2$ possible thresholds. In each of the thresholded graphs, we can compute a $(\ln(K) + 1)$-approximation for the weighted weak domination number in $O(K^2)$ time by reduction to set cover [Vazirani, 2001]. Doing so only introduces an extra factor of order $(\ln(K))^{1/3}$ in the regret bound.

An important property of the bound in Theorem 6 is that it is never worse than the bounds obtained before. The following example shows that the regret bound in Theorem 6 can also be better than previously obtained regret bounds.

**Example 1** (Faulty bandits)**.** Consider a stochastic feedback graph $\mathcal{G}$ for the $K$-armed bandit setting: $p(i, i) = \varepsilon_i \in (0, 1]$ for all $i \in V$ and $p(i, j) = 0$ for all $i \neq j$. In this case, the regret of EDGECATCHER is $\tilde{O}\left(\sqrt{KT/(\min_i \varepsilon_i)}\right)$. On the other hand, Theorem 6 provides the bound $\tilde{O}\left(\sqrt{T \sum_i(1/\varepsilon_i)}\right)$, as $\alpha_{\mathsf{w}}(\mathcal{G}) = 2 \sum_i 1/\varepsilon_i$. In the special case when $\varepsilon_i = \varepsilon \in (0, 1]$ for some $i \in V$ while $\varepsilon_j = 1$ for all $j \neq i$, the regret of EDGECATCHER is $\tilde{O}(\sqrt{KT/\varepsilon})$, while Theorem 6 guarantees a $\tilde{O}(\sqrt{(K + 1/\varepsilon)T})$ regret bound. $\qquad\square$

To derive these tighter bounds, we exploit the additional assumption that the realized feedback graph $G_t$ is observed at the end of each round. This allows us to *simultaneously* estimate the feedback graph and control the regret, rather than performing these two tasks sequentially as in Section 3. In particular, we use this extra information to construct a novel

importance-weighted estimator for the loss, which for $t > 1$ is defined to be

$$\tilde{\ell}_t(i) = \frac{\ell_t(i)}{\hat{P}_t(i)} \mathbb{I}_{\{i \in N_{G_t}^{\text{out}}(I_t)\} \cap \{i \in N_{\hat{G}_t}^{\text{out}}(I_t)\}} \ , \tag{6}$$

where $\hat{P}_t(i) = \sum_{j \in N_{\hat{G}_t}^{\text{in}}(i)} \pi_t(j) \hat{p}_t(j,i)$ is the estimated probability of observing the loss of arm $i$ at round $t$, $\pi_t(i)$ is the distribution we sample $I_t$ from, and $\hat{G}_t$ is the support of the estimated graph $\hat{\mathcal{G}}_t$. Note that we ignore losses that we receive due to missing edges in $\hat{G}_t$. We show that we pay an additive term in the regret for wrongly estimating an edge, which is why it is important to control which edges are in $\hat{G}_t$. Ideally, we would use $P_t(i) = \sum_{j \in N_{\hat{G}_t}^{\text{in}}(i)} \pi_t(j) p(j,i)$ rather than $\hat{P}_t(i)$, as this is the true probability of observing the loss of arm $i$ in round $t$. However, since we do not have access to $p(j,i)$, we use instead an upper confidence estimate of $p(j,i)$ for rounds $t \geq 2$ given by

$$\hat{p}_t(j,i) = \tilde{p}_t(j,i) + \sqrt{\frac{2\tilde{p}_t(j,i)}{t-1} \ln(3K^2 T^2)} + \frac{3}{t-1} \ln(3K^2 T^2) \ ,$$

where $\tilde{p}_t(j,i) = \frac{1}{t-1} \sum_{s=1}^{t-1} \mathbb{I}_{\{(j,i) \in E_s\}}$. We denote by $\hat{\mathcal{G}}_t^{\text{UCB}}$ the stochastic graph with edge probabilities $\hat{p}_t(j,i)$. Note that the support of $\hat{\mathcal{G}}_t^{\text{UCB}}$ is a complete graph because $\hat{p}_t(j,i) > 0$ for all $(j,i) \in V \times V$. These estimators for the edge probabilities are sufficiently good for our purposes whenever event $\mathcal{K}$ occurs, which we define as the event that, for all $t \geq 2$,

$$|\tilde{p}_t(j,i) - p(j,i)| \leq \sqrt{\frac{2\tilde{p}_t(j,i)}{t-1} \ln(3K^2 T^2)} + \frac{3}{t-1} \ln(3K^2 T^2), \quad \forall (j,i) \in V \times V \ .$$

An important property of $\tilde{\ell}_t$ can be found in Lemma 1 below. It tells us that we may treat $\tilde{\ell}_t$ as if event $\mathcal{K}$ is always realized, i.e., $\hat{p}_t(j,i)$ is always an upper bound estimator on $p(j,i)$. The proof of Lemma 1 is implied by Lemma 6 in Appendix D.

**Lemma 1** (Informal). *Let $e_k$ denote the basis vector with $e_k(i) = \mathbb{I}_{\{i=k\}}$ as $i$-th entry for each $i \in [K]$. The loss estimate $\tilde{\ell}_t$ defined in (6) satisfies*

$$R_T = \tilde{O}\left( \mathbb{E}\left[ \sum_{t=2}^{T} \sqrt{\sum_{i=1}^{K} \frac{\pi_t(i)}{(t-1)\hat{P}_t(i)}} \ \middle| \ \mathcal{K} \right] + \max_{k \in V} \mathbb{E}\left[ \sum_{t=2}^{T} \sum_{i=1}^{K} \big(\pi_t(i) - e_k(i)\big) \tilde{\ell}_t(i) \ \middle| \ \mathcal{K} \right] \right). \tag{7}$$

Lemma 1 shows that we only suffer $\tilde{O}\left(\sqrt{\sum_{t=2}^{T} \sum_{i=1}^{K} \frac{\pi_t(i)}{P_t(i)}}\right)$ additional regret compared to when we know $p(j,i)$. Lemma 1 also shows that $\tilde{\ell}_t$ behaves nicely in the sense that, conditioned on $\mathcal{K}$, we have $\tilde{\ell}_t(i) \leq \frac{\ell_t(i)}{P_t(i)} \mathbb{I}_{\{i \in N_{G_t}^{\text{out}}(I_t)\} \cap \{i \in N_{\hat{G}_t}^{\text{out}}(I_t)\}}$. This is a crucial property to bound the regret of our algorithm. We show that, with a modified version of EXP3.G [Alon et al., 2015], the second sum on the right-hand side of (7) is bounded by a term of order $\sqrt{\sum_{t=2}^{T} \sum_{i=1}^{K} \frac{\pi_t(i)}{P_t(i)}}$, meaning that the regret is also bounded similarly. Our final step is to prove that the above term is bounded in terms of the minimum of the weighted independence number and the weighted weak domination number plus self-observability.

## Acknowledgments and Disclosure of Funding

Nicolò Cesa-Bianchi, Federico Fusco and Dirk van der Hoeven gratefully acknowledge partial support by the MIUR PRIN grant Algorithms, Games, and Digital Markets (ALGADIMAR). Nicolò Cesa-Bianchi and Emmanuel Esposito were also supported by the EU Horizon 2020 ICT-48 research and innovation action under grant agreement 951847, project ELISE, and by the project "One Health Action Hub: University Task Force for the resilience of territorial ecosystems" funded by Università degli Studi di Milano. Federico Fusco was also supported by the ERC Advanced Grant 788893 AMDROMA "Algorithmic and Mechanism Design Research in Online Markets".

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
