# A  On the Computation of the Optimal Probability Thresholds

The tasks of finding the independence number and (weak) domination number in a graph are notoriously NP-hard problems. In particular, while for the domination number, by a reduction to set cover, a simple greedy approach yields a logarithmic (in the number $K$ of nodes) approximation [Vazirani, 2001], for the independence number it is known that even computing a $K^{1-\epsilon}$-approximation is hard, for any $\epsilon > 0$ [Håstad, 1999, Zuckerman, 2007].

Our algorithm OptimisticThenCommitGraph solves these computational aspects directly, whereas the hardness of finding $\alpha^*$ and $\delta^*$ may limit the applicability of EdgeCatcher in instances with a large and complex action space. In fact, the computation of the stopping function $\Phi$ involves finding the best thresholds $\varepsilon_s^*$ and $\varepsilon_w^*$, defined in Equations (1) and (2), and therefore repeatedly solving NP-hard problems. In what follows, we present some observations that clarify to which extent (and at which cost) EdgeCatcher can still be implemented efficiently.

First, it is important to note that our algorithm is robust with respect to approximate knowledge of the topological parameters: the definition of $\Phi$ can be tweaked as to consider the approximation factor at the cost of having the same factor showing up in the regret bound (with the same order as the approximated graph parameter). This partly solves the problem for weakly observable graphs (as the $(\ln(K) + 1)$-approximation only gives and extra polylog$(K)$ in the regret) and for the classes of graphs where it is possible to efficiently compute good approximations of the independence number, e.g., planar graphs [Baker, 1994] or bounded-degree graphs [Halldórsson and Radhakrishnan, 1997].

Another approach consists in considering the fractional solutions of the independence and domination number linear programs. While for the former we obtain an approximation given by the integrality gap, for the latter we can show a tight dependence on the fractional weak domination number (thus improving the regret bound), as in Chen et al. [2021].

Furthermore, note that it is always possible to ignore the $\alpha$ and $\delta$ terms in the definition of $\Phi$; it is not hard to see that such an approach yields a regret bound (ignoring polylog terms) of the type $\min\{\sqrt{(K/\varepsilon_1)T}, (K/\varepsilon_2)^{1/3}T^{2/3}\}$, where $\varepsilon_1$, respectively $\varepsilon_2$, is the largest $\varepsilon$ such that supp$(\mathcal{G}_\varepsilon)$ is strongly, respectively weakly, observable. Although suboptimal, this drastic approach gives a regret bound with an optimal dependence on the $T$ and $\varepsilon$ terms (as $\varepsilon_s^* \leq \varepsilon_1$ and $\varepsilon_w^* \leq \varepsilon_2$).

Finally, we conclude by discussing how it is possible to drastically reduce the number of times that EdgeCatcher calls the routine to compute $\alpha$ and $\delta$, at the cost of losing a small multiplicative factor in the regret. Crucially, we do not need to check the stopping condition involving $\Phi$ in every single round: it suffices to do so for a logarithmic number of times. Assume, in fact, to check the stopping condition in RoundRobin only when $\tau$ is a power of 2, i.e., $\tau = 2^b$ for some integer $b$. This single check covers all rounds $\tau'$ such that $\tau/2 = 2^{b-1} \leq \tau' \leq 2^b = \tau$. On the stochastic graph estimate $\hat{\mathcal{G}}_\tau$ we can compute $\alpha_{\varepsilon_\tau}/\varepsilon_\tau$ and $\delta_{\varepsilon_\tau}/\varepsilon_\tau$, which are also 2-approximations for the best respective ratios on any thresholded graph corresponding to rounds of RoundRobin between $\tau/2$ and $\tau$ (note that such an approach would also improve the dependency of $\varepsilon_\tau$ and $\Delta$ on $T$ in Theorems 2 and 3, and thus in the regret bound, from $\ln(T)$ down to $\ln(\ln(T))$ due to an improved union bound).

# B  Missing Results from Section 3

## B.1  Proof of Theorem 2

**Theorem 2.** *If* RoundRobin *(Algorithm 1) is run on the stochastic feedback graph $\mathcal{G}$, then, with probability at least $1 - 1/T$, the estimate $\hat{\mathcal{G}}_\tau$ is an $\varepsilon_\tau$-good approximation of $\mathcal{G}$ simultaneously for all $\tau \leq \hat{\tau}$, where $\hat{\tau} \leq T/K$ is the index of the last iteration of the outer for loop in Algorithm 1.*

*Proof of Theorem 2.* For all edges $e$ and time steps $\tau \leq \hat{\tau}$, we define the following two events: the event $\mathcal{E}_e^\tau = \{\hat{p}_e^\tau \geq \varepsilon_\tau\}$ that $e$ belongs to the support of $\hat{\mathcal{G}}_\tau$, and the event

$\mathcal{F}_e^\tau = \{|\hat{p}_e^\tau - p_e| \leq p_e/2\}$ that $\hat{p}_e^\tau$ is well estimated. For all $\tau \leq \hat{\tau}$, we also define large and small edges in $E$ according to their probabilities: $E_\tau^+ = \{e \in V^2 : p_e \geq 2\varepsilon_\tau\}$ and $E_\tau^- = \{e \in V^2 : p_e < \varepsilon_\tau/2\}$.

First, we look at the complementary event of $\mathcal{E}_e^\tau$ for any $\tau \leq \hat{\tau}$ and $e \in E_\tau^+$. We have:

$$\mathbb{P}\left(\overline{\mathcal{E}}_e^\tau\right) = \mathbb{P}\left(\hat{p}_e^\tau < \varepsilon_\tau\right) \leq \mathbb{P}\left(\hat{p}_e^\tau \leq p_e/2\right) = \mathbb{P}\left(\hat{p}_e^\tau - p_e \leq -p_e/2\right) \leq \mathrm{e}^{-\frac{\tau}{8}p_e} \leq \mathrm{e}^{-\frac{\tau}{4}\varepsilon_\tau} \leq \frac{1}{4KT^2} \ .$$

Note that in the first and second to last inequalities we used the fact that $p_e \geq 2\varepsilon_\tau$, in the last inequality the definition of $\varepsilon_\tau$ and the fact that $K \geq 2$, while in the second inequality we applied the Chernoff lower bound (multiplicative version, see Mitzenmacher and Upfal [2005, part 2 of Theorem 4.5]) on the estimator $\hat{p}_e^\tau$.

If we call $\mathcal{E}$ the event corresponding to part 1 of Definition 1, we have the following:

$$\mathbb{P}\left(\mathcal{E}\right) = \mathbb{P}\left(\bigcap_{\tau \leq \hat{\tau}} \bigcap_{e \in E_\tau^+} \mathcal{E}_e^\tau\right) \geq 1 - \sum_{\tau \leq \hat{\tau}} \sum_{e \in E_\tau^+} \mathbb{P}\left(\hat{p}_e^\tau < \varepsilon_\tau\right) \geq 1 - \sum_{\tau \leq \hat{\tau}} \frac{|E_\tau^+|}{4KT^2} \geq 1 - \frac{1}{4T} \ , \quad (8)$$

where we used that $|E_\tau^+| \leq K^2$ for all $\tau \leq \hat{\tau} \leq T/K$ with probability 1.

Next, we study the complementary event of $\mathcal{F}_e^\tau$ for $e \notin E_\tau^-$. For such $e$ and any $\tau \leq \hat{\tau}$, we can directly use the two-sided Chernoff bound (multiplicative version, as in Mitzenmacher and Upfal [2005, Corollary 4.6]) on the estimator $\hat{p}_e^\tau$:

$$\mathbb{P}\left(\overline{\mathcal{F}}_e^\tau\right) = \mathbb{P}\left(|\hat{p}_e^\tau - p_e| > \frac{1}{2}p_e\right) \leq 2\mathrm{e}^{-\frac{\tau}{12}p_e} \leq 2\mathrm{e}^{-\frac{\tau}{24}\varepsilon_\tau} \leq \frac{1}{2KT^2} \ .$$

Note that we used the definition of $\varepsilon_\tau$ and the facts that $2p_e \geq \varepsilon_\tau$ and $K, T \geq 2$. Now, if we call $\mathcal{F}$ the event corresponding to part 2 of Definition 1, we can proceed via union bounding as in Equation (8) and get

$$\mathbb{P}\left(\mathcal{F}\right) = \mathbb{P}\left(\bigcap_{\tau \leq \hat{\tau}} \bigcap_{e \notin E_\tau^-} \mathcal{F}_e^\tau\right) \geq 1 - \frac{1}{2T} \ . \quad (9)$$

As a third step, we get back to the $\mathcal{E}_e^\tau$ events, but we consider $e \in E_\tau^-$. For $\tau \leq \hat{\tau}$ and $e \in E_\tau^-$ we have:

$$\mathbb{P}\left(\mathcal{E}_e^\tau\right) = \mathbb{P}\left(\hat{p}_e^\tau \geq \varepsilon_\tau\right) \leq \mathbb{P}\left(\hat{p}_e^\tau - p_e \geq \frac{1}{2}\varepsilon_\tau\right) = \mathbb{P}\left(\hat{p}_e^\tau - p_e \geq xp_e\right),$$

where we used $p_e < \varepsilon_\tau/2$ and named $x = \varepsilon_\tau/(2p_e) > 1$. At this point we can use the Chernoff upper bound (multiplicative version, see Mitzenmacher and Upfal [2005, part 1 of Theorem 4.4] with $\delta = x$) and obtain:

$$\mathbb{P}\left(\mathcal{E}_e^\tau\right) \leq \mathbb{P}\left(\hat{p}_e^\tau - p_e \geq xp_e\right) \leq \left(\frac{\mathrm{e}^x}{(1+x)^{1+x}}\right)^{\tau p_e} \leq \mathrm{e}^{-\frac{\tau}{3}xp_e} = \mathrm{e}^{-\frac{\tau}{6}\varepsilon_\tau} \leq \frac{1}{4KT^2} \ .$$

The third inequality follows from $2x/(2+x) \leq \ln(1+x)$ which holds for all positive $x$:

$$\frac{\mathrm{e}^x}{(1+x)^{1+x}} = \mathrm{e}^{x-(1+x)\ln(1+x)} \leq \mathrm{e}^{-x^2/(2+x)} \leq \mathrm{e}^{-x/3}, \quad \forall x \geq 1 \ .$$

If we now call $\mathcal{C}$ the event described in part 3 of Definition 1, we get, using the bound on $\mathbb{P}\left(\mathcal{E}_e^\tau\right)$ and a union bound as in Equations (8) and (9):

$$\mathbb{P}\left(\mathcal{C}\right) = \mathbb{P}\left(\bigcap_{\tau \leq \hat{\tau}} \bigcap_{e \in E_\tau^-} \overline{\mathcal{E}}_e^\tau\right) \geq 1 - \frac{1}{4T} \ . \quad (10)$$

The theorem then follows by a union bound on the complementary events of $\mathcal{E}, \mathcal{F}$ and $\mathcal{C}$. $\quad\square$

## B.2 Proof of Theorem 3

In order to prove the regret bound achieved by BLOCKREDUCTION, we need to show that it is able to compute unbiased estimators for the average loss of observed actions within each time block. This property is guaranteed as long as the learner plays consistently a same action within each time block, and conditioned on the event that each action in the support out-neighborhood of the chosen action is observed at least once in the respective time block (depending on the realizations of the feedback graph).

**Lemma 2.** *Let $G = \text{supp}(\mathcal{G})$ and $c_\tau$ and $\hat{c}_\tau$ defined as in Equations (3) and (4). For each block $B_\tau$, if the learner plays consistently action $a$, then for each $a' \in N_G^{\text{out}}(a)$ the estimators $\hat{c}_\tau(a')$ are unbiased under $\mathcal{E}_{(a,a')}^\tau$:*

$$\mathbb{E}\left[\hat{c}_\tau(a') \,\Big|\, \mathcal{E}_{(a,a')}^\tau\right] = c_\tau(a') , \quad \forall a' \in N_G^{\text{out}}(a) .$$

*Proof.* Recall that $\mathcal{E}_{(a,a')}^\tau$ is the event that the edge $(a, a')$ in $G$ is observed at least once in block $B_\tau$. Substituting the definition (4) of the estimator, we can write

$$\mathbb{E}\left[\hat{c}_\tau(a') \,\Big|\, \mathcal{E}_{(a,a')}^\tau\right] = \sum_{t \in B_\tau} \ell_t(a') \mathbb{E}\left[\frac{\mathbb{I}_{\{(a,a') \in E_t\}}}{\Delta_{(a,a')}^\tau} \,\Bigg|\, \mathcal{E}_{(a,a')}^\tau\right] .$$

Now we just need to prove that the expectation in the right-hand side is equal to $1/\Delta$:

$$\mathbb{E}\left[\frac{\mathbb{I}_{\{(a,a') \in E_t\}}}{\Delta_{(a,a')}^\tau} \,\Bigg|\, \mathcal{E}_{(a,a')}^\tau\right] = \sum_{r=1}^{\Delta} \mathbb{E}\left[\frac{\mathbb{I}_{\{(a,a') \in E_t\}}}{r} \,\Big|\, \Delta_{(a,a')}^\tau = r\right] \mathbb{P}\left(\Delta_{(a,a')}^\tau = r \,\Big|\, \mathcal{E}_{(a,a')}^\tau\right)$$

$$= \sum_{r=1}^{\Delta} \frac{1}{r} \mathbb{P}\left((a, a') \in E_t \,\Big|\, \Delta_{(a,a')}^\tau = r\right) \mathbb{P}\left(\Delta_{(a,a')}^\tau = r \,\Big|\, \mathcal{E}_{(a,a')}^\tau\right)$$

$$= \frac{1}{\Delta} \sum_{r=1}^{\Delta} \mathbb{P}\left(\Delta_{(a,a')}^\tau = r \,\Big|\, \mathcal{E}_{(a,a')}^\tau\right) = \frac{1}{\Delta} .$$

Note that in the third equality we used the fact that, conditioned on $\Delta_{(a,a')}^\tau = r > 0$, the $r$ time steps when $(a, a') \in E_t$ are distributed uniformly at random in the $\Delta$ time steps. $\square$

We can now prove the regret bound of BLOCKREDUCTION in Theorem 3, which we restate below. Its regret depends on the performance of the algorithm $\mathcal{A}$ used on the meta-instance derived from the blocks reduction.

**Theorem 3.** *Consider the problem of online learning with stochastic feedback graph $\mathcal{G}$, and let $\hat{\mathcal{G}}$ be an $\varepsilon$-good approximation of $\mathcal{G}$. Let $\mathcal{A}$ be an algorithm for online learning with arbitrary deterministic feedback graph $G$ with regret bound $R_N^{\mathcal{A}}(G)$ over any sequence of $N$ losses in $[0, 1]$. Then, the regret of BLOCKREDUCTION (Algorithm 2) run with input $(T, \varepsilon/2, \hat{\mathcal{G}}, \mathcal{A})$ is at most $\Delta R_N^{\mathcal{A}}\big(\text{supp}(\hat{\mathcal{G}})\big) + \Delta$, where $N = \lfloor T/\Delta \rfloor$ and $\Delta = \lceil \frac{4}{\varepsilon} \ln(KT) \rceil$.*

*Proof of Theorem 3.* Consider the partition of the $T$ time steps into $N$ blocks $B_1, \ldots, B_N$ of equal size $\Delta$ and let $\mathcal{E}$ be the clean event, corresponding to all edges $e$ in the graph $\text{supp}(\hat{\mathcal{G}}) = \hat{G} = (V, E)$ being realized at least once in each block. Formally, $\mathcal{E} = \bigcap_{\tau=1}^{N} \bigcap_{e \in E} \mathcal{E}_e^\tau$, where $\mathcal{E}_e^\tau$ are defined as in the proof of Lemma 2. By Definition 1 (part 3), all the edges $e \in E$ have a probability $p_e$ in $\mathcal{G}$ that is at least $\varepsilon/2$. Thus, it is immediate to verify that

$$\mathbb{P}\left(\mathcal{E}_e^\tau\right) = 1 - (1 - p_e)^\Delta \geq 1 - \left(1 - \frac{\varepsilon}{2}\right)^\Delta \geq 1 - e^{-\varepsilon\Delta/2} \geq 1 - \frac{1}{K^2 T^2}$$

holds for any edge $e \in E$ using our choice of $\Delta$. We show by union bound that the probability any of these edges never realizes in some block is

$$\mathbb{P}\left(\bigcup_{\tau \leq N} \bigcup_{e \in E} \overline{\mathcal{E}}_e^\tau\right) \leq \sum_{\tau \leq N} \sum_{e \in E} \mathbb{P}\left(\overline{\mathcal{E}}_e^\tau\right) \leq \frac{1}{T} ,$$

where we used that there are at most $K^2$ directed edges (including self-loops) in $\hat{G}$ and we substituted the chosen values of $N$ and $\Delta$.

We can then bound the overall regret $R_T$ as follows:

$$R_T \leq \mathbb{E}\left[\sum_{t=1}^{T} \ell_t(I_t) \,\middle|\, \mathcal{E}\right] - \min_k \sum_{t=1}^{T} \ell_t(k) + T \cdot \mathbb{P}\left(\overline{\mathcal{E}}\right) + (T - \Delta N) \ . \tag{11}$$

Note that the final term is an upper bound to the regret in the final time steps of the algorithm. We just showed that $\mathbb{P}\left(\overline{\mathcal{E}}\right)$ is smaller than $1/T$. This, together with the fact that $T - \Delta N$ is at most $\Delta - 1$, gives the additive $\Delta$ we have in the final statement.

We now focus on the remaining term, which corresponds to the regret conditioned on $\mathcal{E}$. It is equal to

$$\mathbb{E}\left[\sum_{t=1}^{T} \ell_t(I_t) \,\middle|\, \mathcal{E}\right] - \min_k \sum_{t=1}^{T} \ell_t(k) = \Delta \cdot \left(\mathbb{E}\left[\sum_{\tau=1}^{N} \sum_{t \in B_\tau} \frac{\ell_t(I_\tau)}{\Delta} \,\middle|\, \mathcal{E}\right] - \min_k \sum_{\tau=1}^{N} \sum_{t \in B_\tau} \frac{\ell_t(k)}{\Delta}\right)$$

$$= \Delta \cdot \left(\mathbb{E}\left[\sum_{\tau=1}^{N} c_\tau(I_\tau) \,\middle|\, \mathcal{E}\right] - \min_k \sum_{\tau=1}^{N} c_\tau(k)\right) \ , \tag{12}$$

where, we recall it, $c_\tau(i)$ is the average loss of action $i$ in block $B_\tau$. Indeed, our algorithm chooses the same action $I_t = I_\tau$ for all time steps $t \in B_\tau$, and the decision is based on algorithm $\mathcal{A}$.

Consider now the loss estimates $\hat{c}_1, \ldots, \hat{c}_N$ that we provide to algorithm $\mathcal{A}$. These estimates are such that $\mathbb{E}\left[\hat{c}_\tau(i) \,\middle|\, \mathcal{E}\right] = c_\tau(i)$ by Lemma 2. Note that conditioning on $\mathcal{E}$ instead that on the single $\mathcal{E}_e^\tau$ does not affect the fact that the estimators are unbiased: this is due to the fact that the edge realizations are independent from the losses and the strategy of the learner.

Therefore, letting $k^*$ be the action minimizing $c_1(k) + \cdots + c_T(k)$ over $k = 1, \ldots, K$,

$$\mathbb{E}\left[\sum_{\tau=1}^{N} c_\tau(I_\tau) \,\middle|\, \mathcal{E}\right] - \min_k \sum_{\tau=1}^{N} c_\tau(k) = \mathbb{E}\left[\sum_{\tau=1}^{N} \hat{c}_\tau(I_\tau) - \sum_{\tau=1}^{N} \hat{c}_\tau(k^*) \,\middle|\, \mathcal{E}\right] \leq R_N^{\mathcal{A}}(\hat{G}) \ , \tag{13}$$

where $R_N^{\mathcal{A}}(\hat{G})$ is the regret bound of algorithm $\mathcal{A}$ given losses $\hat{c}_1, \ldots, \hat{c}_N$ and feedback graph $\hat{G} = \mathrm{supp}(\hat{\mathcal{G}})$. Finally, substituting Equations (12) and (13) into Equation (11) yields the desired bound. $\qquad\square$

## B.3   Proof of Corollary 1

**Corollary 1.** *Consider the problem of online learning with stochastic feedback graph $\mathcal{G}$, and let $\hat{\mathcal{G}}$ be an $\varepsilon$-good approximation of $\mathcal{G}$ for $\varepsilon \geq 1/T$ and with support $\hat{G}$.*

- *If $\hat{G}$ is strongly observable with independence number $\alpha$, then the regret of* BLOCKRE-DUCTION *run with parameter $\varepsilon/2$ using* EXP3.G *for strongly observable graphs as base algorithm $\mathcal{A}$ satisfies: $R_T \leq 4C_s\sqrt{(\alpha/\varepsilon)T}\left(\ln(KT)\right)^{3/2}$, where $C_s > 0$ is a constant in the regret bound of $\mathcal{A}$.*
- *If $\hat{G}$ is (weakly) observable with weak domination number $\delta$, then the regret of* BLOCKRE-DUCTION *run with parameter $\varepsilon/2$ using* EXP3.G *for weakly observable graphs as base algorithm $\mathcal{A}$ satisfies: $R_T \leq 4C_w(\delta/\varepsilon)^{1/3}\left(\ln(KT)\right)^{2/3}T^{2/3}$, where $C_w > 0$ is a constant in the regret bound of $\mathcal{A}$.*

*Proof of Corollary 1.* The statement follows from Theorem 3, the assumption on $\varepsilon$ (which lets us safely handle the additive $\Delta$ term), and the fact that EXP3.G achieves regret $R_N^{\mathcal{A}} \leq C_s\sqrt{\alpha N}\ln(KN)$ on strongly observable graphs, and regret $R_N^{\mathcal{A}} \leq C_w(\delta \ln K)^{1/3}N^{2/3}$ on (weakly) observable graphs. $\qquad\square$

## B.4 Proof of Theorem 4

To prove Theorem 4 we first need two preliminary lemmata. In Lemma 3 we present some generic properties of the stopping function $\Phi(\mathcal{G}, T)$, while in Lemma 4 we prove that $\Phi(\mathcal{G}, T - \hat{\tau}K)$ is indeed the regret obtained in BLOCKREDUCTION after the stopping condition in ROUNDROBIN is triggered.

**Lemma 3.** *Let $\mathcal{G}$ be a stochastic feedback graph such that $\Phi(\mathcal{G}, T) \neq \infty$, and let $\varepsilon^*$ be the threshold where the $\arg\min$ in the definition of $\Phi(\mathcal{G}, T)$ is attained. Consider a run of the algorithm EDGECATCHER where ROUNDROBIN does not fail while using the stopping function $\Phi$ defined in Equation* (5). *We have the following:*

*(i)* $\Phi(\hat{\mathcal{G}}_{\tau'}, T) \leq 2\Phi(\hat{\mathcal{G}}_{\tau}, T)$, *for all $\tau, \tau'$ such that $\tau \leq \tau' \leq \hat{\tau}$,*
*(ii)* $\Phi(\hat{\mathcal{G}}_{\tau}, T) \leq \sqrt{2}\Phi(\mathcal{G}, T)$ *for all $\tau$ such that $120 \ln(KT)/\varepsilon^* \leq \tau \leq \hat{\tau}$ (if such $\tau$ exists),*

*where $\hat{\tau} \leq \lfloor T/K \rfloor$ is the index of the last iteration of the outer for loop in Algorithm 1.*

*Proof.* We consider a run of EDGECATCHER where ROUNDROBIN does not fail. This means that all the $\hat{\mathcal{G}}_{\tau}$ are $\varepsilon_{\tau}$-good approximation of $\mathcal{G}$, for all $\tau \leq \hat{\tau}$. Focus on the first part of the statement. All edges in $\mathrm{supp}(\hat{\mathcal{G}}_{\tau})$ are contained in $\mathrm{supp}(\hat{\mathcal{G}}_{\tau'})$ since ROUNDROBIN does not fail. This implies that the observability regime only improves as $\tau$ increases. We have two cases: if the best threshold for $\hat{\mathcal{G}}_{\tau}$ (say it corresponds to some edge probability in $\hat{\mathcal{G}}_{\tau}$ without loss of generality) induces a thresholded stochastic feedback graph with strongly observable support $G = (V, E)$ and independence number $\alpha$, we have that $\hat{\mathcal{G}}_{\tau'}$ is strongly observable too; moreover, all the edges $e \in E$ are such that $|p_e - \hat{p}_e^{\tau}| \leq p_e/2$ by Definition 1 (part 2); the same holds for $\tau'$: $|p_e - \hat{p}_e^{\tau'}| \leq p_e/2$. Consider graph $G$ with edge probabilities $\hat{p}_e^{\tau'}$, respectively $p_e$ and $\hat{p}_e^{\tau}$ and let $\varepsilon_1$, respectively $\varepsilon_2$ and $\varepsilon_3$, be their smallest probability (restricting on the edges of $G$). We have that:

$$\min_{\varepsilon \in (0,1]} \left\{ \frac{\alpha((\hat{\mathcal{G}}_{\tau'})_\varepsilon)}{\varepsilon} \; : \; \mathrm{supp}((\hat{\mathcal{G}}_{\tau'})_\varepsilon) \text{ strongly observable} \right\} \leq \frac{\alpha}{\varepsilon_1} \leq 2\frac{\alpha}{\varepsilon_2} \leq 4\frac{\alpha}{\varepsilon_3}$$

$$= 4 \min_{\varepsilon \in (0,1]} \left\{ \frac{\alpha((\hat{\mathcal{G}}_{\tau})_\varepsilon)}{\varepsilon} \; : \; \mathrm{supp}((\hat{\mathcal{G}}_{\tau})_\varepsilon) \text{ strongly observable} \right\} ,$$

where the first inequality follows from suboptimality of graph $G$ with threshold $\varepsilon_1$ for $\hat{\mathcal{G}}_{\tau'}$, the second and the third inequality by the conditions on $p_e$, $\hat{p}_e^{\tau'}$ and $p_e^{\tau}$, and the last equality by definition of $G$ and $\alpha$. If we now substitute this inequality in the definition of $\Phi$, we obtain that $2\Phi(\hat{\mathcal{G}}_{\tau}, T) \geq \Phi(\hat{\mathcal{G}}_{\tau'}, T)$. We can reason in the same exact way considering the (weakly) observable case and obtain $\sqrt[3]{4}\Phi(\hat{\mathcal{G}}_{\tau}, T) \geq \Phi(\hat{\mathcal{G}}_{\tau'}, T)$. Putting the two results together we conclude the proof of point (i).

We move our attention to the second part of the lemma. Because of Theorem 2 together with the lower bound on $\tau$, it holds that $\hat{\mathcal{G}}_{\tau}$ is an $\varepsilon^*/2$-good approximation of $\mathcal{G}$. This implies that all the edges in $\mathrm{supp}(\mathcal{G}_{\varepsilon^*})$ are contained in the support of $\hat{\mathcal{G}}_{\tau}$ and that they are well approximated, as in parts 1 and 2 of Definition 1. We have two cases, according to the topology of the support corresponding to the threshold $\varepsilon^*$ which guarantees the optimal regret for $\mathcal{G}$. First, consider the case that $\varepsilon^*$ corresponds to a strongly observable structure in $\mathrm{supp}(\mathcal{G}_{\varepsilon^*})$ with independence number $\alpha^*$; we have that

$$\min_{\varepsilon \in (0,1]} \left\{ \frac{\alpha((\hat{\mathcal{G}}_{\tau})_\varepsilon)}{\varepsilon} \; : \; \mathrm{supp}((\hat{\mathcal{G}}_{\tau})_\varepsilon) \text{ strongly observable} \right\} \leq 2\frac{\alpha^*}{\varepsilon^*}$$

$$= 2 \min_{\varepsilon \in (0,1]} \left\{ \frac{\alpha(\mathcal{G}_\varepsilon)}{\varepsilon} \; : \; \mathrm{supp}(\mathcal{G}_\varepsilon) \text{ strongly observable} \right\} ,$$

where in the first inequality we used the suboptimality of threshold $\varepsilon^*/2$ for $\hat{\mathcal{G}}_{\tau}$ and the fact that the independence number of $\alpha((\hat{\mathcal{G}}_{\tau})_{\varepsilon^*})$ is at most $\alpha^*$ (and the strong observability is

maintained). Then, we have that

$$\Phi(\hat{\mathcal{G}}_\tau, T) \le 4C_s \sqrt{2\frac{\alpha^*}{\varepsilon^*}T}\big(\ln(KT)\big)^{3/2} = \sqrt{2}\Phi(\mathcal{G}, T) \ ,$$

where the inequality follows naturally from the (possible) suboptimality of the choice of the strongly observable regime and the threshold $\varepsilon^*/2$ for $\hat{\mathcal{G}}_\tau$. We can argue similarly for the case in which the optimal $\varepsilon^*$ corresponds to the weakly observable regime in $\mathcal{G}$. In this case, for the same arguments as per the strongly observable regime, we have that

$$\min_{\varepsilon \in (0,1]} \left\{ \frac{\delta((\hat{\mathcal{G}}_\tau)_\varepsilon)}{\varepsilon} \ : \ \mathrm{supp}((\hat{\mathcal{G}}_\tau)_\varepsilon) \text{ observable} \right\} \le 2\frac{\delta^*}{\varepsilon^*}$$

$$= 2 \min_{\varepsilon \in (0,1]} \left\{ \frac{\delta(\mathcal{G}_\varepsilon)}{\varepsilon} \ : \ \mathrm{supp}(\mathcal{G}_\varepsilon) \text{ observable} \right\} \ .$$

Finally, similarly to the strongly observable case, it holds that

$$\Phi(\hat{\mathcal{G}}_\tau, T) \le 4C_w \left( 2\frac{\delta^*}{\varepsilon^*}\big(\ln(KT)\big)^2 \right)^{1/3} T^{2/3} = \sqrt[3]{2}\Phi(\mathcal{G}, T) \le \sqrt{2}\Phi(\mathcal{G}, T) \ .$$

This concludes the proof. $\qquad\square$

**Lemma 4.** *Consider a run of* EDGECATCHER *(Algorithm 3). Assume that the invocation of* ROUNDROBIN *returns a stochastic feedback graph $\hat{\mathcal{G}}$ that is an $\hat{\varepsilon}$-good approximation of $\mathcal{G}$ satisfying $\Phi(\hat{\mathcal{G}}, T - \hat{\tau}K) \le \hat{\tau}K$, where $\hat{\tau}$ is the index of the last iteration of the outer for loop in Algorithm 1. Then, the regret experienced by the invocation of* BLOCKREDUCTION *is at most $\Phi(\hat{\mathcal{G}}, T - \hat{\tau}K)$.*

*Proof.* Denote with $R_{T'}^{\mathrm{BR}}$ the worst-case regret experienced by BLOCKREDUCTION in the final $T' = T - \hat{\tau}K$ time steps, under the assumption on $\hat{\mathcal{G}}$ in the statement, and let $\hat{\varepsilon}^*$ be the best threshold as in Algorithm 3. We have two cases, according to $\hat{\varepsilon}^*$ referring to strongly or (weakly) observable graphs. If $\hat{\varepsilon}^* = \hat{\varepsilon}_s^*$, then, by the part of Corollary 1 relative to strongly observable graphs, we have that

$$R_{T'}^{\mathrm{BR}} \le 4C_s \sqrt{\frac{\hat{\alpha}^*}{\hat{\varepsilon}_s^*}T'}\big(\ln(KT')\big)^{3/2} = \Phi(\hat{\mathcal{G}}_{\hat{\varepsilon}^*}, T') \ .$$

If $\hat{\varepsilon}^* = \hat{\varepsilon}_w^*$, then we can apply the part of Corollary 1 relative to (weakly) observable graphs and obtain that

$$R_{T'}^{\mathrm{BR}} \le 4C_w \left( \frac{\hat{\delta}^*}{\hat{\varepsilon}_w^*}\big(\ln(KT')\big)^2 \right)^{1/3} (T')^{2/3} = \Phi(\hat{\mathcal{G}}_{\hat{\varepsilon}^*}, T') \ .$$

$\qquad\square$

At this point, we have all the essential ingredients to prove the regret bound of EDGECATCHER as stated in Theorem 4. We rewrite the statement of Theorem 4 for convenience.

**Theorem 4.** *Consider the problem of online learning with stochastic feedback graph $\mathcal{G}$ on $T$ time steps. If $\mathrm{supp}(\mathcal{G}_{\varepsilon(K,T)})$ is observable for $\varepsilon(K,T) = CK^3(\ln(KT))^2/T$ for a given constant $C > 0$, then there exists an algorithm whose regret $R_T$ satisfies (ignoring polylog factors in $K$ and $T$) $R_T \le \min\left\{ \sqrt{(\alpha^*/\varepsilon_s^*)T}, \big(\delta^*/\varepsilon_w^*\big)^{1/3}T^{2/3} \right\}$.*

*Proof of Theorem 4.* We condition the analysis on the clean event $\mathcal{E}$ that ROUNDROBIN does not fail. Let $\tilde{\varepsilon}$ be the largest $\varepsilon$ such that $\mathrm{supp}(\mathcal{G}_\varepsilon)$ is observable, and $\tilde{\tau}$ be the smallest (random) integer such that $\mathrm{supp}(\hat{\mathcal{G}}_{\tilde{\tau}})$ is observable for $\hat{\mathcal{G}}_{\tilde{\tau}}$ in ROUNDROBIN. We have some immediate bound on these quantities. First, $\tilde{\varepsilon} \ge \varepsilon(K,T)$, by the assumption on $\mathrm{supp}(\mathcal{G}_{\varepsilon(K,T)})$ being observable. Second, $\tilde{\tau} \le \frac{120}{\tilde{\varepsilon}}\ln(KT)$; this is due to the fact that, after $\tau = \lceil \frac{120}{\tilde{\varepsilon}}\ln(KT) \rceil$ time steps, the estimated graph $\hat{\mathcal{G}}_\tau$ is an $\tilde{\varepsilon}/2$-good approximation of $\mathcal{G}$ and thus contains all

the edges in $\mathrm{supp}(\mathcal{G}_{\tilde{\varepsilon}})$ by Definition 1 (part 1) with $\varepsilon = \tilde{\varepsilon}/2$, and because of the conditioning on $\mathcal{E}$. All in all, we can summarize these observations by noticing that

$$\frac{T}{2K} \geq 120\frac{\ln(KT)}{\varepsilon(K,T)} \geq 120\frac{\ln(KT)}{\tilde{\varepsilon}} \geq \tilde{\tau} \ ,$$

where the first inequality is true as long as $\varepsilon(K,T) \geq 240K\ln(KT)/T$. Using point $(i)$ of Lemma 3 and the inequality we just showed, we observe that

$$\Phi(\hat{\mathcal{G}}_{\lfloor\frac{T}{2K}\rfloor},T) \leq 2\Phi(\hat{\mathcal{G}}_{\tilde{\tau}},T) \leq 8C_w\Big(2\frac{KT^2}{\tilde{\varepsilon}}\ln(KT)^2\Big)^{1/3} \leq 8C_w\Big(2\frac{KT^2}{\varepsilon(K,T)}\ln(KT)^2\Big)^{1/3} \leq \frac{T}{2} \ ,$$

as long as $\varepsilon(K,T) \geq 2\cdot 16^3 C_w^3 K(\ln(KT))^2/T$. Note that in the previous chain of inequalities we considered the (possibly suboptimal) choice of the (weakly) observable structure of the graph with threshold $\tilde{\varepsilon}$ and upper bound on $\delta$ given by $K$. The inequality we just showed implies that the stopping criterion in ROUNDROBIN is triggered and thus we can apply Lemma 4.

Now, let $\tau^*$ be the smallest $\tau$ such that $\Phi(\mathcal{G},T) = \Phi(\mathcal{G}_{\varepsilon^*},T) \leq \tau K$, being $\varepsilon^*$ the optimal threshold for $\mathcal{G}$. In this second step, we want to show that $\hat{\tau}$ is not too far away from $\tau^*$ for the interesting values of $\tau^*$; namely, that $\hat{\tau} \leq 4\tau^*$ as long as $\Phi(\mathcal{G},T)$ is not $\tilde{\Omega}(T)$.

First, consider the case that $\Phi(\mathcal{G},T)$ refers to the strongly observable regime in $\Phi(\mathcal{G}_{\varepsilon^*},T)$. By minimality of $\tau^*$, we have the following:

$$\tau^*K \geq \Phi(\mathcal{G},T) = 4C_s\sqrt{\frac{\alpha^*}{\varepsilon^*}}T\big(\ln(KT)\big)^{3/2} \geq \frac{1}{2}\tau^*K \ . \tag{14}$$

We now set the constant appearing in the definition of $\varepsilon(K,T)$ from the statement to be $C = 2\cdot 16^3 C_w^3$. With this choice, the previously stated requirements for $\varepsilon(K,T)$ are satisfied, while at the same time it holds that $\Phi(\mathcal{G},T) \leq C_s^2 T(\ln(KT))^2/(15K)$; this is immediate to verify by arguing that $\Phi(\mathcal{G},T)$ is at most the regret incurred by using the (possibly suboptimal, weakly) observable structure of $\mathcal{G}$ truncated at $\varepsilon(K,T)$. Then, from the second inequality of (14), it follows that $\tau^* \leq 2C_s^2 T(\ln(KT))^2/(15K^2)$. We can rewrite the first inequality of (14) as follows:

$$\varepsilon^* \geq 16C_s^2\frac{\alpha^*}{(K\tau^*)^2}T\big(\ln(KT)\big)^3 \geq 120\frac{\ln(KT)}{\tau^*} \ .$$

Consider now to what happens at the $\overline{\tau} = \lceil 120\ln(KT)/\varepsilon^*\rceil \leq 4\tau^*$ iteration of ROUNDROBIN. The estimated graph $\hat{\mathcal{G}}_{\overline{\tau}}$ in that iteration is an $\varepsilon^*/2$-good approximation of $\mathcal{G}$, thus it contains all the edges of $\mathcal{G}$, with the probabilities correctly estimated up to a constant multiplicative factor, as detailed in Definition 1 (part 2). Thus,

$$\Phi(\hat{\mathcal{G}}_{4\tau^*},T) \leq 2\Phi(\hat{\mathcal{G}}_{\overline{\tau}},T) \leq 2\sqrt{2}\Phi(\mathcal{G},T) \leq 4\tau^*K,$$

which implies that the stopping time $\hat{\tau}$ is attained before $4\tau^*$. Note that the first inequality is due to point $(i)$ of Lemma 3, whereas the second inequality follows from point $(ii)$ of Lemma 3.

Similarly, we consider the case that $\Phi(\mathcal{G},T)$ refers to the weakly observable regime in $\Phi(\mathcal{G}_{\varepsilon^*},T)$. By minimality of $\tau^*$, we have the following:

$$\tau^*K \geq \Phi(\mathcal{G},T) = 4C_w\left(\frac{\delta^*}{\varepsilon^*}\big(\ln(KT)\big)^2\right)^{1/3}T^{2/3} \geq \frac{1}{2}\tau^*K \ . \tag{15}$$

By the choice of $\varepsilon(K,T)$, we have that $\Phi(\mathcal{G},T) \leq T\sqrt{2C_w^3\ln(KT)/(15K)}$. Then, from the second inequality of (15), it follows that $\tau^* \leq T\sqrt{8C_w^3\ln(KT)/(15K^3)}$. Consider now the first inequality, we can rewrite it to obtain:

$$\varepsilon^* \geq 64C_w^3\frac{\delta^*}{(K\tau^*)^3}\big(T\ln(KT)\big)^2 \geq 120\frac{\ln(KT)}{\tau^*} \ .$$

We can now use the same argument as in the strongly observable case and conclude that $\hat{\tau} \leq 4\tau^*$.

At this point, we are ready to show that our algorithm EDGECATCHER exhibits the desired regret bounds. We are conditioning on the good event $\mathcal{E}$; this happens with probability at least $1 - \frac{1}{T}$, so we just analyze this case, as the complementary of $\mathcal{E}$ yields at most an extra additive 1, in expectation, to the regret bound.

Recall that $R_T$ is the worst-case regret; thus,

$$R_T \leq \hat{\tau}K + \Phi(\hat{\mathcal{G}}, T - \hat{\tau}K) \leq 2\hat{\tau}K \leq 8\tau^*K \leq 16\Phi(\mathcal{G}, T) \ ,$$

where in the first inequality we used the decomposition in regret before and after the commitment and the bound on Lemma 4 (which is applicable given the conditioning on $\mathcal{E}$ and thus all the $\mathcal{G}_\tau$ are $\varepsilon_\tau$-good approximations of $\mathcal{G}$), in the second one the definition of $\hat{\tau}$, in the third one the fact that $\hat{\tau} \leq 4\tau^*$, and in the last the definition of $\tau^*$ as minimal $\tau$ such that $\Phi(\mathcal{G}, T) \leq \tau K$. □

## C  Proofs of Lower Bounds

The main idea in the lower bounds is that the adversary sets all edge probabilities equal to $\varepsilon \in (0, 1]$ in order to define a stochastic feedback graph $\mathcal{G}$ with a specific support $G$ that satisfies adequate properties. This requires the attribution of additional power to the adversary because we allow it to choose the edge probabilities; nevertheless, this is fine from a worst-case perspective because it corresponds to choosing a particularly difficult instance among those that have certain characteristics. Doing so makes the edge between each (ordered) pair of nodes either realize independently at each round $t$ with probability equal to $\varepsilon$, or never realize. Moreover, there exists a vertex that is at least marginally better than the other ones with respect to the expected loss. The learner only obtains information about the loss of the optimal node whenever it plays a node that is adjacent to it in $G = \mathrm{supp}(\mathcal{G})$ and the edge between the played node and the optimal node is realized. Since that edge is realized only with probability $\varepsilon$, it is significantly harder for the the learner to detect the optimal node, which allows the adversary to increase the size of the gaps between the optimal node and the suboptimal ones. More specifically, while in the deterministic setting playing once action $a$ is enough to observe the loss incurred by a neighbouring action $a'$, the learner will now need $1/\varepsilon$ time steps, in expectation, to observe the loss of $a'$ if the edge $(a, a')$ only realizes with probability $\varepsilon$. Further notice that, in the setting considered within the proofs of our lower bounds, the learner may even know the true distribution $\mathcal{G}$ and observe the realization of the entire feedback graph $G_t$ at the end of each round $t$.

We start with a lower bound for the strongly observable case considering stochastic feedback graphs $\mathcal{G}$ with $\alpha(\mathcal{G}) > 1$. The following result can be recovered by adapting the proof of Alon et al. [2017, Theorem 5] that holds for any graph of interest (directed or undirected).

**Theorem 7.** *Pick any directed or undirected graph $G = (V, E)$ with $\alpha(G) > 1$ and any $\varepsilon \in (0, 1]$. There exists a stochastic feedback graph $\mathcal{G}$ with $\mathrm{supp}(\mathcal{G}) = G$ and such that, for all $T \geq 0.0064\alpha(\mathcal{G}_\varepsilon)^3/\varepsilon$ and for any possibly randomized algorithm $\mathcal{A}$, there exists a sequence $\ell_1, \ldots, \ell_T$ of loss functions on which the expected regret of $\mathcal{A}$ with respect to the stochastic generation of $G_1, \ldots, G_T \sim \mathcal{G}$ is at least $0.017\sqrt{\alpha(\mathcal{G}_\varepsilon)T/\varepsilon}$.*

*Proof.* The structure of this proof follows the same rationale of the lower bound by Alon et al. [2017, Theorem 5] with additional considerations due to the stochasticity of the feedback graph. To prove the lower bound we will use Yao's minimax principle [Yao, 1977], which shows that it is sufficient to provide a probabilistic strategy for the adversary on which the expected regret of any deterministic algorithm is lower bounded.

We can assume that $G$ has all self-loops. If $G$ is missing some self-loops, we may add them for the sake of the lower bound: this only makes the problem easier for the learner. Also note that the addition of self-loops does not change the independence number of $G$. Now let $\mathcal{G}$ be such that $p(i, j) \in \{0, \varepsilon\}$ and $p(i, j) = \varepsilon$ if and only if $(i, j) \in E$, for all $i, j \in V$. Note that $\alpha(G) = \alpha(\mathcal{G})$ and $\mathcal{G} = \mathcal{G}_\varepsilon$. We also remark that the following lower bound for such a $\mathcal{G}$ will be a lower bound for the instance having a stochastic feedback graph obtained from the starting graph, without the addition of self-loops, by setting the realization probability of all its edges to $\varepsilon$. Without loss of generality, we order the nodes depending on an (arbitrary)

independent set of $G$ of size $\alpha(G)$ so that $1, 2, \ldots, \alpha(G)$ are the nodes belonging to said independent set, and $\alpha(G) + 1, \ldots, |V|$ correspond to all the other nodes in $G$.

We will use the following distribution of losses. We sample $Z$ from some (later defined) distribution $Q$ over the independent set chosen above. Conditioned on $Z = i$, the loss $\ell_t(j)$ is sampled from an independent Bernoulli distribution with mean $\frac{1}{2}$ if $j \neq i$ and $j \leq \alpha(G)$, it is sampled from an independent Bernoulli with mean $\frac{1}{2} - \beta$ if $j = i$ for some $\beta \in [0, \frac{1}{4}]$, and it is set to 1 otherwise.

We denote by $T_i$ the number of times node $i$ was chosen by the algorithm after $T$ rounds and denote by $T_{\text{bad}} = \sum_{i > \alpha(G)} T_i$ the number of times the algorithm chooses an action not in the independent set. We use $\mathbb{E}_i[\cdot] = \mathbb{E}[\cdot \,|\, Z = i]$ and $\mathbb{P}_i(\cdot) = \mathbb{P}(\cdot \,|\, Z = i)$ to denote the expectation and probability over $(G_1, \ell_1), \ldots, (G_T, \ell_T)$ conditioned on $Z = i$, respectively. We denote by $\ell_t(I_t)$ the loss of algorithm $A$ playing $I_t$ in round $t$. We emphasize that the complete loss sequence and the (partial) loss sequence observed by the learner may differ depending not only on the actions of the learner but also on the realization of the edges in the feedback graph. This last observation will be used to lower bound the regret of the learner also in terms of $\varepsilon$, the probability of an edge realization.

We set $Q(i) = \frac{1}{\alpha(G)}$ if $i$ is in the independent set and $Q(i) = 0$ otherwise. Following Alon et al. [2017, Equation (8)] we have, for any deterministic algorithm $A$, that

$$\max_{k \in V} \mathbb{E}\left[ \sum_{t=1}^{T} \big( \ell_t(I_t) - \ell_t(k) \big) \right] \geq \beta \left( T - \frac{1}{\alpha(G)} \sum_{i \leq \alpha(G)} \mathbb{E}_i[T_i] \right). \tag{16}$$

We now consider an auxiliary distribution $\mathbb{P}_0$, also over $(G_1, \ell_1), \ldots, (G_T, \ell_T)$, which is equivalent to the distribution $\mathbb{P}_i$ that we specified above, but with $\beta = 0$ for all nodes. We denote by $\mathbb{E}_0$ the corresponding expectation. We also denote by $\lambda_t$ the feedback set at time $t$, composed by the realization $G_t$ of the feedback graph together with the set of losses observed by the learner in round $t$, and by $\lambda^t = (\lambda_1, \ldots, \lambda_t)$ the tuple of all feedback sets up to and including round $t$. Since the algorithm is deterministic, its action $I_t$ in round $t$ is fully determined by $\lambda^{t-1}$. Therefore, $\mathbb{E}_i[T_i \,|\, \lambda^T] = \mathbb{E}_0[T_i \,|\, \lambda^T]$. When $\lambda^{t-1}$ is understood from the context, let $\mathbb{P}_{j,t} = \mathbb{P}_j(\cdot \,|\, \lambda^{t-1})$ be the conditional probability measure of feedback sets $\lambda_t$ at time $t$. We have that

$$\mathbb{E}_i[T_i] - \mathbb{E}_0[T_i] = \sum_{\lambda^T} \mathbb{P}_i(\lambda^T) \mathbb{E}_i[T_i \,|\, \lambda^T] - \sum_{\lambda^T} \mathbb{P}_0(\lambda^T) \mathbb{E}_0[T_i \,|\, \lambda^T]$$

$$= \sum_{\lambda^T} \mathbb{P}_i(\lambda^T) \mathbb{E}_i[T_i \,|\, \lambda^T] - \sum_{\lambda^T} \mathbb{P}_0(\lambda^T) \mathbb{E}_i[T_i \,|\, \lambda^T]$$

$$\leq T \sum_{\lambda^T \,:\, \mathbb{P}_i(\lambda^T) > \mathbb{P}_0(\lambda^T)} \big( \mathbb{P}_i(\lambda^T) - \mathbb{P}_0(\lambda^T) \big) .$$

By using Pinsker's inequality and the chain rule for the relative entropy, we can further observe that

$$\sum_{\lambda^T \,:\, \mathbb{P}_i(\lambda^T) > \mathbb{P}_0(\lambda^T)} \big( \mathbb{P}_i(\lambda^T) - \mathbb{P}_0(\lambda^T) \big) \leq \sqrt{\tfrac{1}{2} D_{\text{KL}}(\mathbb{P}_0 \,\|\, \mathbb{P}_i)}$$

$$= \sqrt{\frac{1}{2} \sum_{t=1}^{T} \sum_{\lambda^{t-1}} \mathbb{P}_0(\lambda^{t-1}) D_{\text{KL}}(\mathbb{P}_{0,t} \,\|\, \mathbb{P}_{i,t})} ,$$

which, combined with the previous inequality, allows us to affirm that

$$\mathbb{E}_i[T_i] - \mathbb{E}_0[T_i] \leq \sqrt{\frac{1}{2} \sum_{t=1}^{T} \sum_{\lambda^{t-1}} \mathbb{P}_0(\lambda^{t-1}) D_{\text{KL}}(\mathbb{P}_{0,t} \,\|\, \mathbb{P}_{i,t})} . \tag{17}$$

At this point, observe that $\text{supp}(\mathcal{G}) = G = (V, E)$. Fix any $\lambda^{t-1}$ and consider $D_{\text{KL}}(\mathbb{P}_{0,t} \,\|\, \mathbb{P}_{i,t})$ where, we recall, $\mathbb{P}_{0,t}(\lambda_t) = \mathbb{P}_0(\lambda_t \,|\, \lambda^{t-1})$ and $\mathbb{P}_{i,t}(\lambda_t) = \mathbb{P}_i(\lambda_t \,|\, \lambda^{t-1})$. Recall that $\lambda^{t-1}$ fully

determines the node $I_t$ picked by the algorithm in round $t$. If $(I_t, i) \notin E$, then $\mathbb{P}_{0,t}$ and $\mathbb{P}_{i,t}$ have the same distribution and the relative entropy term is 0. If $(I_t, i) \in E$, then the loss of node $i$ in $\lambda_t$ follows a Bernoulli distribution with mean $\frac{1}{2}$ under $\mathbb{P}_0$ and follows a Bernoulli distribution with mean $\frac{1}{2} - \beta$ under $\mathbb{P}_i$. Denote by $\mathcal{E}_t$ the event that edge $(I_t, i)$ is realized in $G_t$. Note that $\mathbb{P}_0(\mathcal{E}_t) = \mathbb{P}_i(\mathcal{E}_t) = \varepsilon$. Using the log-sum inequality and the fact that the relative entropy between the two aforementioned Bernoulli distributions is given by $\frac{1}{2} \ln\left(\frac{1}{1-4\beta^2}\right)$, we can see that

$$
\begin{aligned}
D_{\mathrm{KL}}(\mathbb{P}_{0,t} \,\|\, \mathbb{P}_{i,t}) &= D_{\mathrm{KL}}\Big(\varepsilon \mathbb{P}_{0,t}(\cdot \,|\, \mathcal{E}_t) + (1-\varepsilon)\mathbb{P}_{0,t}(\cdot \,|\, \overline{\mathcal{E}_t}) \,\Big\|\, \varepsilon \mathbb{P}_{i,t}(\cdot \,|\, \mathcal{E}_t) + (1-\varepsilon)\mathbb{P}_{i,t}(\cdot \,|\, \overline{\mathcal{E}_t})\Big) \\
&= D_{\mathrm{KL}}\Big(\varepsilon \mathbb{P}_{0,t}(\cdot \,|\, \mathcal{E}_t) + (1-\varepsilon)\mathbb{P}_{0,t}(\cdot \,|\, \overline{\mathcal{E}_t}) \,\Big\|\, \varepsilon \mathbb{P}_{i,t}(\cdot \,|\, \mathcal{E}_t) + (1-\varepsilon)\mathbb{P}_{0,t}(\cdot \,|\, \overline{\mathcal{E}_t})\Big) \\
&\leq \varepsilon D_{\mathrm{KL}}\big(\mathbb{P}_{0,t}(\cdot \,|\, \mathcal{E}_t) \,\|\, \mathbb{P}_{i,t}(\cdot \,|\, \mathcal{E}_t)\big) + (1-\varepsilon) D_{\mathrm{KL}}\big(\mathbb{P}_{0,t}(\cdot \,|\, \overline{\mathcal{E}_t}) \,\|\, \mathbb{P}_{0,t}(\cdot \,|\, \overline{\mathcal{E}_t})\big) \\
&= \varepsilon D_{\mathrm{KL}}\big(\mathbb{P}_{0,t}(\cdot \,|\, \mathcal{E}_t) \,\|\, \mathbb{P}_{i,t}(\cdot \,|\, \mathcal{E}_t)\big) \\
&= -\frac{\varepsilon}{2} \ln\big(1 - 4\beta^2\big) \leq 8 \ln(4/3)\beta^2 \varepsilon \ .
\end{aligned}
$$
(18)

With this inequality, we may upper bound the sum in the right-hand side of (17) by considering, for each $t$, only the tuples $\lambda^{t-1}$ for which $i \in N_G^{\mathrm{out}}(I_t)$ holds. Indeed, the KL divergence for any other possible $\lambda^{t-1}$ is equal to 0 because the edge $(I_t, i)$ never realizes (it is not in the support of $\mathcal{G}$, hence $p(I_t, i) = 0$). As a consequence,

$$
\begin{aligned}
\sum_{t=1}^{T} \sum_{\lambda^{t-1}} \mathbb{P}_0(\lambda^{t-1}) D_{\mathrm{KL}}(\mathbb{P}_{0,t} \,\|\, \mathbb{P}_{i,t}) &\leq \sum_{t=1}^{T} \mathbb{P}_0(i \in N_G^{\mathrm{out}}(I_t)) 8 \ln(4/3)\beta^2 \varepsilon \\
&= 8 \ln(4/3)\beta^2 \varepsilon \mathbb{E}_0[|\{t : i \in N_G^{\mathrm{out}}(I_t)\}|] \\
&\leq 8 \ln(4/3)\beta^2 \varepsilon \mathbb{E}_0[T_i + T_{\mathrm{bad}}] \ .
\end{aligned}
$$
(19)

We may claim that $\mathbb{E}_0[T_{\mathrm{bad}}] \leq 0.04\sqrt{\frac{\alpha(G)}{\varepsilon}T}$, because otherwise the expected regret under $\mathbb{P}_0$ would have been at least

$$
\begin{aligned}
\max_{k \in V} \mathbb{E}_0\left[\sum_{t=1}^{T}(\ell_t(I_t) - \ell_t(k))\right] &= \mathbb{E}_0\left[T_{\mathrm{bad}} + \frac{1}{2}\sum_{j \leq \alpha(G)} T_j\right] - \frac{1}{2}T \\
&= \mathbb{E}_0\left[\frac{1}{2}T_{\mathrm{bad}} + \frac{1}{2}\left(T_{\mathrm{bad}} + \sum_{j \leq \alpha(G)} T_j\right)\right] - \frac{1}{2}T \\
&= \mathbb{E}_0\left[\frac{1}{2}T_{\mathrm{bad}}\right] \\
&> 0.02\sqrt{\frac{\alpha(G)}{\varepsilon}T} \ .
\end{aligned}
$$

Combining Equations (17) and (19), and using that $\mathbb{E}_0[T_{\mathrm{bad}}] \leq 0.04\sqrt{\frac{\alpha(G)}{\varepsilon}T}$, we find that

$$
\mathbb{E}_i[T_i] - \mathbb{E}_0[T_i] \leq 2T\beta\sqrt{\varepsilon \ln(4/3)\mathbb{E}_0\left[T_i + 0.04\sqrt{\frac{\alpha(G)}{\varepsilon}T}\right]} \ .
$$

This implies that the regret can be further lower bounded, continuing from (16), by

$$\beta \left( T - \frac{1}{\alpha(G)} \sum_{i=1}^{\alpha(G)} \mathbb{E}_0[T_i] - \frac{1}{\alpha(G)} \sum_{i=1}^{\alpha(G)} 2T\beta \sqrt{\varepsilon \ln(4/3) \mathbb{E}_0 \left[ T_i + 0.04 \sqrt{\frac{\alpha(G)}{\varepsilon}} T \right]} \right)$$

$$\geq \beta \left( T - \frac{1}{\alpha(G)} \sum_{i=1}^{\alpha(G)} \mathbb{E}_0[T_i] - 2T\beta \sqrt{\varepsilon \ln(4/3) \mathbb{E}_0 \left[ \frac{1}{\alpha(G)} \sum_{i=1}^{\alpha(G)} T_i + 0.04 \sqrt{\frac{\alpha(G)}{\varepsilon}} T \right]} \right)$$

$$\geq \beta T \left( 1 - \frac{1}{\alpha(G)} - 2\beta \sqrt{\varepsilon \ln(4/3) \left( \frac{T}{\alpha(G)} + 0.04 \sqrt{\frac{\alpha(G)}{\varepsilon}} T \right)} \right) \ ,$$

where the first inequality is Jensen's inequality for concave functions and the second inequality is due to the fact that $\sum_{i=1}^{\alpha(G)} \mathbb{E}_0[T_i] \leq T$ by definition of $T_i$. Since we assumed that $T \geq 0.0064 \alpha(G)^3/\varepsilon$, we have that $0.04\sqrt{\frac{\alpha(G)}{\varepsilon}}T \leq \frac{T}{2\alpha(G)}$ and thus

$$\max_{k \in V} \mathbb{E} \left[ \sum_{t=1}^{T} (\ell_t(I_t) - \ell_t(k)) \right] \geq \beta T \left( 1 - \frac{1}{\alpha(G)} - 2\beta \sqrt{\frac{3}{2} \ln(4/3) \frac{\varepsilon T}{\alpha(G)}} \right)$$

$$\geq \beta T \left( \frac{1}{2} - 2\beta \sqrt{\frac{3}{2} \ln(4/3) \frac{\varepsilon T}{\alpha(G)}} \right) \ ,$$

where in the second inequality we used the assumption that $\alpha(G) \geq 2$. By setting $\beta = \frac{1}{33} \sqrt{\frac{\alpha(G)}{2 \ln(4/3) \varepsilon T}} \in (0, \frac{1}{4}]$, we may complete the proof as

$$\max_{k \in V} \mathbb{E} \left[ \sum_{t=1}^{T} (\ell_t(I_t) - \ell_t(k)) \right] \geq \frac{1}{33} \left( \frac{1}{2} - \frac{\sqrt{3}}{33} \right) \sqrt{\frac{\alpha(G) T}{2 \ln(4/3) \varepsilon}} \geq 0.017 \sqrt{\frac{\alpha(G)}{\varepsilon} T} \ .$$

$\square$

Given that this lower bound leaves the case $\alpha(\mathcal{G}) = 1$ uncovered, we provide an additional lower bound that considers any feedback graph. This new bound is tight up to logarithmic factors, for instance, in all cases where $\alpha(\mathcal{G})$ is constant.

**Theorem 8.** *Pick any directed or undirected graph $G = (V, E)$ with $|V| = K \geq 2$ and any $\varepsilon \in (0, 1]$. There exists a stochastic feedback graph $\mathcal{G}$ with $\mathrm{supp}(\mathcal{G}) = G$ and such that, for all $T \geq 1/(2\varepsilon)$ and for any possibly randomized algorithm $\mathcal{A}$, there exists a sequence $\ell_1, \ldots, \ell_T$ of loss functions on which the expected regret of $\mathcal{A}$ with respect to the stochastic generation of $G_1, \ldots, G_T \sim \mathcal{G}$ is at least $\frac{1}{32} \sqrt{2T/\varepsilon}$.*

*Proof.* Following a similar rationale as in the proof of Theorem 7, we can consider $G$ to be the complete graph (with all self-loops) because the problem for it is easier than that with any other graph. In fact, adding edges never makes the problem harder to solve. Moreover, we can define $\mathcal{G}$ by setting all edge probabilities to $\varepsilon$ so that $\mathcal{G}_\varepsilon = \mathcal{G}$ and $\mathrm{supp}(\mathcal{G}) = G$. We remark that the lower bound with such a $\mathcal{G}$ is also a lower bound for the instance obtained by considering the initial (possibly non-complete) graph and assigning realization probability $\varepsilon$ to all its edges. Applying Yao's minimax principle allows us to reduce our current aim to proving a lower bound for the expected regret of any deterministic algorithm against a randomized adversary.

We can then construct the sequence of loss functions by defining their distribution. Let $v \in V$ be an arbitrary vertex, say, $v = 1$. Pick $Z \in \{-1, +1\}$ uniformly at random and define $\beta = \frac{1}{4}(2\varepsilon T)^{-1/2} \in [0, \frac{1}{4}]$. Then, let the loss at any time $t$ be independently $\ell_t(i) \sim \mathsf{Bern}\left(\frac{1}{2}\right)$ for $i \neq 1$ while $\ell_t(1) \sim \mathsf{Bern}\left(\frac{1}{2} - \beta Z\right)$. Define $\mathbb{P}_1(\cdot) = \mathbb{P}(\cdot \mid Z = +1)$ and $\mathbb{P}_2(\cdot) = \mathbb{P}(\cdot \mid Z = -1)$, as well as $\mathbb{E}_1[\cdot] = \mathbb{E}[\cdot \mid Z = +1]$ and $\mathbb{E}_2[\cdot] = \mathbb{E}[\cdot \mid Z = -1]$. We also define $\mathbb{P}_0(\cdot)$ and $\mathbb{E}_0[\cdot]$, obtained in an analogous manner as the previous ones by setting $\beta = 0$.

At this point, let $T_1$ be the number of times $t$ that the algorithm selects vertex $I_t = 1$ after $T$ rounds. Following a similar computation as in Equations (17) and (19), we first denote by $\mathbb{P}_{j,t} = \mathbb{P}_j(\cdot \mid \lambda^{t-1})$ the conditional probability over feedback sets $\lambda_t$, and notice that

$$
\begin{aligned}
\mathbb{E}_1[T_1] - \mathbb{E}_2[T_1] &\leq T\sqrt{\frac{1}{2}\sum_{t=1}^{T}\sum_{\lambda^{t-1}}\mathbb{P}_2(\lambda^{t-1})D_{\mathrm{KL}}(\mathbb{P}_{2,t}\,\|\,\mathbb{P}_{1,t})} \\
&\leq T\sqrt{\varepsilon\beta T\ln\left(1 + \frac{4\beta}{1-2\beta}\right)} \\
&\leq 2\beta T\sqrt{2\varepsilon T} \ .
\end{aligned}
\tag{20}
$$

Conditioning on $Z = +1$, the algorithm incurs an expected instantaneous regret equal to $\beta$ whenever it picks any vertex $i \neq 1$. Otherwise, conditioning on $Z = -1$, the algorithm incurs the same expected instantaneous regret each time it selects vertex 1. The expected regret thus becomes

$$
\begin{aligned}
\max_{k\in V}\mathbb{E}\left[\sum_{t=1}^{T}(\ell_t(I_t) - \ell_t(k))\right] &\geq \frac{1}{2}\mathbb{E}_1[\beta(T - T_1)] + \frac{1}{2}\mathbb{E}_2[\beta T_1] \\
&\geq \frac{\beta}{2}T - \frac{\beta}{2}(\mathbb{E}_1[T_1] - \mathbb{E}_2[T_1]) \\
&\geq \beta T\left(\frac{1}{2} - \beta\sqrt{2\varepsilon T}\right) = \frac{1}{4}\beta T = \frac{1}{32}\sqrt{\frac{2T}{\varepsilon}} \ ,
\end{aligned}
$$

where the third inequality follows by Equation (20), and we also use our choice of $\beta$. $\qquad\square$

We can additionally prove further lower bounds for the weakly observable case. Here we also adapt the proof for the lower bound in the case of a deterministic feedback graph by having each edge realize only with probability $\varepsilon \in (0, 1]$ at each time step. We make the same considerations as in the previous lower bound for strongly observable graphs. In this case, however, we refer to Alon et al. [2015, Theorem 7]. As in the case of deterministic feedback graph, we need the following combinatorial lemma.

**Lemma 5** (Alon et al. [2015, Lemma 8]). *Let $G = (V, E)$ be a directed graph over $|V| = n$ vertices, and let $W \subseteq V$ be a set of vertices whose minimal dominating set is of size $k$. Then, there exists an independent set $U \subseteq W$ of size $|U| \geq \frac{1}{50}k/\ln n$, such that any vertex of $G$ dominates at most $\ln n$ vertices of $U$.*

We can then prove the desired lower bound which states what follows.

**Theorem 9.** *Pick any directed or undirected, weakly observable graph $G = (V, E)$ with $|V| = K$ and $\delta(G) \geq 100\ln K$, and any $\varepsilon \in (0, 1]$. There exists a stochastic feedback graph $\mathcal{G}$ with $\mathrm{supp}(\mathcal{G}) = G$ and such that, for all $T \geq 2K/(\varepsilon\ln K)$ and for any possibly randomized algorithm $\mathcal{A}$, there exists a sequence $\ell_1, \ldots, \ell_T$ of loss functions on which the expected regret of $\mathcal{A}$ with respect to the stochastic generation of $G_1, \ldots, G_T \sim \mathcal{G}$ is at least $\frac{1}{150}\left(\frac{\delta(\mathcal{G}_\varepsilon)}{\varepsilon\ln^2 K}\right)^{1/3}T^{2/3}$.*

*Proof.* The proof follows the steps of the lower bound from Alon et al. [2015, Theorem 7]. As in the previous lower bounds, we use Yao's minimax principle to infer that it suffices to design a probabilistic adversarial strategy that leads to a sufficiently large lower bound for the expected regret of any deterministic algorithm.

We consider any weakly observable $G = (V, E)$ having $|V| = K$ vertices and $\delta(G) \geq 100\ln K$. Since the adversary may choose edge probabilities, it can pick them all equal to $\varepsilon$ so that $\mathcal{G} = \mathcal{G}_\varepsilon$ and $\mathrm{supp}(\mathcal{G}) = G$. By Lemma 5 we know that $G$ contains an independent set $U$ of size $|U| = m \geq \delta(G)/(50\ln K)$ such that any $v \in V$ dominates no more than $\ln K$ vertices of $U$. We will denote actions in $U$ as "good" actions, whereas all the others will be denoted as "bad" actions. Given our assumption on $\delta(G)$, we observe that $m \geq 2$. A further observation we can make is that $N_G^{\mathrm{in}}(i) \subseteq V \setminus U$ for all $i \in U$ because $U$ is independent, meaning that we need to pick a bad action in order to be able to observe the loss of any good action.

As similarly done in the proof of Theorem 7, we sample $Z$ from our "target" set $U$ uniformly at random. This choice induces a distribution of the losses $\ell_t(i)$ for all $t$ and all $i$ independently. To be precise, given $\beta = m^{1/3}(32\varepsilon T \ln K)^{-1/3} \in [0, \frac{1}{4}]$, the loss is $\ell_t(i) \sim \mathsf{Bern}(\frac{1}{2} - \beta)$ if $i = Z$, while it is $\ell_t(i) \sim \mathsf{Bern}(\frac{1}{2})$ if $i \in U$, $i \neq Z$. The loss is deterministically set to $\ell_t(i) = 1$ for any other vertex $i \in V \setminus U$.

Taking up the same notation introduced in the proof of Theorem 7, we denote by $T_i$ the number of times action $i$ is played by the deterministic algorithm after $T$ rounds, while $T_{\text{bad}} = \sum_{i \in V \setminus U} T_i$. In particular, $I_t$ is the action chosen by the algorithm at time $t$. We also use $\mathbb{P}_i(\cdot) = \mathbb{P}(\cdot \mid Z = i)$ and $\mathbb{E}_i[\cdot] = \mathbb{E}[\cdot \mid Z = i]$ with a similar definition, including the auxiliary distribution $\mathbb{P}_0$ and the corresponding expectation $\mathbb{E}_0$ obtained by setting $\beta = 0$. Moreover, for each good action $i$ we introduce $X_i = \sum_{t=1}^{T} \mathbb{I}_{\{I_t \in N_G^{\text{in}}(i)\}}$ to denote the number of times the algorithm picks a bad action from $N_G^{\text{in}}(i)$.

Notice that we can restrict our reasoning to algorithms that have $T_{\text{bad}} \leq \beta T$ (otherwise reducing to this case by only introducing a factor 3 in the regret bound), as similarly argued in the proof of Alon et al. [2015, Theorem 7]. This implies that

$$\sum_{i \in U} X_i \leq T_{\text{bad}} \ln K \leq \beta T \ln K \tag{21}$$

since each $j \in V \setminus U$ dominates at most $\ln K$ vertices of $U$.

Recalling Equation (17), we are interested in bounding

$$\mathbb{E}_i[T_i] - \mathbb{E}_0[T_i] \leq T \sqrt{\frac{1}{2} \sum_{t=1}^{T} \sum_{\lambda^{t-1}} \mathbb{P}_0(\lambda^{t-1}) D_{\text{KL}}(\mathbb{P}_{0,t} \,\|\, \mathbb{P}_{i,t})} \;, \tag{22}$$

where $\mathbb{P}_{j,t} = \mathbb{P}_j(\cdot \mid \lambda^{t-1})$ is the conditional probability over feedback sets $\lambda_t$. The KL divergence in the above sum is $D_{\text{KL}}(\mathbb{P}_{0,t} \,\|\, \mathbb{P}_{i,t}) \leq 8 \ln(4/3)\beta^2 \varepsilon$, where we use a similar reasoning as in Equation (18). As a consequence,

$$\sum_{t=1}^{T} \sum_{\lambda^{t-1}} \mathbb{P}_0(\lambda^{t-1}) D_{\text{KL}}(\mathbb{P}_{0,t} \,\|\, \mathbb{P}_{i,t}) \leq \sum_{t=1}^{T} \mathbb{P}_0(I_t \in N_G^{\text{in}}(i)) 8 \ln(4/3)\beta^2 \varepsilon$$

$$\leq 4\beta^2 \varepsilon \mathbb{E}_0[|\{t : I_t \in N_G^{\text{in}}(i)\}|]$$

$$= 4\beta^2 \varepsilon \mathbb{E}_0[X_i] \;,$$

which together with Equation (22) allows us to show that

$$\mathbb{E}_i[T_i] - \mathbb{E}_0[T_i] \leq \beta T \sqrt{2\varepsilon \mathbb{E}_0[X_i]} \;. \tag{23}$$

Let us now consider the expected regret for the deterministic algorithm at hand. We know that it must be at least

$$\max_{k \in V} \mathbb{E}\left[\sum_{t=1}^{T}(\ell_t(I_t) - \ell_t(k))\right] \geq \frac{1}{m} \sum_{i \in U} \mathbb{E}_i[\beta(T - T_i)] = \beta T - \frac{\beta}{m} \sum_{i \in U} \mathbb{E}_i[T_i]$$

because the algorithm incurs at least $\beta$ regret each time it picks an action different from $Z$. By Equations (21) and (23), and using the concavity of the square root, the summation on the right-hand side is such that

$$\frac{1}{m} \sum_{i \in U} \mathbb{E}_i[T_i] \leq \beta T \sqrt{\frac{2\varepsilon}{m} \sum_{i \in U} \mathbb{E}_0[X_i]} + \frac{1}{m} \mathbb{E}_0\left[\sum_{i \in U} T_i\right]$$

$$\leq T \sqrt{\frac{2\beta^3 \varepsilon}{m} T \ln K} + \frac{T}{m}$$

$$= \frac{1}{4}T + \frac{1}{m}T \leq \frac{3}{4}T \;, \tag{24}$$

where the equality follows by our choice of $\beta$, whereas the last inequality holds because $m \geq 2$. Hence, the expected regret is

$$\max_{k \in V} \mathbb{E}\left[\sum_{t=1}^{T}(\ell_t(I_t) - \ell_t(k))\right] \geq \frac{\beta}{4}T = \frac{1}{4}\left(\frac{m}{32\varepsilon \ln K}\right)^{1/3}T^{2/3} \geq \frac{1}{50}\left(\frac{\delta(G)}{\varepsilon \ln^2 K}\right)^{1/3}T^{2/3} .$$

$\square$

An additional theorem is required in order to cover the case $\delta(\mathcal{G}) < 100 \ln K$. In the same way as in Alon et al. [2015], we follow a simple reasoning with generic weakly observable graphs. The following lower bound holds for weakly observable graphs of any size and is tight up to logarithmic factors for instances having $\delta(\mathcal{G}) < 100 \ln K$.

**Theorem 10.** *Pick any directed or undirected, weakly observable graph $G = (V, E)$ with $|V| \geq 2$ and any $\varepsilon \in (0, 1]$. There exists a stochastic feedback graph $\mathcal{G}$ with $\mathrm{supp}(\mathcal{G}) = G$ and such that, for all $T \geq 2\sqrt{2}/\varepsilon$ and for any possibly randomized algorithm $\mathcal{A}$, there exists a sequence $\ell_1, \ldots, \ell_T$ of loss functions on which the expected regret of $\mathcal{A}$ with respect to the stochastic generation of $G_1, \ldots, G_T \sim \mathcal{G}$ is at least $\frac{\sqrt{2}}{16}\varepsilon^{-1/3}T^{2/3}$.*

*Proof.* The proof follows a similar structure as that of Alon et al. [2015, Theorem 11]. We consider the same instance constituted by a graph $G = (V, E)$ having $|V| \geq 3$ vertices, since it is the minimum number of vertices in order for $G$ to be weakly observable. In fact, any graph with exactly 2 vertices is either unobservable or strongly observable. By definition, there exists a vertex in this graph with no self-loop and with at least one incoming edge missing from any of the remaining vertices. Without loss of generality, let $v = 1$ be such a vertex and let $2 \notin N_G^{\mathrm{in}}(v)$ be one of the vertices without an edge towards $v$. We may consider the case where all edge probabilities are set to $\varepsilon$ (implying that $\mathcal{G} = \mathcal{G}_\varepsilon$ and $\mathrm{supp}(\mathcal{G}) = G$), given that we essentially assume the adversary can select them.

We can apply Yao's minimax principle, as usual, to reduce this problem to that of lower bounding the expected regret for any deterministic algorithm against a randomized adversary. Hence, we need to design a distribution for the loss functions $\ell_1, \ldots, \ell_T$ provided to the algorithm. Let $\beta = \frac{1}{2\sqrt{2}}(\varepsilon T)^{-1/3} \in [0, \frac{1}{4}]$ and pick $Z \in \{-1, +1\}$ uniformly at random. For all $t$, we choose the losses such that $\ell_t(1) \sim \mathsf{Bern}(1/2 - \beta Z)$, $\ell_t(2) \sim \mathsf{Bern}(1/2)$, and $\ell_t(j) = 1$ for all $j \neq 1$ independently. Similarly to the construction in the proof of Theorem 9, we have "good" actions $\{1, 2\}$ incurring at most $\beta$ expected instantaneous regret, while all remaining actions are "bad" since they incur at least $1/2$ instantaneous regret in expectation.

We reuse the same definitions for $T_i$ and $X_i$ as in the proof of Theorem 9 for any fixed deterministic algorithm. On the other hand, we let $\mathbb{P}_1(\cdot) = \mathbb{P}(\cdot \mid Z = +1)$ and $\mathbb{P}_2(\cdot) = \mathbb{P}(\cdot \mid Z = -1)$. We analogously define $\mathbb{E}_1[\cdot] = \mathbb{E}[\cdot \mid Z = +1]$ and $\mathbb{E}_2[\cdot] = \mathbb{E}[\cdot \mid Z = -1]$. Finally, we introduce $\mathbb{P}_0(\cdot)$ and $\mathbb{E}_0[\cdot]$ obtained as the previous ones by setting $Z = 0$.

Following the same rationale that led to Equation (23), we can show that

$$\mathbb{E}_i[T_i] - \mathbb{E}_0[T_i] \leq \beta T \sqrt{2\varepsilon \mathbb{E}_i[X_1]}$$

for $i \in \{1, 2\}$. This implies, via similar steps as in Equation (24), that

$$\frac{1}{2}\mathbb{E}_1[T_1] + \frac{1}{2}\mathbb{E}_2[T_2] \leq \beta T \sqrt{2\varepsilon \mathbb{E}[X_1]} + \frac{T}{2} . \tag{25}$$

Finally, if $\mathbb{E}[X_1] > \frac{1}{32}\beta^{-2}\varepsilon^{-1}$, the algorithm's expected regret becomes

$$\max_{k \in V} \mathbb{E}\left[\sum_{t=1}^{T}(\ell_t(I_t) - \ell_t(k))\right] \geq \frac{1}{2}\mathbb{E}[X_1] > \frac{1}{64}\beta^{-2}\varepsilon^{-1} = \frac{1}{8}\varepsilon^{-1/3}T^{2/3} ,$$

where the last equality holds by our choice of $\beta$. Otherwise, when $\mathbb{E}[X_1] \leq \frac{1}{32}\beta^{-2}\varepsilon^{-1}$, the right-hand side of Equation (25) is bounded by $\frac{3}{4}T$ and thus the regret must be

$$\max_{k \in V} \mathbb{E}\left[\sum_{t=1}^{T}(\ell_t(I_t) - \ell_t(k))\right] \geq \frac{1}{2}\mathbb{E}_1[\beta(T - T_1)] + \frac{1}{2}\mathbb{E}_2[\beta(T - T_2)] \geq \frac{\beta}{4}T = \frac{\sqrt{2}}{16}\varepsilon^{-1/3}T^{2/3} .$$

$\square$

## D   Be Optimistic If You Can, Commit If You Must

In this section, we describe Algorithm 4 and the analysis we use to obtain the results of Section 5. First of all, we briefly state the rationale for the design of this new algorithm. The main idea is similar in spirit to that of EDGECATCHER: Algorithm 4 constantly updates the estimates for the edge probabilities of the underlying $\mathcal{G}$ and computes the best regret regime it can achieve. However, EDGECATCHER has to wait until it can determine the best regret

---

**Algorithm 4:** OPTIMISTICTHENCOMMITGRAPH (OTCG)

---

**Environment**: stochastic feedback graph $\mathcal{G}$, sequence of losses $\ell_1, \ell_2, \ldots, \ell_T$;
**Input**: time horizon $T$ and actions $V = \{1, 2, \ldots, K\}$;
**Initialize**: sample $I_1$ uniformly at random, receive $G_1$;
**for** $t = 2, \ldots, T$ **do**

    **if** *Equation* (26) *has never been true* **then**           `// optimistic phase`

        Set $\tilde{p}_t(j, i) = \frac{1}{t-1} \sum_{s=1}^{t-1} \mathbb{I}_{\{(j,i) \in E_s\}}$ ;

        Set $\hat{p}_t(j, i) = \tilde{p}_t(j, i) + \sqrt{\frac{2\tilde{p}_t(j,i)}{t-1} \ln(3K^2 T^2)} + \frac{3}{t-1} \ln(3K^2 T^2)$ ;

        Set $\hat{\mathcal{G}}_t^{\mathrm{UCB}} = \{\hat{p}_t(j, i) : i, j \in V\}$;

        Compute $\theta_t$ and $\varepsilon_t^\theta$ as in Equation (33) ;

        Set $\hat{\mathcal{G}}_t = \{\hat{p}_t(j, i) \mathbb{I}_{\{\hat{p}_t(j,i) \geq \varepsilon_t^\theta\}} : i, j \in V\}$ and $\hat{G}_t = \mathrm{supp}(\hat{\mathcal{G}}_t)$, ;

        Compute $p_t^{\min} = \min_i \min_{j \in N_{\hat{G}_t}^{\mathrm{in}}(i)} \hat{p}_t(j, i)$;

        Set $\gamma_t = \min\left\{ \left(\min_{s \in [2,t]} t p_s^{\min}\right)^{-1/2}, \frac{1}{2} \right\}$;

        Set $\eta_{t-1} = \left( 16/(\min_{s \in [2,t]} (p_s^{\min})^2) + 4t/(\min_{s \in [2,t]} p_s^{\min}) + \sum_{s=2}^{t-1} \theta_s(\hat{\mathcal{G}}_s) \right)^{-1/2}$;

        Set $\psi_t$ to be the uniform distribution over $V$;

        Set $q_t(i) \propto \exp\left( \eta_{t-1} \sum_{s=2}^{t-1} \tilde{\ell}_t(i) \right)$;

        Set $\pi_t(i) = (1 - \gamma_t) q_t(i) + \gamma_t \psi_t(i)$;

    **if** *Equation* (26) *is true for any* $t' - 1 < t$ **then**         `// commit phase`

        Set $t^\star$ to the first round $t' - 1$ in which Equation (26) is true;

        Set $\tilde{\mathcal{G}} = \{\tilde{p}(j, i) : i, j \in V\}$ as the stochastic graph with

        $\tilde{p}(j, i) = \frac{1}{t^\star} \sum_{s=1}^{t^\star} \mathbb{I}_{\{(j,i) \in E_s\}}$;

        Set $\hat{\mathcal{G}} = \{\tilde{p}(j, i) \mathbb{I}_{\{\tilde{p}(j,i) \geq \varepsilon_{t^\star}\}} : i, j \in V\}$ with $\varepsilon_{t^\star}$ as in Equation (36);

        Set $\hat{\mathcal{G}}_{\varepsilon_{\delta,\sigma}^\star} = \{\tilde{p}(j, i) \mathbb{I}_{\{\tilde{p}(j,i) \geq \varepsilon_{\delta,\sigma}^\star\}} : i, j \in V\}$ with $\varepsilon_{\delta,\sigma}^\star$ as in Equation (37);

        Set $\tilde{p}_t(j, i) = \frac{1}{t-1} \sum_{s=1}^{t-1} \mathbb{I}_{\{(j,i) \in E_s\}}$ ;

        Set $\hat{p}_t(j, i) = \tilde{p}_t(j, i) + \sqrt{\frac{2\tilde{p}_t(j,i)}{t-1} \ln(3K^2 T^2)} + \frac{3}{t-1} \ln(3K^2 T^2)$ ;

        Set $\hat{\mathcal{G}}_t^{\mathrm{UCB}} = \{\hat{p}_t(j, i) : i, j \in V\}$;

        Set $\hat{\mathcal{G}}_t = \hat{\mathcal{G}}_t^{\mathrm{UCB}}$ and $\hat{G}_t = \mathrm{supp}(\hat{\mathcal{G}}_t)$;

        Set $\gamma = \min\left\{ \left(\delta_{\mathsf{w}}(\hat{\mathcal{G}}_{\varepsilon_{\delta,\sigma}^\star}) \ln(KT)\right)^{1/3} T^{-1/3}, \frac{1}{2} \right\}$;

        Set $\eta = \sqrt{\ln(K) \left( 2T \left( \delta_{\mathsf{w}}(\hat{\mathcal{G}}_{\varepsilon_{\delta,\sigma}^\star})/\gamma + \sigma(\hat{\mathcal{G}}_{\varepsilon_{\delta,\sigma}^\star}) \right) \right)^{-1}}$;

        Set $\psi_t$ according to (38);

        Set $q_t(i) \propto \exp\left( \eta \sum_{s=t^\star+1}^{t-1} \tilde{\ell}_t(i) \right)$;

        Set $\pi_t(i) = (1 - \gamma) q_t(i) + \gamma \psi_t(i)$;

    Sample $I_t \sim \pi_t$;

    Receive $G_t$ and $\{(i, \ell_t(i)) : i \in N_{G_t}^{\mathrm{out}}(I_t)\}$;

    Compute $\tilde{\ell}_t(i)$ as in (6);

---

regime before actually tackling the learning task. On the contrary, Algorithm 4 begins by optimistically assuming that the best thresholded graph has a strongly observable support

while simultaneously updating the edge probability estimates; this is made possible given the additional assumption on receiving the realized graph $G_t = (V, E_t) \sim \mathcal{G}$ together with the observed losses at the end of each round $t$. At any point in time, as soon as Algorithm 4 finds that it can achieve a better regret regime by switching to the weakly observable one (by computing the optimal threshold on the current estimate for $\mathcal{G}$), it commits to weak observability. We can prove that this strategy is able to achieve the best possible regret over all thresholded feedback graphs, analogously to EDGECATCHER, but with a dependency on the improved graph-theoretic parameters introduced in Section 5.

Consequently, there are two regimes of Algorithm 4. In the first regime, the algorithm works under the assumption that $\text{supp}(\mathcal{G})$ is strongly observable; in the second regime, the algorithm works under the assumption that $\text{supp}(\mathcal{G})$ is observable. The switch happens in round $t^\star + 1$, where $t^\star$ is the first round $t - 1$ in which

$$\Psi_{t-1} \geq \Lambda_{t-1}, \tag{26}$$

is true. The term $\Psi_t$ is an upper bound on the regret after the first $t$ rounds, and is given by

$$\Psi_t = \min\Big\{ t, 2 + 11(\ln(3K^2T^2))^2 \max_{s\in[2,t]} \theta_s(\hat{\mathcal{G}}_s)$$
$$+ \big(12\ln(K) + 4\sqrt{2\ln(3K^2T^2)}\big)\sqrt{t \max_{s\in[2,t]} \theta_s(\hat{\mathcal{G}}_s)} \Big\} , \tag{27}$$

where $\hat{\mathcal{G}}_t$ minimizes $\theta_t$, which is defined in Equation (33). The term $\theta_t(\hat{\mathcal{G}}_t)$ is an upper bound the second-order term in the regret bound of Exponential Weights. Crucially, the same term $\theta_t(\hat{\mathcal{G}}_t)$ does not require us to compute a weighted independence number at each round: we can explicitly compute it in $O(K^4)$ time. Furthermore, in Lemma 11 we show that, conditioning on the event $\mathcal{K}$, the term $\theta_t(\hat{\mathcal{G}}_t)$ is upper bounded by the minimum thresholded weighted independence number of $\mathcal{G}$, which in turn is useful when bounding the regret. We recall that the event $\mathcal{K}$, introduced in Section 5, corresponds to the event that

$$|\tilde{p}_t(j,i) - p(j,i)| \leq \sqrt{\frac{2\tilde{p}_t(j,i)}{t-1}\ln(3K^2T^2)} + \frac{3}{t-1}\ln(3K^2T^2), \quad \forall(j,i) \in V \times V$$

for all $t \geq 2$ simultaneously.

Similarly, $\Lambda_t$ is an upper bound on the regret of Algorithm 4 *if it were to switch regime in round $t$* and is given by

$$\Lambda_t = \min_\varepsilon \Big\{ 41T^{2/3}\big(\ln(3K^2T^2)\delta_{\mathsf{w}}((\hat{\mathcal{G}}_t)_\varepsilon)\big)^{1/3} + 41\sqrt{\ln(3K^2T^2)\sigma((\hat{\mathcal{G}}_t)_\varepsilon)T} \Big\} , \tag{28}$$

where $\hat{\mathcal{G}}_t = \{\tilde{p}_t(j,i)\mathbb{I}_{\{\tilde{p}_t(j,i)\geq 60\ln(KT)/t\}} : i, j \in V\}$. In other words, Algorithm 4 changes regime whenever it thinks that the regret of a (weakly) observable graph is smaller than the regret of a strongly observable graph. In the following, we prove that $\Psi_t$ and $\Lambda_t$ are indeed upper bounds on the regret, but first we state Lemma 6, which is a central result in this section. More precisely, it provides an upper bound for the cost of not using the exact edge probabilities $p(j,i)$ but instead using upper confidence bound estimates $\hat{p}_t(j,i)$. Note that the bound scales with $\bar{\pi}_t(i) = \sum_{j\in N^{\text{in}}_{\hat{\mathcal{G}}_t}(i)} \pi_t(j)$. For $\mathcal{G}_\varepsilon$ having a strongly observable support, this is an important property of the bound since we require that $\bar{\pi}_t(i) \leq 1 - \pi_t(i)$ for vertices $i$ without a self-loop in $\text{supp}(\mathcal{G}_\varepsilon)$ to ensure that we can bound the regret in terms of the weighted independence number.

**Lemma 6.** *Define $\bar{\pi}_t(i) = \sum_{j\in N^{\text{in}}_{\hat{\mathcal{G}}_t}(i)} \pi_t(j)$. For any distribution $u$ over $[K]$ and $t^\star \leq T$, with estimator (6) we have that*

$$\mathbb{E}\left[\sum_{t=1}^{t^\star}\sum_{i=1}^{K}(\pi_t(i) - u(i))\ell_t(i)\right] \leq 2 + \sum_{t=2}^{t^\star}\mathbb{E}\left[\frac{6\ln(3K^2T^2)}{t-1}\sum_{i=1}^{K}\frac{\pi_t(i)\bar{\pi}_t(i)}{\hat{P}_t(i)} \,\bigg|\, \mathcal{K}\right]$$
$$+ \mathbb{E}\left[\sum_{t=2}^{t^\star}2\sqrt{2\frac{\ln(3K^2T^2)}{t-1}}\sqrt{\sum_{i=1}^{K}\frac{\pi_t(i)\bar{\pi}_t(i)}{\hat{P}_t(i)}} \,\bigg|\, \mathcal{K}\right] + \mathbb{E}\left[\sum_{t=2}^{t^\star}\sum_{i=1}^{K}(\pi_t(i) - u(i))\tilde{\ell}_t(i) \,\bigg|\, \mathcal{K}\right].$$

*Proof.* For $t > 1$, by the empirical Bernstein bound [Audibert et al., 2007, Theorem 1], with probability at least $1 - \frac{1}{K^2 T^2}$ we have that

$$
\left| \frac{1}{t-1} \sum_{s=1}^{t-1} \mathbb{I}_{\{(j,i) \in E_s\}} - p(j,i) \right| \leq \sqrt{2 \frac{\overline{\sigma}_t^2 \ln(3K^2T^2)}{t-1}} + \frac{3}{t-1} \ln(3K^2T^2)
$$

$$
\leq \sqrt{\frac{2\hat{p}_t(j,i)}{t-1} \ln(3K^2T^2)} + \frac{3}{t-1} \ln(3K^2T^2) , \quad (29)
$$

where we used the fact that

$$
\overline{\sigma}_t^2 = \frac{1}{t-1} \sum_{s'=1}^{t-1} \left( \mathbb{I}_{\{(j,i) \in E_{s'}\}} - \frac{1}{t-1} \sum_{s=1}^{t-1} \mathbb{I}_{\{(j,i) \in E_s\}} \right)^2 \leq \frac{1}{t-1} \sum_{s=1}^{t-1} \mathbb{I}_{\{(j,i) \in E_s\}} \leq \hat{p}_t(j,i) .
$$

Thus, by the union bound over $K^2$ edges and $t^\star$ rounds, we have that equation (29) holds for all edges and time steps $t \geq 2$ with probability at least $1 - \frac{1}{T}$. This means that $\mathbb{P}(\mathcal{K}) \geq 1 - \frac{1}{T}$ by definition of $\mathcal{K}$.

By using the tower rule and the fact that $\ell_t(i) \in [0,1]$, we can see that

$$
\mathbb{E}\left[ \sum_{t=1}^{t^\star} \sum_{i=1}^{K} (\pi_t(i) - u(i))\ell_t(i) \right]
$$

$$
= \mathbb{P}\left(\overline{\mathcal{K}}\right) \mathbb{E}\left[ \sum_{t=1}^{t^\star} \sum_{i=1}^{K} (\pi_t(i) - u(i))\ell_t(i) \,\middle|\, \overline{\mathcal{K}} \right] + (1 - \mathbb{P}(\mathcal{K}))\mathbb{E}\left[ \sum_{t=1}^{t^\star} \sum_{i=1}^{K} (\pi_t(i) - u(i))\ell_t(i) \,\middle|\, \mathcal{K} \right]
$$

$$
\leq \mathbb{P}\left(\overline{\mathcal{K}}\right) T + \mathbb{E}\left[ \sum_{t=1}^{t^\star} \sum_{i=1}^{K} (\pi_t(i) - u(i))\ell_t(i) \,\middle|\, \mathcal{K} \right]
$$

$$
\leq 2 + \mathbb{E}\left[ \sum_{t=2}^{t^\star} \sum_{i=1}^{K} (\pi_t(i) - u(i))\ell_t(i) \,\middle|\, \mathcal{K} \right] . \quad (30)
$$

Let $X_t = \mathbb{I}_{\{i \in N_{G_t}^{\text{out}}(I_t)\} \cap \{i \in N_{\hat{G}_t}^{\text{out}}(I_t)\}}$ be the indicator of the event that $i$ belongs to both $N_{G_t}^{\text{out}}(I_t)$ and $N_{\hat{G}_t}^{\text{out}}(I_t)$, and let $\xi_t(i) = \hat{P}_t(i) - P_t(i) = \sum_{j \in N_{\hat{G}_t}^{\text{in}}(i)} \pi_t(j)(\hat{p}_t(j,i) - p(j,i))$. We continue by applying Lemma 13 on the expectation in the right-hand side of (30), obtaining that

$$
\mathbb{E}\left[ \sum_{t=2}^{t^\star} \sum_{i=1}^{K} (\pi_t(i) - u(i))\ell_t(i) \,\middle|\, \mathcal{K} \right]
$$

$$
= \mathbb{E}\left[ \sum_{t=2}^{t^\star} \sum_{i=1}^{K} (\pi_t(i) - u(i))\tilde{\ell}_t(i) \,\middle|\, \mathcal{K} \right] + \mathbb{E}\left[ \sum_{t=2}^{t^\star} \sum_{i=1}^{K} (\pi_t(i) - u(i))\xi_t(i) \frac{X_t \ell_t(i)}{P_t(i)\hat{P}_t(i)} \,\middle|\, \mathcal{K} \right]
$$

$$
\leq \mathbb{E}\left[ \sum_{t=2}^{t^\star} \sum_{i=1}^{K} (\pi_t(i) - u(i))\tilde{\ell}_t(i) \,\middle|\, \mathcal{K} \right] + \mathbb{E}\left[ \sum_{t=2}^{t^\star} \sum_{i=1}^{K} \pi_t(i)\xi_t(i) \frac{X_t \ell_t(i)}{P_t(i)\hat{P}_t(i)} \,\middle|\, \mathcal{K} \right] ,
$$

where the inequality is due to the fact that the loss is nonnegative and the fact that $\xi_t(i) > 0$ because $\hat{p}_t(j,i) - p(j,i) > 0$ is true, given $\mathcal{K}$. We already know that $\hat{p}_t(j,i) \geq \tilde{p}_t(j,i)$ by definition of $\hat{p}_t(j,i)$. As long as $\mathcal{K}$ holds, we also know that $\tilde{p}_t(j,i) - p(j,i) \leq \sqrt{\frac{2\tilde{p}_t(j,i)}{t-1} \ln(3K^2T^2)} + \frac{3}{t-1} \ln(3K^2T^2)$ is true. Then, we can use all the above observations

to demonstrate that the term $\xi_t(i)$ satisfies

$$\xi_t(i) = \sum_{j \in N^{\text{in}}_{\hat{G}_t}(i)} \pi_t(j)(\hat{p}_t(j,i) - p(j,i))$$

$$\leq \sum_{j \in N^{\text{in}}_{\hat{G}_t}(i)} \pi_t(j) \left( \sqrt{\frac{2\hat{p}_t(j,i)}{t-1} \ln(3K^2T^2)} + \frac{3}{t-1} \ln(3K^2T^2) \right)$$

$$+ \sum_{j \in N^{\text{in}}_{\hat{G}_t}(i)} \pi_t(j) \left( \tilde{p}_t(j,i) - p(j,i) \right)$$

$$\leq 2 \sum_{j \in N^{\text{in}}_{\hat{G}_t}(i)} \pi_t(j) \left( \sqrt{\frac{2\hat{p}_t(j,i)}{t-1} \ln(3K^2T^2)} + \frac{3}{t-1} \ln(3K^2T^2) \right) \qquad (31)$$

By the Cauchy-Schwarz inequality, it holds that

$$\sum_{j \in N^{\text{in}}_{\hat{G}_t}(i)} \pi_t(j)\sqrt{a_j} = \sum_{j \in N^{\text{in}}_{\hat{G}_t}(i)} \sqrt{\pi_t(j)}\sqrt{\pi_t(j)a_j} \leq \sqrt{\bar{\pi}_t(i) \sum_{j \in N^{\text{in}}_{\hat{G}_t}(i)} \pi_t(j)a_j}$$

with $a_j \geq 0$ for all $j \in N^{\text{in}}_{\hat{G}_t}(i)$, where we recall that $\bar{\pi}_t(i) = \sum_{j \in N^{\text{in}}_{\hat{G}_t}(i)} \pi_t(j)$. We can use this property to further bound $\xi_t(i)$ in (31) as

$$\xi_t(i) \leq 2 \sum_{j \in N^{\text{in}}_{\hat{G}_t}(i)} \pi_t(j) \left( \sqrt{\frac{2\hat{p}_t(j,i)}{t-1} \ln(3K^2T^2)} + \frac{3}{t-1} \ln(3K^2T^2) \right)$$

$$\leq 2\sqrt{2\bar{\pi}_t(i)\frac{\hat{P}_t(i)\ln(3K^2T^2)}{t-1}} + \bar{\pi}_t(i)\frac{6\ln(3K^2T^2)}{t-1} \quad .$$

At this point, we can use the inequality for $\xi_t(i)$ to show that

$$\mathbb{E}\left[ \sum_{t=2}^{t^\star} \sum_{i=1}^{K} \pi_t(i)\xi_t(i) \frac{X_t\ell_t(i)}{P_t(i)\hat{P}_t(i)} \;\middle|\; \mathcal{K} \right]$$

$$\leq \mathbb{E}\left[ \sum_{t=2}^{t^\star} 2\sqrt{2\frac{\ln(3K^2T^2)}{t-1}} \sum_{i=1}^{K} \pi_t(i) \frac{X_t\ell_t(i)\sqrt{\bar{\pi}_t(i)}}{P_t(i)\sqrt{\hat{P}_t(i)}} \;\middle|\; \mathcal{K} \right]$$

$$+ \sum_{t=2}^{t^\star} \mathbb{E}\left[ \frac{6\ln(3K^2T^2)}{t-1} \sum_{i=1}^{K} \pi_t(i)\bar{\pi}_t(i) \frac{X_t\ell_t(i)}{P_t(i)\hat{P}_t(i)} \;\middle|\; \mathcal{K} \right]$$

$$\leq \mathbb{E}\left[ \sum_{t=2}^{t^\star} 2\sqrt{2\frac{\ln(3K^2T^2)}{t-1}} \sum_{i=1}^{K} \pi_t(i)\sqrt{\frac{\bar{\pi}_t(i)}{\hat{P}_t(i)}} \;\middle|\; \mathcal{K} \right] \qquad (32)$$

$$+ \sum_{t=2}^{t^\star} \mathbb{E}\left[ \frac{6\ln(3K^2T^2)}{t-1} \sum_{i=1}^{K} \frac{\bar{\pi}_t(i)\pi_t(i)}{\hat{P}_t(i)} \;\middle|\; \mathcal{K} \right]$$

$$\leq \mathbb{E}\left[ \sum_{t=2}^{t^\star} 2\sqrt{2\frac{\ln(3K^2T^2)}{t-1}} \sqrt{\sum_{i=1}^{K} \frac{\pi_t(i)\bar{\pi}_t(i)}{\hat{P}_t(i)}} \;\middle|\; \mathcal{K} \right]$$

$$+ \sum_{t=2}^{t^\star} \mathbb{E}\left[ \frac{6\ln(3K^2T^2)}{t-1} \sum_{i=1}^{K} \frac{\pi_t(i)\bar{\pi}_t(i)}{\hat{P}_t(i)} \;\middle|\; \mathcal{K} \right] ,$$

where in the second inequality we used the fact that $\ell_t(i) \leq 1$ and that $\mathbb{E}_{t-1}[X_t] = P_t(i)$, while the final inequality is Jensen's inequality for concave functions.

By combining the above, we may complete the proof:

$$\mathbb{E}\left[\sum_{t=1}^{t^\star}\sum_{i=1}^{K}(\pi_t(i)-u(i))\ell_t(i)\right] \le 2 + \sum_{t=2}^{t^\star}\mathbb{E}\left[\frac{6\ln(3K^2T^2)}{t-1}\sum_{i=1}^{K}\frac{\pi_t(i)\bar{\pi}_t(i)}{\hat{P}_t(i)}\,\middle|\,\mathcal{K}\right]$$

$$+\,\mathbb{E}\left[\sum_{t=2}^{t^\star}2\sqrt{2\frac{\ln(3K^2T^2)}{t-1}}\sqrt{\sum_{i=1}^{K}\frac{\pi_t(i)\bar{\pi}_t(i)}{\hat{P}_t(i)}}\,\middle|\,\mathcal{K}\right] + \mathbb{E}\left[\sum_{t=2}^{t^\star}\sum_{i=1}^{K}(\pi_t(i)-u(i))\tilde{\ell}_t(i)\,\middle|\,\mathcal{K}\right].$$

$\square$

## D.1 Initial Regime of OTCG

To understand the initial regime of OTCG (Algorithm 4), consider the following. Since the support of $\hat{\mathcal{G}}_t^{\mathrm{UCB}}$ is the complete graph, there always exists a threshold $\varepsilon$ for which $\mathrm{supp}((\hat{\mathcal{G}}_t^{\mathrm{UCB}})_\varepsilon)$ is strongly observable. For ease of notation, given any stochastic feedback graph $\mathcal{G}$ with edge probabilities $p(j,i)$, we introduce

$$P_t(i,\mathcal{G}) = \sum_{j\in N^{\mathrm{in}}_{\mathrm{supp}(\mathcal{G})}(i)}\pi_t(j)p(j,i)\ .$$

Denote by $\mathcal{S}$ the family of strongly observable graphs over vertices $V=[K]$; we can then define $\varepsilon_t^\theta$ as

$$\varepsilon_t^\theta = \underset{\varepsilon\,:\,\mathrm{supp}((\hat{\mathcal{G}}_t^{\mathrm{UCB}})_\varepsilon)\in\mathcal{S}}{\arg\min}\theta_t((\hat{\mathcal{G}}_t^{\mathrm{UCB}})_\varepsilon)$$

$$= \underset{\varepsilon\,:\,\mathrm{supp}((\hat{\mathcal{G}}_t^{\mathrm{UCB}})_\varepsilon)\in\mathcal{S}}{\arg\min}\left(\frac{2}{\min_i\min_{j\in N^{\mathrm{in}}_{\mathrm{supp}((\hat{\mathcal{G}}_t^{\mathrm{UCB}})_\varepsilon)}(i)}\hat{p}_t(j,i)} + \sum_{i\in N^{\mathrm{in}}_{\mathrm{supp}((\hat{\mathcal{G}}_t^{\mathrm{UCB}})_\varepsilon)}(i)}\frac{2\pi_t(i)}{P_t(i,(\hat{\mathcal{G}}_t^{\mathrm{UCB}})_\varepsilon)}\right).$$

(33)

A crucial property of $\hat{\mathcal{G}}_t$ (that is, $\hat{\mathcal{G}}_t^{\mathrm{UCB}}$ thresholded at $\varepsilon_t^\theta$) is that, if $\hat{p}_t(j,i)\ge p(j,i)$ for all edges $(j,i)$, by Lemma 11 we have that

$$\min_{\varepsilon\,:\,\mathrm{supp}((\hat{\mathcal{G}}_t^{\mathrm{UCB}})_\varepsilon)\in\mathcal{S}}\theta_t((\hat{\mathcal{G}}_t^{\mathrm{UCB}})_\varepsilon) = \tilde{O}\left(\min_{\varepsilon\,:\,\mathrm{supp}(\mathcal{G}_\varepsilon)\in\mathcal{S}}\alpha_{\mathsf{w}}(\mathcal{G}_\varepsilon)\right)\ ,$$

which is a property we will use when computing the final regret bound of Algorithm 4. It also ensures that we can bound the cost of not knowing $p(j,i)$ in Lemma 6 by $\min_{\varepsilon\,:\,\mathrm{supp}((\hat{\mathcal{G}}_t^{\mathrm{UCB}})_\varepsilon)\in\mathcal{S}}\theta_t((\hat{\mathcal{G}}_t^{\mathrm{UCB}})_\varepsilon)$, which is also important in computing the final regret bound of Algorithm 4. We thus upper bound the regret of the initial regime of OTCG in terms of $\theta_t$ in what follows.

**Lemma 7.** *For any distribution $u$ over $[K]$, after $t^\star\le T$ rounds Algorithm 4 guarantees*

$$\mathbb{E}\left[\sum_{t=1}^{t^\star}\sum_{i=1}^{K}(\pi_t(i)-u(i))\tilde{\ell}_t(i)\,\middle|\,\mathcal{K}\right] \le 2 + 11(\ln(3K^2T^2))^2\,\mathbb{E}\left[\max_{t\in[2,t^\star]}\theta_t(\hat{\mathcal{G}}_t)\,\middle|\,\mathcal{K}\right]$$

$$+\left(12\ln(K)+4\sqrt{2\ln(3K^2T^2)}\right)\mathbb{E}\left[\sqrt{t^\star\max_{t\in[2,t^\star]}\theta_t(\hat{\mathcal{G}}_t)}\,\middle|\,\mathcal{K}\right]\ .$$

*Proof.* We start with an application of Lemma 6:

$$\mathbb{E}\left[\sum_{t=1}^{t^\star}\sum_{i=1}^{K}(\pi_t(i)-u(i))\ell_t(i)\right] \le 2 + \sum_{t=2}^{t^\star}\mathbb{E}\left[\frac{6\ln(3K^2T^2)}{t-1}\sum_{i=1}^{K}\frac{\pi_t(i)\bar{\pi}_t(i)}{\hat{P}_t(i)}\,\middle|\,\mathcal{K}\right]$$

$$+\,\mathbb{E}\left[\sum_{t=2}^{t^\star}2\sqrt{2\frac{\ln(3K^2T^2)}{t-1}}\sqrt{\sum_{i=1}^{K}\frac{\pi_t(i)\bar{\pi}_t(i)}{\hat{P}_t(i)}}\,\middle|\,\mathcal{K}\right] + \mathbb{E}\left[\sum_{t=2}^{t^\star}\sum_{i=1}^{K}(\pi_t(i)-u(i))\tilde{\ell}_t(i)\,\middle|\,\mathcal{K}\right]\ ,$$

where, we recall it, $\bar{\pi}_t(i) = \sum_{j \in N^{\text{in}}_{\hat{G}_t}(i)} \pi_t(j)$. Now, for $i$ without a self-loop in $\hat{G}_t$ we have that $\bar{\pi}_t(i) \le 1 - \pi_t(i)$. Now, conditioning on $\mathcal{K}$, we may follow the reasoning surrounding Equation (35) to find that

$$\sum_{i=1}^{K} \frac{\pi_t(i)\bar{\pi}_t(i)}{\hat{P}_t(i)} \le \theta_t(\hat{G}_t) \ .$$

We now use $\sum_{t=1}^{T} \frac{1}{t} \le \ln(T) + 1$, $\sum_{t=1}^{T} \frac{1}{\sqrt{t}} \le 2\sqrt{T}$, and the above inequality to obtain that

$$\mathbb{E}\left[\sum_{t=1}^{t^\star}\sum_{i=1}^{K}(\pi_t(i) - u(i))\ell_t(i)\right] \le 2 + \sum_{t=2}^{t^\star}\mathbb{E}\left[\frac{6\ln(3K^2T^2)}{t-1}\theta_t(\hat{G}_t) \,\bigg|\, \mathcal{K}\right]$$

$$+ \mathbb{E}\left[\sum_{t=2}^{t^\star} 2\sqrt{2\frac{\ln(3K^2T^2)}{t-1}}\sqrt{\theta_t(\hat{G}_t)} \,\bigg|\, \mathcal{K}\right] + \mathbb{E}\left[\sum_{t=2}^{t^\star}\sum_{i=1}^{K}(\pi_t(i) - u(i))\tilde{\ell}_t(i) \,\bigg|\, \mathcal{K}\right]$$

$$\le 2 + 6(\ln(3K^2T^2))^2 \,\mathbb{E}\left[\max_{t \in [2,t^\star]}\theta_t(\hat{G}_t) \,\bigg|\, \mathcal{K}\right] + \mathbb{E}\left[4\sqrt{2\ln(3K^2T^2)t^\star \max_{t \in [2,t^\star]}\theta_t(\hat{G}_t)} \,\bigg|\, \mathcal{K}\right]$$

$$+ \mathbb{E}\left[\sum_{t=2}^{t^\star}\sum_{i=1}^{K}(\pi_t(i) - u(i))\tilde{\ell}_t(i) \,\bigg|\, \mathcal{K}\right] \ .$$

By applying Lemma 8, we can complete the proof:

$$\mathbb{E}\left[\sum_{t=1}^{t^\star}\sum_{i=1}^{K}(\pi_t(i) - u(i))\ell_t(i)\right] \le 2 + 6(\ln(3K^2T^2))^2 \,\mathbb{E}\left[\max_{t \in [2,t^\star]}\theta_t(\hat{G}_t) \,\bigg|\, \mathcal{K}\right]$$

$$+ \mathbb{E}\left[4\sqrt{2\ln(3K^2T^2)t^\star \max_{t \in [2,t^\star]}\theta_t(\hat{G}_t)} \,\bigg|\, \mathcal{K}\right]$$

$$+ \mathbb{E}\left[7\ln(K)\sqrt{\sum_{t=2}^{t^\star}\theta_t(\hat{G}_t) + \max_{t \in [2,t^\star]}\theta_t(\hat{G}_t)} \,\bigg|\, \mathcal{K}\right]$$

$$+ \mathbb{E}\left[\max_{t \in [2,t^\star]}\frac{4\ln(K)}{p_t^{\min}} + 5\ln(K)\sqrt{\max_{t \in [2,t^\star]}\frac{t^\star}{p_t^{\min}}} \,\bigg|\, \mathcal{K}\right]$$

$$\le 2 + 11(\ln(3K^2T^2))^2 \,\mathbb{E}\left[\max_{t \in [2,t^\star]}\theta_t(\hat{G}_t) \,\bigg|\, \mathcal{K}\right]$$

$$+ \left(12\ln(K) + 4\sqrt{2\ln(3K^2T^2)}\right)\mathbb{E}\left[\sqrt{t^\star \max_{t \in [2,t^\star]}\theta_t(\hat{G}_t)} \,\bigg|\, \mathcal{K}\right] \ ,$$

where we used that $\frac{1}{p_t^{\min}} \le \theta_t(\hat{G}_t)$ for all $t \in [2, t^\star]$. $\qquad\square$

In the proof of Lemma 7 we make use of the following auxiliary result, which bounds the regret of $\pi_t$ given $\mathcal{K}$.

**Lemma 8.** *For any distribution $u$ over $[K]$, after $t^\star \le T$ rounds Algorithm 4 guarantees*

$$\mathbb{E}\left[\sum_{t=2}^{t^\star}\sum_{i=1}^{K}(\pi_t(i) - u(i))\tilde{\ell}_t(i) \,\bigg|\, \mathcal{K}\right] \le \mathbb{E}\left[7\ln(K)\sqrt{\sum_{t=2}^{t^\star}\theta_t(\hat{G}_t) + \max_{t \in [2,t^\star]}\theta_t(\hat{G}_t)} \,\bigg|\, \mathcal{K}\right]$$

$$+ \mathbb{E}\left[\max_{t \in [2,t^\star]}\frac{4\ln(K)}{p_t^{\min}} + 5\ln(K)\sqrt{\max_{t \in [2,t^\star]}\frac{t^\star}{p_t^{\min}}} \,\bigg|\, \mathcal{K}\right] \ .$$

*Proof.* We want to apply Lemma 12, which bounds the regret of Exponential Weights. Recall that Algorithm 4 defines

$$p_t^{\min} = \min_{i \in V} \min_{j \in N^{\text{in}}_{\text{supp}(\hat{G}_t)}(i)} \hat{p}_t(j, i)$$

as the minimum (positive) edge probability in $\hat{\mathcal{G}}_t$. Observe that for any node $i$ without a self-loop in $\text{supp}(\hat{\mathcal{G}}_t)$ we have that

$$
\begin{aligned}
\hat{P}_t(i) &= \sum_{j \neq i} \hat{p}_t(j,i) \left( (1-\gamma_t) q_t(i) + \frac{\gamma_t}{K} \right) \\
&\geq p_t^{\min} \sum_{j \neq i} \left( (1-\gamma_t) q_t(i) + \frac{\gamma_t}{K} \right) \\
&= (1 - \pi_t(i)) p_t^{\min} \\
&= \left( 1 - (1-\gamma_t) q_t(i) - \frac{\gamma_t}{K} \right) p_t^{\min} \\
&\geq \frac{\gamma_t}{2} p_t^{\min} \ .
\end{aligned}
\tag{34}
$$

Using (34) and the definitions of $\eta_{t-1}$ and $\gamma_t$, together with the fact that $\ell_t(i) \in [0,1]$, we can see that

$$
\eta_{t-1} \tilde{\ell}_t(i) \leq \eta_{t-1} \frac{1}{\hat{P}_t(i)} \leq \eta_{t-1} \frac{2}{\gamma_t p_t^{\min}} \leq 1 \ ,
$$

where the last inequality is due to the fact that $\eta_{t-1} \leq \frac{1}{2} \gamma_t p_t^{\min}$. Given event $\mathcal{K}$, since for any node $i$ without a self-loop in $\text{supp}(\hat{\mathcal{G}}_t)$ we have that $\eta_{t-1} \tilde{\ell}_t(i) \leq 1$, we may apply Lemma 12 with $S_t = S = \{i : i \notin N_{\text{supp}(\hat{\mathcal{G}}_t)}^{\text{in}}(i)\}$ to obtain that

$$
\mathbb{E}\left[ \sum_{t=2}^{t^\star} \sum_{i=1}^{K} (q_t(i) - u(i)) \tilde{\ell}_t(i) \,\middle|\, \mathcal{K} \right]
$$

$$
\leq \mathbb{E}\left[ \frac{\ln K}{\eta_{t^\star}} + \sum_{t=2}^{t^\star} \eta_{t-1} \left( \sum_{i \in S_t} q_t(i)(1-q_t(i)) \tilde{\ell}_t(i)^2 + \sum_{i \notin S_t} q_t(i) \tilde{\ell}_t(i)^2 \right) \,\middle|\, \mathcal{K} \right] \ .
$$

We now bound

$$
\begin{aligned}
\mathbb{E}\left[ \sum_{i \in S_t} q_t(i)(1-q_t(i)) \tilde{\ell}_t(i)^2 \,\middle|\, \mathcal{K} \right] &= \mathbb{E}\left[ \sum_{i \in S_t} q_t(i)(1-q_t(i)) \frac{P_t(i) \ell_t(i)^2}{\hat{P}_t(i)(P_t(i) + \xi_t(i))} \,\middle|\, \mathcal{K} \right] \\
&\leq \mathbb{E}\left[ \sum_{i \in S_t} q_t(i) \frac{(1-q_t(i))}{\hat{P}_t(i)} \,\middle|\, \mathcal{K} \right] \\
&= \mathbb{E}\left[ \sum_{i \in S_t} \frac{q_t(i)(1-q_t(i))}{P_t(i, \hat{\mathcal{G}}_t)} \,\middle|\, \mathcal{K} \right] \\
&\leq \mathbb{E}\left[ \sum_{i \in S_t} \frac{2q_t(i)}{p_t^{\min}} \,\middle|\, \mathcal{K} \right] \leq \mathbb{E}\left[ \frac{2}{p_t^{\min}} \,\middle|\, \mathcal{K} \right] \ .
\end{aligned}
$$

For $i \notin S_t$, since $\pi_t(i) \geq \frac{1}{2} q_t(i)$ and $\hat{P}_t(i) - P_t(i) \geq 0$ given $\mathcal{K}$, we have that

$$
\mathbb{E}\left[ \sum_{i \notin S_t} q_t(i) \tilde{\ell}_t(i)^2 \,\middle|\, \mathcal{K} \right] \leq \mathbb{E}\left[ \sum_{i \notin S_t} \frac{q_t(i)}{\hat{P}_t(i)} \,\middle|\, \mathcal{K} \right] \leq \mathbb{E}\left[ \sum_{i \notin S_t} \frac{2\pi_t(i)}{P_t(i, \hat{\mathcal{G}}_t)} \,\middle|\, \mathcal{K} \right] \ ,
$$

which combined with the preceding inequality means that, given $\mathcal{K}$, we have that

$$
\sum_{i \in S_t} \frac{q_t(i)(1-q_t(i))}{\hat{P}_t(i)} + \sum_{i \notin S_t} \frac{q_t(i)}{\hat{P}_t(i)} \leq \frac{2}{p_t^{\min}} + \sum_{i \notin S_t} \frac{2\pi_t(i)}{P_t(i, \hat{\mathcal{G}}_t)} = \theta_t(\hat{\mathcal{G}}_t) \ .
\tag{35}
$$

Therefore, we have that

$$\mathbb{E}\left[\sum_{t=2}^{t^\star}\sum_{i=1}^{K}(q_t(i)-u(i))\tilde{\ell}_t(i)\,\middle|\,\mathcal{K}\right]$$

$$\leq \mathbb{E}\left[\frac{\ln K}{\eta_{t^\star}}+\sum_{t=2}^{t^\star}\eta_{t-1}\left(\sum_{i\in S_t}\frac{q_t(i)(1-q_t(i))}{P_t(i,\hat{\mathcal{G}}_t)}+\sum_{i\notin S_t}\frac{q_t(i)}{P_t(i,\hat{\mathcal{G}}_t)}\right)\,\middle|\,\mathcal{K}\right]$$

$$\leq \mathbb{E}\left[\frac{\ln K}{\eta_{t^\star}}+\sum_{t=2}^{t^\star}\eta_{t-1}\theta_t(\hat{\mathcal{G}}_t)\,\middle|\,\mathcal{K}\right]\ .$$

Now, using a slightly modified version of [Gaillard et al., 2014, Lemma 14] (replacing $|a_i|\leq 1$ by $|a_i|\leq \max_i |a_i|$) we can see that

$$\sum_{t=2}^{t^\star}\eta_{t-1}\theta_t(\hat{\mathcal{G}}_t)\leq \sum_{t=2}^{t^\star}\theta_t(\hat{\mathcal{G}}_t)\sqrt{1+\sum_{s=2}^{t-1}\theta_s(\hat{\mathcal{G}}_s)}$$

$$\leq 3\sqrt{\sum_{t=2}^{t^\star}\theta_t(\hat{\mathcal{G}}_t)}+\max_{t\in[2,t^\star]}\theta_t(\hat{\mathcal{G}}_t)\ .$$

As a final step in this proof, we want to consider the distribution $\pi_t$ the algorithm actually samples actions from instead of $q_t$. We can bound $\sum_{t=2}^{t^\star}\gamma_t\leq 2\sqrt{\max_{t\in[2,t^\star]}\frac{t^\star}{p_t^{\min}}}$ and

$$\frac{1}{\eta_{t^\star}}\leq \frac{4}{\min_{t\in[2,t^\star]}p_t^{\min}}+\sqrt{\frac{t^\star}{\min_{t\in[2,t^\star]}p_t^{\min}}}+\sqrt{\sum_{t=2}^{t^\star}\theta_t(\hat{\mathcal{G}}_t)}\ .$$

Thus, combining the above we find that

$$\mathbb{E}\left[\sum_{t=2}^{t^\star}\sum_{i=1}^{K}(\pi_t(i)-u(i))\tilde{\ell}_t(i)\,\middle|\,\mathcal{K}\right]\leq \mathbb{E}\left[\sum_{t=2}^{t^\star}\sum_{i=1}^{K}(q_t(i)-u(i))\tilde{\ell}_t(i)+\sum_{t=2}^{t^\star}\gamma_t\,\middle|\,\mathcal{K}\right]$$

$$\leq \mathbb{E}\left[7\ln(K)\sqrt{\sum_{t=2}^{t^\star}\theta_t(\hat{\mathcal{G}}_t)}+\max_{t\in[2,t^\star]}\theta_t(\hat{\mathcal{G}}_t)\,\middle|\,\mathcal{K}\right]$$

$$+\mathbb{E}\left[\max_{t\in[2,t^\star]}\frac{4\ln(K)}{p_t^{\min}}+5\ln(K)\sqrt{\max_{t\in[2,t^\star]}\frac{t^\star}{p_t^{\min}}}\,\middle|\,\mathcal{K}\right]\ .$$

$\square$

### D.2   Regret After Round $t^\star$

With Lemma 7 at hand, we can control the regret in the first $t^\star$ rounds. However, we also need to control the regret in the remaining rounds, which we show how to do here. Recall that $\tilde{\mathcal{G}}$ is the graph with edge probabilities $\tilde{p}(j,i)=\frac{1}{t^\star}\sum_{s=1}^{t^\star}\mathbb{I}_{\{(j,i)\in E_s\}}$. At the end of round $t^\star$ we have that $\hat{\mathcal{G}}=\tilde{\mathcal{G}}_{\varepsilon_{t^\star}}$ is an $\varepsilon_{t^\star}$-good approximation of $\mathcal{G}$ with high probability, where

$$\varepsilon_{t^\star}=\frac{60\ln(KT)}{t^\star}\ . \tag{36}$$

We set

$$\varepsilon_{\delta,\sigma}^\star=\underset{\varepsilon\,:\,\mathrm{supp}(\hat{\mathcal{G}}_\varepsilon)\text{ observable}}{\arg\min}\ (\delta_{\mathsf{w}}(\hat{\mathcal{G}}_\varepsilon)\ln(3K^2T^2))^{1/3}T^{2/3}+\sqrt{\sigma(\hat{\mathcal{G}}_\varepsilon)T\ln(3K^2T^2)} \tag{37}$$

and define the corresponding stochastic graph by $\hat{\mathcal{G}}_{\varepsilon^\star_{\delta,\sigma}} = \{\tilde{p}(j,i)\mathbb{I}_{\{\tilde{p}(j,i)\geq\varepsilon^\star_{\delta,\sigma}\}} : i,j \in V\}$. We denote its support by $\hat{G}^\star = \mathrm{supp}(\hat{\mathcal{G}}_{\varepsilon^\star_{\delta,\sigma}})$. We also require any estimated minimum weight weakly dominating set in round $t$, given by

$$D_t^\star = \underset{D\in\mathcal{D}(\hat{G}^\star)}{\arg\min} \sum_{i\in D} \frac{1}{\min_{j\in N^{\mathrm{out}}_{\hat{G}^\star}(i)} \hat{p}_t(i,j)} \ ,$$

where $\mathcal{D}(\hat{G}^\star)$ corresponds to the family of weakly dominating sets in $\hat{G}^\star$. We define

$$\psi_t(i) \propto \begin{cases} \left(\min_{j\in N^{\mathrm{out}}_{\hat{G}^\star}(i)} \hat{p}_t(i,j)\right)^{-1} & \text{for } i \in D_t^\star \\ 0 & \text{for } i \notin D_t^\star \end{cases} \tag{38}$$

to be the exploration distribution in round $t$. Note that this distribution is non-uniform over the weakly dominating set $D_t^\star$. This is because we want to ensure that the loss of each node is observed roughly equally often. If we were to sample uniformly at random, then this would not be possible because the probability that an edge realizes is not necessarily identical for all edges; however, note that the distribution is in fact uniform if the estimated edge probabilities are uniform.

**Lemma 9.** *Suppose that $\hat{\mathcal{G}}$ is an $\varepsilon_{t^\star}$-good approximation of $\mathcal{G}$. For any distribution $u$ over $[K]$, Algorithm 4 guarantees*

$$\mathbb{E}\left[\sum_{t=t^\star+1}^{T}\sum_{i=1}^{K}(\pi_t(i)-u(i))\tilde{\ell}_t(i) \ \middle| \ \mathcal{K}\right]$$

$$\leq 16\delta_{\mathsf{w}}(\hat{\mathcal{G}}_{\varepsilon^\star_{\delta,\sigma}})\ln(3K^2T^2) + 5(\delta_{\mathsf{w}}(\hat{\mathcal{G}}_{\varepsilon^\star_{\delta,\sigma}})\ln(3K^2T^2))^{1/3}T^{2/3} + 4\sqrt{\sigma(\hat{\mathcal{G}}_{\varepsilon^\star_{\delta,\sigma}})T\ln(K)} \ .$$

*Proof.* Consider the set $S = \{i : i \notin N^{\mathrm{in}}_{\hat{G}^\star}(i)\}$ of nodes without a self-loop in $\hat{G}^\star$. Observe that for any node $i \in S$, given $\mathcal{K}$, we have that for some node $k \in D_t^\star$ with $t > t^\star$,

$$\hat{P}_t(i) = \sum_{j\neq i} \hat{p}_t(j,i)\left((1-\gamma)q_t(i) + \gamma\psi_t(i)\right)$$

$$\geq \gamma\hat{p}_t(k,i)\psi_t(k)$$

$$\geq \frac{\gamma}{\sum_{k\in D_t^\star}\left(\min_{j\in N^{\mathrm{out}}_{\hat{G}^\star}(k)} \hat{p}_t(k,j)\right)^{-1}} \ .$$

Observe that $\mathbb{E}[\hat{p}_t(j,i) \mid \mathcal{K}] \geq p(j,i) \geq \frac{1}{2}\tilde{p}(j,i)$ for all edges $(j,i)$ in $\hat{G}^\star$ by definition of $\varepsilon_{t^\star}$-good approximation. This implies that

$$\hat{P}_t(i) \geq \frac{\gamma}{2\delta_{\mathsf{w}}(\hat{\mathcal{G}}_{\varepsilon^\star_{\delta,\sigma}})} \tag{39}$$

holds for any node $i \in S$, conditioning on $\mathcal{K}$. We apply Lemma 12 with $S_t = \emptyset$ to obtain

$$\mathbb{E}\left[\sum_{t=t^\star+1}^{T}\sum_{i=1}^{K}(q_t(i)-u(i))\tilde{\ell}_t(i) \ \middle| \ \mathcal{K}\right] \leq \mathbb{E}\left[\frac{\ln K}{\eta} + \sum_{t=t^\star+1}^{T}\eta\sum_{i=1}^{K}q_t(i)\tilde{\ell}_t(i)^2 \ \middle| \ \mathcal{K}\right]$$

$$\leq \mathbb{E}\left[\frac{\ln K}{\eta} + \sum_{t=t^\star+1}^{T}\eta\sum_{i=1}^{K}\frac{q_t(i)}{\hat{P}_t(i)} \ \middle| \ \mathcal{K}\right] \ ,$$

where we used the fact that $\hat{P}_t(i) - P_t(i) \geq 0$, given $\mathcal{K}$. Recalling Equation (39) and using the fact that $\pi_t(i) \geq \frac{1}{2}q_t(i)$, we can see that

$$\mathbb{E}\left[\sum_{i\in S}\frac{q_t(i)}{\hat{P}_t(i)} \ \middle| \ \mathcal{K}\right] \leq \mathbb{E}\left[\sum_{i\in S}\frac{2\pi_t(i)}{\hat{P}_t(i)} \ \middle| \ \mathcal{K}\right] \leq \mathbb{E}\left[\frac{4\delta_{\mathsf{w}}(\hat{\mathcal{G}}_{\varepsilon^\star_{\delta,\sigma}})}{\gamma} \ \middle| \ \mathcal{K}\right] \ .$$

Considering the sum over $i \notin S$, we have

$$\mathbb{E}\left[\sum_{i \notin S} \frac{q_t(i)}{\hat{P}_t(i)} \,\middle|\, \mathcal{K}\right] \leq \mathbb{E}\left[\sum_{i \notin S} \frac{2\pi_t(i)}{\hat{P}_t(i)} \,\middle|\, \mathcal{K}\right]$$

$$\leq \mathbb{E}\left[\sum_{i \notin S} \frac{2}{\hat{p}_t(i,i)} \,\middle|\, \mathcal{K}\right] \leq \mathbb{E}\left[\sum_{i \notin S} \frac{4}{\tilde{p}(i,i)} \,\middle|\, \mathcal{K}\right] \leq 4\sigma(\hat{\mathcal{G}}_{\varepsilon_{\delta,\sigma}^\star}) \ .$$

Thus, we have that

$$\mathbb{E}\left[\sum_{i=1}^{K} \frac{q_t(i)}{\hat{P}_t(i)} \,\middle|\, \mathcal{K}\right] \leq 4\mathbb{E}\left[\frac{\delta_{\mathsf{w}}(\hat{\mathcal{G}}_{\varepsilon_{\delta,\sigma}^\star})}{\gamma} + \sigma(\hat{\mathcal{G}}_{\varepsilon_{\delta,\sigma}^\star}) \,\middle|\, \mathcal{K}\right] \ , \tag{40}$$

which means that we can use $\eta = \sqrt{\frac{\ln(K)}{4T}}\left(\delta_{\mathsf{w}}(\hat{\mathcal{G}}_{\varepsilon_{\delta,\sigma}^\star})/\gamma + \sigma(\hat{\mathcal{G}}_{\varepsilon_{\delta,\sigma}^\star})\right)^{-1}$ to obtain

$$\mathbb{E}\left[\sum_{t=t^\star+1}^{T} \sum_{i=1}^{K}(\pi_t(i) - u(i))\tilde{\ell}_t(i) \,\middle|\, \mathcal{K}\right] \leq \mathbb{E}\left[\sum_{t=t^\star+1}^{T} \sum_{i=1}^{K}(q_t(i) - u(i))\tilde{\ell}_t(i) \,\middle|\, \mathcal{K}\right] + \gamma T$$

$$\leq \frac{\ln K}{\eta} + 4\eta T\left(\frac{\delta_{\mathsf{w}}(\hat{\mathcal{G}}_{\varepsilon_{\delta,\sigma}^\star})}{\gamma} + \sigma(\hat{\mathcal{G}}_{\varepsilon_{\delta,\sigma}^\star})\right) + \gamma T$$

$$= 4\sqrt{T\ln(K)\left(\frac{\delta_{\mathsf{w}}(\hat{\mathcal{G}}_{\varepsilon_{\delta,\sigma}^\star})}{\gamma} + \sigma(\hat{\mathcal{G}}_{\varepsilon_{\delta,\sigma}^\star})\right)} + \gamma T \ .$$

Now, observe that $T \leq 8\delta_{\mathsf{w}}(\hat{\mathcal{G}}_{\varepsilon_{\delta,\sigma}^\star})\ln(3K^2T^2)$ whenever the algorithm's parameter $\gamma = \min\left\{\left(\delta_{\mathsf{w}}(\hat{\mathcal{G}}_{\varepsilon_{\delta,\sigma}^\star})\ln(3K^2T^2)\right)^{1/3}T^{-1/3}, \frac{1}{2}\right\} = \frac{1}{2}$. As a consequence,

$$\mathbb{E}\left[\sum_{t=t^\star+1}^{T} \sum_{i=1}^{K}(\pi_t(i) - u(i))\tilde{\ell}_t(i) \,\middle|\, \mathcal{K}\right]$$

$$\leq 4\sqrt{T\ln(K)\left(\frac{\delta_{\mathsf{w}}(\hat{\mathcal{G}}_{\varepsilon_{\delta,\sigma}^\star})}{\gamma} + \sigma(\hat{\mathcal{G}}_{\varepsilon_{\delta,\sigma}^\star})\right)} + \gamma T$$

$$\leq 16\delta_{\mathsf{w}}(\hat{\mathcal{G}}_{\varepsilon_{\delta,\sigma}^\star})\ln(3K^2T^2) + 5(\delta_{\mathsf{w}}(\hat{\mathcal{G}}_{\varepsilon_{\delta,\sigma}^\star})\ln(3K^2T^2))^{1/3}T^{2/3} + 4\sqrt{\sigma(\hat{\mathcal{G}}_{\varepsilon_{\delta,\sigma}^\star})T\ln(K)} \ ,$$

which completes the proof. $\qquad\square$

For the following lemma, we will use a simplifying assumption on $T$: we will assume that $T$ is such that

$$2 + \left(37\delta_{\mathsf{w}}(\hat{\mathcal{G}}_{\varepsilon_{\delta,\sigma}^\star}) + 12\sigma(\hat{\mathcal{G}}_{\varepsilon_{\delta,\sigma}^\star})\right)\ln(3K^2T^2)^2 + 12\delta_{\mathsf{w}}(\hat{\mathcal{G}}_{\varepsilon_{\delta,\sigma}^\star})^{2/3}\left(\ln(3K^2T^2)\right)^{5/3}T^{1/3}$$

$$\leq 28\left(\delta_{\mathsf{w}}(\hat{\mathcal{G}}_{\varepsilon_{\delta,\sigma}^\star})\ln(3K^2T^2)\right)^{1/3}T^{2/3} + 29\sqrt{\ln(3K^2T^2)\sigma(\hat{\mathcal{G}}_{\varepsilon_{\delta,\sigma}^\star})T} \ . \tag{41}$$

**Lemma 10.** *Suppose that (41) holds and that $\hat{\mathcal{G}}$ is an $\varepsilon_{t^\star}$-good approximation of $\mathcal{G}$. For any distribution $u$ over $[K]$, Algorithm 4 guarantees*

$$\mathbb{E}\left[\sum_{t=t^\star+1}^{T} \sum_{i=1}^{K}(\pi_t(i) - u(i))\ell_t(i)\right]$$

$$\leq 41\left(\ln(3K^2T^2)\delta_{\mathsf{w}}(\hat{\mathcal{G}}_{\varepsilon_{\delta,\sigma}^\star})\right)^{1/3}T^{2/3} + 41\sqrt{\ln(3K^2T^2)\sigma(\hat{\mathcal{G}}_{\varepsilon_{\delta,\sigma}^\star})T} \ .$$

*We also have that*

$$\mathbb{E}\left[\sum_{t=t^\star+1}^{T} \sum_{i=1}^{K}(\pi_t(i) - u(i))\ell_t(i)\right] \leq$$

$$\min_{\varepsilon \geq 2\varepsilon_{t^\star}} \left\{82\left(\ln(3K^2T^2)\delta_{\mathsf{w}}(\mathcal{G}_\varepsilon)\right)^{1/3}T^{2/3} + 82\sqrt{\ln(3K^2T^2)\sigma(\mathcal{G}_\varepsilon)T} \ : \ \mathrm{supp}(\mathcal{G}_\varepsilon) \text{ observable}\right\} \ .$$

*Proof.* Following the proof of Lemma 6, we can see that

$$\mathbb{E}\left[\sum_{t=t^\star+1}^{T}\sum_{i=1}^{K}(\pi_t(i)-u(i))\ell_t(i)\right] \le 2 + \sum_{t=t^\star+1}^{T}\mathbb{E}\left[\frac{6\ln(3K^2T^2)}{t-1}\sum_{i=1}^{K}\frac{\pi_t(i)\bar{\pi}_t(i)}{\hat{P}_t(i)}\,\middle|\,\mathcal{K}\right]$$

$$+\mathbb{E}\left[\sum_{t=t^\star+1}^{T}2\sqrt{2\frac{\ln(3K^2T^2)}{t-1}}\sqrt{\sum_{i=1}^{K}\frac{\pi_t(i)\bar{\pi}_t(i)}{\hat{P}_t(i)}}\,\middle|\,\mathcal{K}\right]+\mathbb{E}\left[\sum_{t=t^\star+1}^{T}\sum_{i=1}^{K}(\pi_t(i)-u(i))\tilde{\ell}_t(i)\,\middle|\,\mathcal{K}\right].$$

Now, using the same reasoning that led to Equation (40), we have that

$$\mathbb{E}\left[\sum_{t=t^\star+1}^{T}\sum_{i=1}^{K}(\pi_t(i)-u(i))\ell_t(i)\right] \le 2 + \sum_{t=t^\star+1}^{T}\mathbb{E}\left[\frac{12\ln(3K^2T^2)}{t-1}\left(\frac{\delta_{\mathsf{w}}(\hat{\mathcal{G}}_{\varepsilon^\star_{\delta,\sigma}})}{\gamma}+\sigma(\hat{\mathcal{G}}_{\varepsilon^\star_{\delta,\sigma}})\right)\,\middle|\,\mathcal{K}\right]$$

$$+\mathbb{E}\left[\sum_{t=t^\star+1}^{T}4\sqrt{\frac{\ln(3K^2T^2)}{t-1}}\sqrt{\frac{\delta_{\mathsf{w}}(\hat{\mathcal{G}}_{\varepsilon^\star_{\delta,\sigma}})}{\gamma}+\sigma(\hat{\mathcal{G}}_{\varepsilon^\star_{\delta,\sigma}})}\,\middle|\,\mathcal{K}\right]$$

$$+\mathbb{E}\left[\sum_{t=t^\star+1}^{T}\sum_{i=1}^{K}(\pi_t(i)-u(i))\tilde{\ell}_t(i)\,\middle|\,\mathcal{K}\right]$$

$$\le 2 + \mathbb{E}\left[12\ln(3K^2T^2)^2\left(\frac{\delta_{\mathsf{w}}(\hat{\mathcal{G}}_{\varepsilon^\star_{\delta,\sigma}})}{\gamma}+\sigma(\hat{\mathcal{G}}_{\varepsilon^\star_{\delta,\sigma}})\right)\,\middle|\,\mathcal{K}\right]$$

$$+\mathbb{E}\left[8\sqrt{T\ln(3K^2T^2)\left(\frac{\delta_{\mathsf{w}}(\hat{\mathcal{G}}_{\varepsilon^\star_{\delta,\sigma}})}{\gamma}+\sigma(\hat{\mathcal{G}}_{\varepsilon^\star_{\delta,\sigma}})\right)}\,\middle|\,\mathcal{K}\right]$$

$$+\mathbb{E}\left[\sum_{t=t^\star+1}^{T}\sum_{i=1}^{K}(\pi_t(i)-u(i))\tilde{\ell}_t(i)\,\middle|\,\mathcal{K}\right],$$

where we used that $\sum_{t=1}^{T}\frac{1}{\sqrt{t}}\le 2\sqrt{T}$ and $\sum_{t=2}^{T}\frac{1}{t-1}\le 1+\ln(T)\le\ln(3K^2T^2)$ for $K,T\ge 2$.
Following the final steps in the proof of Lemma 9, we can show that

$$\mathbb{E}\left[8\sqrt{T\ln(3K^2T^2)\left(\frac{\delta_{\mathsf{w}}(\hat{\mathcal{G}}_{\varepsilon^\star_{\delta,\sigma}})}{\gamma}+\sigma(\hat{\mathcal{G}}_{\varepsilon^\star_{\delta,\sigma}})\right)}\,\middle|\,\mathcal{K}\right]$$

$$\le 32\delta_{\mathsf{w}}(\hat{\mathcal{G}}_{\varepsilon^\star_{\delta,\sigma}})\ln(3K^2T^2)+8T^{2/3}\left(\delta_{\mathsf{w}}(\hat{\mathcal{G}}_{\varepsilon^\star_{\delta,\sigma}})\ln(3K^2T^2)\right)^{1/3}+8\sqrt{T\sigma(\hat{\mathcal{G}}_{\varepsilon^\star_{\delta,\sigma}})\ln(3K^2T^2)}.$$

Hence, by applying Lemma 9, we obtain that

$$\mathbb{E}\left[\sum_{t=t^\star+1}^{T}\sum_{i=1}^{K}(\pi_t(i)-u(i))\tilde{\ell}_t(i)\,\middle|\,\mathcal{K}\right]$$

$$\le 16\delta_{\mathsf{w}}(\hat{\mathcal{G}}_{\varepsilon^\star_{\delta,\sigma}})\ln(3K^2T^2)+5T^{2/3}(\delta_{\mathsf{w}}(\hat{\mathcal{G}}_{\varepsilon^\star_{\delta,\sigma}})\ln(3K^2T^2))^{1/3}+4\sqrt{T\sigma(\hat{\mathcal{G}}_{\varepsilon^\star_{\delta,\sigma}})\ln(K)}.$$

Finally, by definition of $\gamma$ we notice that

$$12\ln(3K^2T^2)^2\frac{\delta_{\mathsf{w}}(\hat{\mathcal{G}}_{\varepsilon^\star_{\delta,\sigma}})}{\gamma}\le 24\delta_{\mathsf{w}}(\hat{\mathcal{G}}_{\varepsilon^\star_{\delta,\sigma}})\ln(3K^2T^2)^2+12T^{1/3}\delta_{\mathsf{w}}(\hat{\mathcal{G}}_{\varepsilon^\star_{\delta,\sigma}})^{2/3}\left(\ln(3K^2T^2)\right)^{5/3}.$$

Thus, combining the above we obtain

$$\mathbb{E}\left[\sum_{t=t^\star+1}^{T}\sum_{i=1}^{K}(\pi_t(i)-u(i))\ell_t(i)\right]$$

$$\le 2 + 37\delta_{\mathsf{w}}(\hat{\mathcal{G}}_{\varepsilon^\star_{\delta,\sigma}})\ln(3K^2T^2)^2+12T^{1/3}\delta_{\mathsf{w}}(\hat{\mathcal{G}}_{\varepsilon^\star_{\delta,\sigma}})^{2/3}\left(\ln(3K^2T^2)\right)^{5/3}$$

$$+13T^{2/3}\left(\delta_{\mathsf{w}}(\hat{\mathcal{G}}_{\varepsilon^\star_{\delta,\sigma}})\ln(3K^2T^2)\right)^{1/3}+12\sqrt{T\sigma(\hat{\mathcal{G}}_{\varepsilon^\star_{\delta,\sigma}})\ln(3K^2T^2)}$$

$$+12\sigma(\hat{\mathcal{G}}_{\varepsilon^\star_{\delta,\sigma}})\ln(3K^2T^2)^2.$$

Since we assumed that (41) holds, we can show that

$$\mathbb{E}\left[\sum_{t=t^\star+1}^{T}\sum_{i=1}^{K}(\pi_t(i)-u(i))\ell_t(i)\right]$$

$$\leq 41T^{2/3}\left(\delta_{\mathsf{w}}(\hat{\mathcal{G}}_{\varepsilon_{\delta,\sigma}^\star})\ln(3K^2T^2)\right)^{1/3} + 41\sqrt{T\sigma(\hat{\mathcal{G}}_{\varepsilon_{\delta,\sigma}^\star})\ln(3K^2T^2)}\ ,$$

which is the first result in the statement. For the second result, recall that $\varepsilon_{\delta,\sigma}^\star$ is the minimizer of the above bound by its definition in (37). Since $\hat{\mathcal{G}}$ is an $\varepsilon_{t^\star}$-good approximation of $\mathcal{G}$, we conclude that

$$\mathbb{E}\left[\sum_{t=t^\star+1}^{T}\sum_{i=1}^{K}(\pi_t(i)-u(i))\ell_t(i)\right] \leq$$
$$\min_{\varepsilon\geq 2\varepsilon_{t^\star}}\left\{82T^{2/3}\left(\ln(3K^2T^2)\delta_{\mathsf{w}}(\mathcal{G}_\varepsilon)\right)^{1/3} + 82\sqrt{\ln(3K^2T^2)\sigma(\mathcal{G}_\varepsilon)T}\ :\ \mathrm{supp}(\mathcal{G}_\varepsilon)\ \mathrm{observable}\right\}\ .$$

$\square$

### D.3 Regret After $T$ Rounds

We now have all the intermediate results we need to prove the overall regret bound of Algorithm 4.

**Theorem 11.** *Suppose that* (41) *holds. Then, for any distribution $u$ over $[K]$, Algorithm 4 satisfies*

$$\mathbb{E}\left[\sum_{t=1}^{T}\sum_{i=1}^{K}(\pi_t(i)-u(i))\ell_t(i)\right] \leq \min\Bigg\{T\ ,$$

$$6+2\min_{\varepsilon\,:\,\mathrm{supp}(\mathcal{G}_\varepsilon)\ \mathrm{strongly\ observable}}\Bigg\{198\alpha_{\mathsf{w}}(\mathcal{G}_\varepsilon)(\ln(2K^3T^2))^3$$

$$+\left(12\ln(K)+4\sqrt{2\ln(3K^2T^2)}\right)\sqrt{18t^\star\alpha_{\mathsf{w}}(\mathcal{G}_\varepsilon)\ln(2K^3T^2)}\Bigg\}\ ,$$

$$4+164\ln(3K^2T^2)\min_{\varepsilon\,:\,\mathrm{supp}(\mathcal{G}_\varepsilon)\ \mathrm{observable}}\left((\delta_{\mathsf{w}}(\mathcal{G}_\varepsilon))^{1/3}T^{2/3}+\sqrt{\sigma(\mathcal{G}_\varepsilon)T}\right)\Bigg\}\ .$$

*Proof.* Let us recall that in Equations (27) and (28) we define

$$\Psi_{t^\star}=\min\Bigg\{t^\star,\ 2+11(\ln(3K^2T^2))^2\max_{t\in[2,t^\star]}\theta_t(\hat{\mathcal{G}}_t)$$

$$+\left(12\ln(K)+4\sqrt{2\ln(3K^2T^2)}\right)\sqrt{t^\star\max_{t\in[2,t^\star]}\theta_t(\hat{\mathcal{G}}_t)}\Bigg\}$$

and

$$\Lambda_{t^\star}=41\left(\ln(3K^2T^2)\delta_{\mathsf{w}}(\hat{\mathcal{G}}_{\varepsilon_{\delta,\sigma}^\star})\right)^{1/3}T^{2/3}+41\sqrt{\ln(3K^2T^2)\sigma(\hat{\mathcal{G}}_{\varepsilon_{\delta,\sigma}^\star})T}\ .$$

Denote by $\mathcal{E}$ the event that $\tilde{\mathcal{G}}_{\varepsilon_t}=\{\tilde{p}_t(j,i)\mathbb{I}_{\{\tilde{p}_t(j,i)\geq 60\ln(KT)/t\}}\ :\ i,j\in V\}$ is a $\varepsilon_t$-good approximation of $\mathcal{G}$ with $\varepsilon_t=60\ln(KT)/t$ for all $t\leq T$. By Lemma 14, we have that $\mathcal{E}$

occurs with probability at least $1 - \frac{1}{T}$ and thus, for any $t^\star \in [1, T]$, we have that

$$\mathbb{E}\left[\sum_{t=1}^{T}\sum_{i=1}^{K}(\pi_t(i) - u(i))\ell_t(i)\right]$$

$$\leq 1 + \mathbb{E}\left[\sum_{t=1}^{T}\sum_{i=1}^{K}(\pi_t(i) - u(i))\ell_t(i) \,\middle|\, \mathcal{E}\right]$$

$$= 1 + \mathbb{E}\left[\sum_{t=1}^{t^\star}\sum_{i=1}^{K}(\pi_t(i) - u(i))\ell_t(i) \,\middle|\, \mathcal{E}\right] + \mathbb{E}\left[\sum_{t=t^\star+1}^{T}\sum_{i=1}^{K}(\pi_t(i) - u(i))\ell_t(i) \,\middle|\, \mathcal{E}\right]$$

$$\leq 1 + \mathbb{E}\left[\Psi_{t^\star} + \Lambda_{t^\star} \mid \mathcal{K}, \mathcal{E}\right] \ ,$$

where the last inequality is due to Lemmas 7 and 10. We now consider two cases depending on whether Algorithm 4 commits to the weakly observable regret regime at any time step or it never does so. In the first case, say Equation (26) never holds for any $t \in [2, T]$. We consequently have that

$$\mathbb{E}\left[\sum_{t=1}^{T}\sum_{i=1}^{K}(\pi_t(i) - u(i))\ell_t(i)\right] \leq 1 + \mathbb{E}\left[\min\left\{\Psi_{t^\star}, \Lambda_{t^\star}\right\} \mid \mathcal{K}, \mathcal{E}\right] \ .$$

We first try to upper bound the conditional expectation of $\Lambda_{t^\star}$. By definition of $\varepsilon$-good approximation of $\mathcal{G}$, we have

$$\mathbb{E}\left[\Lambda_{t^\star} \mid \mathcal{K}, \mathcal{E}\right] = \mathbb{E}\left[\min_{\varepsilon \in [0,1]}\left\{41\left(\ln(3K^2T^2)\delta_{\mathsf{w}}((\hat{\mathcal{G}}_{t^\star})_\varepsilon)\right)^{1/3}T^{2/3}\right.\right.$$

$$\left.\left. + 41\sqrt{\ln(3K^2T^2)\sigma((\hat{\mathcal{G}}_{t^\star})_\varepsilon)T} \ : \ \mathrm{supp}((\hat{\mathcal{G}}_{t^\star})_\varepsilon) \text{ observable}\right\} \,\middle|\, \mathcal{K}, \mathcal{E}\right]$$

$$\leq 2\mathbb{E}\left[\min_{\varepsilon \in [2\varepsilon_{t^\star}, 1]}\left\{41\left(\ln(3K^2T^2)\delta_{\mathsf{w}}((\hat{\mathcal{G}}_{t^\star})_\varepsilon)\right)^{1/3}T^{2/3}\right.\right.$$

$$\left.\left. + 41\sqrt{\ln(3K^2T^2)\sigma((\hat{\mathcal{G}}_{t^\star})_\varepsilon)T} : \mathrm{supp}(\mathcal{G}_\varepsilon) \text{ observable}\right\} \,\middle|\, \mathcal{K}, \mathcal{E}\right] \ .$$

To cover the remaining thresholds in $[0, 2\varepsilon_{t^\star})$, we define $\varepsilon_\Lambda^\star = \max \mathcal{Q}$ as the largest threshold $\varepsilon$ that minimizes

$$\mathcal{Q} = \underset{\varepsilon \in [0,1]}{\arg\min}\left\{41\left(\ln(3K^2T^2)\delta_{\mathsf{w}}((\hat{\mathcal{G}}_{t^\star})_\varepsilon)\right)^{1/3}T^{2/3}\right.$$

$$\left. + 41\sqrt{\ln(3K^2T^2)\sigma((\hat{\mathcal{G}}_{t^\star})_\varepsilon)T} \ : \ \mathrm{supp}(\mathcal{G}_\varepsilon) \text{ observable}\right\} \ .$$

If $\varepsilon_\Lambda^\star < 2\varepsilon_{t^\star}$, meaning that $\varepsilon_\Lambda^\star$ as well as the other thresholds in $\mathcal{Q}$ do not belong to the already covered interval $[2\varepsilon_{t^\star}, 1]$, then $t^\star < \frac{120\ln(KT)}{\varepsilon_\Lambda^\star} = 120\ln(KT)t_{\varepsilon_\Lambda^\star}$ with $t_{\varepsilon_\Lambda^\star} = 1/\varepsilon_\Lambda^\star$. Thus, we must have that

$$t^\star \leq 120\ln(KT)\left(\left(\delta_{\mathsf{w}}(\mathcal{G}_{\varepsilon_\Lambda^\star})\right)^{1/3}t_{\varepsilon_\Lambda^\star}^{2/3} + \sqrt{\sigma(\mathcal{G}_{\varepsilon_\Lambda^\star})t_{\varepsilon_\Lambda^\star}}\right)$$

$$\leq \min_{\varepsilon \in [0,1]}\left\{120\ln(KT)\left((\delta_{\mathsf{w}}(\mathcal{G}_\varepsilon))^{1/3}T^{2/3} + \sqrt{\sigma(\mathcal{G}_\varepsilon)T}\right) \ : \ \mathrm{supp}(\mathcal{G}_\varepsilon) \text{ observable}\right\} \ ,$$

where the first inequality is due to the fact that $\delta_{\mathsf{w}}(\mathcal{G}_{\varepsilon_\Lambda^\star}) \geq t_{\varepsilon_\Lambda^\star}$ or $\sigma(\mathcal{G}_{\varepsilon_\Lambda^\star}) \geq t_{\varepsilon_\Lambda^\star}$ or both are true because either $p(i, i) = \varepsilon_\Lambda^\star$ for some $i$ such that $i \in N_{\mathrm{supp}(\mathcal{G}_{\varepsilon_\Lambda^\star})}^{\mathrm{in}}(i)$ or one of the minimum outgoing edge probabilities for a vertex in some minimum weight weakly dominating set is equal to $\varepsilon_\Lambda^\star$.

On the other hand, we also need to upper bound the conditional expectation of $\Psi_{t^\star}$. By Lemma 11 and recalling the definition of $\alpha_{\mathsf{w}}$ from Section 5, we have that

$$\mathbb{E}\left[\max_{t \in [2,t^\star]} \theta_t(\hat{\mathcal{G}}_t) \;\middle|\; \mathcal{K}\right] \leq \mathbb{E}\left[\min_{\varepsilon\,:\,\mathrm{supp}(\mathcal{G}_\varepsilon)\ \mathrm{strongly\ observable}} 18\alpha_{\mathsf{w}}(\mathcal{G}_\varepsilon)\ln(2K^3T^2) \;\middle|\; \mathcal{K}\right] \ .$$

and thus

$$2 + \mathbb{E}\left[11(\ln(3K^2T^2))^2 \max_{t \in [2,t^\star]} \theta_t(\hat{\mathcal{G}}_t) \;\middle|\; \mathcal{K}, \mathcal{E}\right]$$

$$+ \mathbb{E}\left[\left(12\ln(K) + 4\sqrt{2\ln(3K^2T^2)}\right)\sqrt{t^\star \max_{t \in [2,t^\star]} \theta_t(\hat{\mathcal{G}}_t)} \;\middle|\; \mathcal{K}, \mathcal{E}\right]$$

$$\leq 2 + \min_{\varepsilon\,:\,\mathrm{supp}(\mathcal{G}_\varepsilon)\ \mathrm{strongly\ observable}} \left\{ 198\alpha_{\mathsf{w}}(\mathcal{G}_\varepsilon)(\ln(2K^3T^2))^3 \right.$$

$$\left. + \left(12\ln(K) + 4\sqrt{2\ln(3K^2T^2)}\right)\sqrt{18t^\star \alpha_{\mathsf{w}}(\mathcal{G}_\varepsilon)\ln(2K^3T^2)} \right\} \ .$$

Since $120\ln(KT) \leq 82\ln(3K^2T^2)$, we can combine the above to obtain

$$\mathbb{E}\left[\sum_{t=1}^{T}\sum_{i=1}^{K}(\pi_t(i) - u(i))\ell_t(i)\right] \leq \min\Bigg\{ T,$$

$$3 + \min_{\varepsilon}\bigg\{ 198\alpha_{\mathsf{w}}(\mathcal{G}_\varepsilon)(\ln(2K^3T^2))^3 +$$

$$\left(12\ln(K) + 4\sqrt{2\ln(3K^2T^2)}\right)\sqrt{18t^\star \alpha_{\mathsf{w}}(\mathcal{G}_\varepsilon)\ln(2K^3T^2)} \ :\ \mathrm{supp}(\mathcal{G}_\varepsilon)\ \mathrm{strongly\ observable}\bigg\},$$

$$1 + \min_{\varepsilon}\bigg\{ 82\ln(3K^2T^2)\left((\delta_{\mathsf{w}}(\mathcal{G}_\varepsilon))^{1/3}\,T^{2/3} + \sqrt{\sigma(\mathcal{G}_\varepsilon)T}\right) \ :\ \mathrm{supp}(\mathcal{G}_\varepsilon)\ \mathrm{observable}\bigg\}\Bigg\} \ .$$

In the second case, $t^\star$ is the first round in which (26) holds. Therefore, we must have

$$\mathbb{E}\left[\sum_{t=1}^{T}\sum_{i=1}^{K}(\pi_t(i) - u(i))\ell_t(i)\right] \leq 1 + 2\mathbb{E}\left[\Psi_{t^\star} \mid \mathcal{K}, \mathcal{E}\right]$$

and

$$\mathbb{E}\left[\sum_{t=1}^{T}\sum_{i=1}^{K}(\pi_t(i) - u(i))\ell_t(i)\right]$$

$$\leq 1 + \mathbb{E}\left[\sum_{t=1}^{t^\star-1}\sum_{i=1}^{K}(\pi_t(i) - u(i))\ell_t(i) \;\middle|\; \mathcal{E}\right] + 1 + \mathbb{E}\left[\sum_{t=t^\star+1}^{T}\sum_{i=1}^{K}(\pi_t(i) - u(i))\ell_t(i) \;\middle|\; \mathcal{E}\right]$$

$$\leq \mathbb{E}\left[\Psi_{t^\star-1} + \Lambda_{t^\star} \mid \mathcal{K}, \mathcal{E}\right] + 2$$

$$\leq \mathbb{E}\left[\Lambda_{t^\star-1} + \Lambda_{t^\star} \mid \mathcal{K}, \mathcal{E}\right] + 2 \ ,$$

which combined give us

$$\mathbb{E}\left[\sum_{t=1}^{T}\sum_{i=1}^{K}(\pi_t(i) - u(i))\ell_t(i)\right] \leq 1 + \mathbb{E}\left[\min\left\{\Lambda_{t^\star-1} + \Lambda_{t^\star} + 1, 2\Psi_{t^\star}\right\} \mid \mathcal{K}, \mathcal{E}\right] \ .$$

Following the proof of the bound in the case where (26) never holds for any $t \in [2, T]$, we can see that

$$\mathbb{E}\left[\sum_{t=1}^{T}\sum_{i=1}^{K}(\pi_t(i) - u(i))\ell_t(i)\right] \leq \min\Big\{T \,,$$

$$6 + 2 \min_{\varepsilon \,:\, \text{supp}(\mathcal{G}_\varepsilon) \text{ strongly observable}} \Big\{198\alpha_{\mathsf{w}}(\mathcal{G}_\varepsilon)(\ln(2K^3T^2))^3$$

$$+ \Big(12\ln(K) + 4\sqrt{2\ln(3K^2T^2)}\Big)\sqrt{18t^\star\alpha_{\mathsf{w}}(\mathcal{G}_\varepsilon)\ln(2K^3T^2)}\Big\} \,,$$

$$4 + 164\ln(3K^2T^2) \min_{\varepsilon \,:\, \text{supp}(\mathcal{G}_\varepsilon) \text{ observable}} \Big((\delta_{\mathsf{w}}(\mathcal{G}_\varepsilon))^{1/3}\,T^{2/3} + \sqrt{\sigma(\mathcal{G}_\varepsilon)T}\Big)\Big\} \,,$$

which completes the proof. $\qquad\square$

### D.4 Auxiliary Lemmas for OTCG

In this section, we prove some results that are useful in the above regret analysis of OTCG (Algorithm 4). Recall that $\mathcal{S}$ is the family of strongly observable graphs over vertices $V = [K]$.

**Lemma 11.** *Suppose that there exists a threshold $\varepsilon$ such that $\text{supp}(\mathcal{G}_\varepsilon) \in \mathcal{S}$. Then, we have that*

$$\mathbb{E}\left[\max_{t \in [2, t^\star]} \min_{\varepsilon \,:\, \text{supp}((\hat{\mathcal{G}}_t^{\text{UCB}})_\varepsilon) \in \mathcal{S}} \theta_t((\hat{\mathcal{G}}_t^{\text{UCB}})_\varepsilon) \,\Big|\, \mathcal{K}\right] \leq \mathbb{E}\left[\min_{\varepsilon \,:\, \text{supp}(\mathcal{G}_\varepsilon) \in \mathcal{S}} 18\alpha_{\mathsf{w}}(\mathcal{G}_\varepsilon)\ln(2K^3T^2) \,\Big|\, \mathcal{K}\right]$$

*Proof.* Let us recall the definition of $\theta_t$:

$$\theta_t((\hat{\mathcal{G}}_t^{\text{UCB}})_\varepsilon) = \frac{2}{\min_i \min_{j \in N^{\text{in}}_{\text{supp}((\hat{\mathcal{G}}_t^{\text{UCB}})_\varepsilon)}(i)} p(j, i)} + \sum_{i \in N^{\text{in}}_{\text{supp}((\hat{\mathcal{G}}_t^{\text{UCB}})_\varepsilon)}(i)} \frac{2\pi_t(i)}{P_t(i, (\hat{\mathcal{G}}_t^{\text{UCB}})_\varepsilon)} \,.$$

By definition of the weighted independence number (see Appendix E for further details), we have that

$$\frac{2}{\min_i \min_{j \in N^{\text{in}}_{\text{supp}((\hat{\mathcal{G}}_t^{\text{UCB}})_\varepsilon)}(i)} p(j, i)} \leq 2\alpha_{\mathsf{w}}((\hat{\mathcal{G}}_t^{\text{UCB}})_\varepsilon) \,.$$

By Lemma 17, we have that

$$2 \sum_{i \in N^{\text{in}}_{\text{supp}((\hat{\mathcal{G}}_t^{\text{UCB}})_\varepsilon)}(i)} \frac{\pi_t(i)}{P_t(i, (\hat{\mathcal{G}}_t^{\text{UCB}})_\varepsilon)} \leq 16\alpha_{\mathsf{w}}((\hat{\mathcal{G}}_t^{\text{UCB}})_\varepsilon)\ln(2K^3T^2) \,,$$

where we used that $\gamma_t\psi_t(i) \geq \frac{1}{KT}$ and $\hat{p}_t(j, i) \geq \frac{1}{T}$.

Given $\mathcal{K}$, we have that $\hat{p}_t(j, i) \geq p(j, i)$ and thus it holds that

$$\mathbb{E}\left[\max_{t \in [2, t^\star]} \min_{\varepsilon \,:\, \text{supp}((\hat{\mathcal{G}}_t^{\text{UCB}})_\varepsilon) \in \mathcal{S}} \theta_t((\hat{\mathcal{G}}_t^{\text{UCB}})_\varepsilon) \,\Big|\, \mathcal{K}\right]$$

$$\leq \mathbb{E}\left[\max_{t \in [2, t^\star]} \min_{\varepsilon \,:\, \text{supp}((\hat{\mathcal{G}}_t^{\text{UCB}})_\varepsilon) \in \mathcal{S}} 18\alpha_{\mathsf{w}}((\hat{\mathcal{G}}_t^{\text{UCB}})_\varepsilon)\ln(2K^3T^2) \,\Big|\, \mathcal{K}\right]$$

$$\leq \mathbb{E}\left[\min_{\varepsilon \,:\, \text{supp}(\mathcal{G}_\varepsilon) \in \mathcal{S}} 18\alpha_{\mathsf{w}}(\mathcal{G}_\varepsilon)\ln(2K^3T^2) \,\Big|\, \mathcal{K}\right] \,.$$

$\qquad\square$

The following result is a variant of the bound in Alon et al. [2015, Lemma 4] with a decreasing learning rate.

**Lemma 12.** *Let $q_1, \ldots, q_T$ be the probability vectors defined by $q_t(i) \propto \exp(-\eta_{t-1} \sum_{s=1}^{t-1} \ell_s(i))$ for a sequence of loss functions $\ell_1, \ldots, \ell_T$ such that $\ell_t(i) \geq 0$ for all $t$ and $i$. Let $\eta_0 = \eta_1 \geq \ldots \geq \eta_T$. For each $t$, let $S_t$ be a subset of $[K]$ such that $\eta_{t-1}\ell_t(i) \leq 1$ for all $i \in S_t$. Then, for any distribution $u$ it holds that*

$$\sum_{t=1}^{T}\sum_{i=1}^{K}(q_t(i) - u(i))\ell_t(i) \leq \frac{\ln(K)}{\eta_T} + \sum_{t=1}^{T}\eta_{t-1}\left(\sum_{i \in S_t} q_t(i)(1 - q_t(i))\ell_t(i)^2 + \sum_{i \notin S_t} q_t(i)\ell_t(i)^2\right).$$

*Proof.* The proof follows from a minor adaptation of the proof of Alon et al. [2015, Lemma 4]. We start from Van der Hoeven et al. [2018, Lemma 1]:

$$\sum_{t=1}^{T}\sum_{i=1}^{K}(q_t(i) - u(i))\ell_t(i)$$
$$\leq \frac{\ln(K)}{\eta_T} + \sum_{t=1}^{T}\left(\sum_{i=1}^{K} q_t(i)\ell_t(i) + \frac{1}{\eta_{t-1}}\ln\left(\sum_{i=1}^{K} q_t(i)\exp(-\eta_{t-1}\ell_t(i))\right)\right). \tag{42}$$

Now, since $\ell_t(i) \geq 0$ we may use $\exp(x) \leq 1 + x + x^2$ for $x \leq 1$ and $\ln(1 - x) \leq -x$ for all $x < 1$ to show that

$$\frac{1}{\eta_{t-1}}\ln\left(\sum_{i=1}^{K} q_t(i)\exp(-\eta_{t-1}\ell_t(i))\right) \leq \frac{1}{\eta_{t-1}}\ln\left(1 - \sum_{i=1}^{K} q_t(i)(\eta_{t-1}\ell_t(i) - \eta_{t-1}^2\ell_t(i)^2)\right)$$
$$\leq -\sum_{i=1}^{K} q_t(i)(\ell_t(i) - \eta_{t-1}\ell_t(i)^2) .$$

Combined with equation (42), this gives us

$$\sum_{t=1}^{T}\sum_{i=1}^{K}(q_t(i) - u(i))\ell_t(i) \leq \frac{\ln(K)}{\eta_T} + \sum_{t=1}^{T}\sum_{i=1}^{K}\eta_{t-1}q_t(i)\ell_t(i)^2 .$$

We define $\bar{\ell}_t = \sum_{i \in S_t} q_t(i)\ell_t(i)$. Since $\ell_t(i) \geq 0$ we have that $\eta_{t-1}(\ell_t(i) - \bar{\ell}_t) \geq -1$ by construction. Since adding the same $\bar{\ell}_t$ to each $\ell_t(i)$ on the r.h.s. of equation (42) does not influence the regret we have

$$\sum_{t=1}^{T}\sum_{i=1}^{K}(q_t(i) - u(i))\ell_t(i) \leq \frac{\ln(K)}{\eta_T} + \sum_{t=1}^{T}\sum_{i=1}^{K}\eta_{t-1}q_t(i)(\ell_t(i) - \bar{\ell}_t)^2 .$$

To complete the proof we follow the proof of Alon et al. [2015, Lemma 4], which gives us

$$\sum_{t=1}^{T}\sum_{i=1}^{K}(q_t(i) - u(i))\ell_t(i) \leq \frac{\ln(K)}{\eta_T} + \sum_{t=1}^{T}\eta_{t-1}\left(\sum_{i \in S_t} q_t(i)(1 - q_t(i))\ell_t(i)^2 + \sum_{i \notin S_t} q_t(i)\ell_t(i)^2\right).$$
$\square$

**Lemma 13.** *Let $\xi_t(i) = \sum_{j \in N_{\hat{G}_t}^{\text{in}}(i)} \pi_t(i)(\hat{p}_t(j,i) - p(j,i))$. In any round $t$, we have that*

$$\sum_{i=1}^{K}(\pi_t(i) - u(i))\hat{\ell}_t(i)$$
$$= \sum_{i=1}^{K}(\pi_t(i) - u(i))\tilde{\ell}_t(i) + \sum_{i=1}^{K}(\pi_t(i) - u(i))\xi_t(i)\frac{\mathbb{I}_{\{i \in N_{G_t}^{\text{out}}(I_t)\} \cap \{i \in N_{\hat{G}_t}^{\text{out}}(I_t)\}}\ell_t(i)}{P_t(i)\hat{P}_t(i)} .$$

*Proof.* Let $X_t = \mathbb{I}_{\{i \in N_{G_t}^{\text{out}}(I_t)\} \cap \{i \in N_{\hat{G}_t}^{\text{out}}(I_t)\}}$ and denote by

$$\xi_t(i) = \hat{P}_t(i) - P_t(i) = \sum_{j \in N_{\hat{G}_t}^{\text{in}}(i)} \pi_t(i)(\hat{p}_t(j,i) - p(j,i)) .$$

We have that

$$\tilde{\ell}_t(i) = \frac{X_t \ell_t(i)}{\hat{P}_t(i)} = \frac{X_t \ell_t(i)}{P_t(i) + \xi_t(i)}$$

$$= \frac{X_t \ell_t(i)(P_t(i) + \xi_t(i))}{P_t(i)(P_t(i) + \xi_t(i))} - \xi_t(i)\frac{X_t \ell_t(i)}{P_t(i)(P_t(i) + \xi_t(i))}$$

$$= \frac{X_t \ell_t(i)}{P_t(i)} - \xi_t(i)\frac{X_t \ell_t(i)}{P_t(i)(P_t(i) + \xi_t(i))}$$

$$= \hat{\ell}_t(i) - \xi_t(i)\frac{X_t \ell_t(i)}{P_t(i)\hat{P}_t(i)} \quad .$$

Therefore, for any distribution $u$ we have that

$$\sum_{i=1}^{K}(\pi_t(i) - u(i))\hat{\ell}_t(i) = \sum_{i=1}^{K}(\pi_t(i) - u(i))\tilde{\ell}_t(i) + \sum_{i=1}^{K}(\pi_t(i) - u(i))\xi_t(i)\frac{X_t \ell_t(i)}{P_t(i)\hat{P}_t(i)} \quad ,$$

which completes the proof. □

**Lemma 14.** *Let* $\tilde{\mathcal{G}}_{\varepsilon_t} = \{\tilde{p}_t(j,i)\mathbb{I}_{\{\tilde{p}_t(j,i)\geq\varepsilon_t\}} : i,j \in V\}$ *and* $\varepsilon_t = 60\ln(KT)/t$ *for all* $t \in [2,T]$. *Then, with probability at least* $1 - 1/T$, $\tilde{\mathcal{G}}_{\varepsilon_t}$ *is an* $\varepsilon_t$-*good approximation of* $\mathcal{G}$ *for all* $t \in [2,T]$.

*Proof.* Let $E_t^+ = \{(i,j) \in V^2 : p(i,j) \geq 2\varepsilon_t\}$ and $E_t^- = \{(i,j) \in V^2 : p(i,j) < \varepsilon_t/2\}$ be the two sets of edges as defined in the proof of Theorem 2. We let $\mathcal{E}_{(i,j)}^t = \{\tilde{p}_t(i,j) \geq \varepsilon_t\}$ and $\mathcal{F}_{(i,j)}^t = \{|\tilde{p}_t(i,j) - p(i,j)| \leq p(i,j)/2\}$, for all $(i,j) \in V^2$ and all $t \in [2,T]$, be the events as similarly denoted in that same proof. We consequently define the events $\mathcal{E}$, $\mathcal{F}$, and $\mathcal{C}$ as

$$\mathcal{E} = \bigcap_{t=1}^{T} \bigcap_{(i,j)\in E_t^+} \mathcal{E}_{(i,j)}^t \quad, \qquad \mathcal{F} = \bigcap_{t=1}^{T} \bigcap_{(i,j)\notin E_t^-} \mathcal{F}_{(i,j)}^t \quad, \qquad \mathcal{C} = \bigcap_{t=1}^{T} \bigcap_{(i,j)\in E_t^-} \overline{\mathcal{E}}_{(i,j)}^t \quad.$$

The following steps hold for all $K \geq 2$ and all $T \geq 2$.

We begin by observing that $\mathbb{P}\left(\tilde{p}_t(i,j) < \varepsilon_t\right) \leq \exp(-t\varepsilon_t/4) \leq 1/(4K^2T^2)$ for all $t \in [2,T]$ and all $(i,j) \in E_t^+$, by a simple adaptation of the same argument in the proof of Theorem 2. Then,

$$\mathbb{P}\left(\mathcal{E}\right) \geq 1 - \sum_{t=1}^{T} \frac{|E_t^+|}{4K^2T^2} \geq 1 - \frac{1}{4T} \quad,$$

which follows from the fact that $|E_t^+| \leq K^2$ for all $t \in [2,T]$. We can similarly argue that $\mathbb{P}\left(|\tilde{p}_t(i,j) - p(i,j)| > p(i,j)/2\right) \leq 2\exp(-t\varepsilon_t/24) \leq 1/(2K^2T^2)$ for all $t \in [2,T]$ and all $(i,j) \notin E_t^-$; this implies that $\mathbb{P}\left(\mathcal{F}\right) \geq 1 - 1/(2T)$. Finally, we observe that $\mathbb{P}\left(\tilde{p}_t(i,j) \geq \varepsilon_t\right) \leq \exp(-t\varepsilon_t/6) \leq 1/(4K^2T^2)$ for all $t \in [2,T]$ and all $(i,j) \in E_t^-$, hence $\mathbb{P}\left(\mathcal{C}\right) \geq 1 - 1/(4T)$. The statement follows by union bound over the complements of $\mathcal{E}$, $\mathcal{F}$, and $\mathcal{C}$. □

## E  Weighted Independence Number

To improve the regret bounds in the case of strongly observable support, we need to introduce another graph-theoretic quantity: the *weighted independence number* $\alpha_{\mathsf{w}}(G, w)$, where $w \in \mathbb{R}_+^K$ is a vector of positive weights assigned to the vertices of our strongly observable graph $G = (V, E)$ with $V = [K]$. Let $w(U) = \sum_{i\in U} w_i$ denote the weight of a subset of vertices $U \subseteq V$. The weighted independence number is defined as

$$\alpha_{\mathsf{w}}(G, w) = \max_{S\in\mathcal{I}(G)} w(S) \quad,$$

that is, the weight of a maximum weight independent set. This set is chosen among all sets in the family $\mathcal{I}(G)$ of independent sets of $G$. It can be equivalently defined by the following

integer linear program:

$$\alpha_{\mathsf{w}}(G,w) = \max_{x} \quad \sum_{i=1}^{K} w_i x_i$$
$$\text{s.t.} \quad x_i + x_j \leq 1 \qquad \forall (i,j) \in E, i \neq j$$
$$x_i \in \{0,1\} \qquad \forall i \in V$$

We plan to define $w$ according to our needs in what follows.

## E.1   Undirected Graph

Let $\mathcal{G}$ be a stochastic feedback graph with edge probabilities $p(i,j)$ and such that its support $\mathrm{supp}(\mathcal{G}) = G = (V,E)$ is undirected and strongly observable. Moreover, let $N(i)$ be the neighborhood in $G$ of any vertex $i \in V$ (excluding $i$) and let $C(i) = N(i) \cup \{i\}$ be the extended neighborhood of $i$ including vertex $i$ itself.

We can use the edge probabilities from $\mathcal{G}$ to define a weight for each vertex $i$ as

$$w_{\mathcal{G}}(i) = w_i = \left( \frac{1}{|C(i)|} \sum_{j \in C(i)} p(j,i) \right)^{-1} .$$

This vertex weight is equal to the inverse of the arithmetic mean of the incident edge probabilities (including its self-loop). Note that the two probabilities $p(i,j)$ and $p(j,i)$ in the two directions of any undirected edge $(i,j) \in E$ need not be equal.

This definition allows us to upper bound the second-order term in the regret for vertices with self-loop (as similarly done in the analysis of EXP3.G [Alon et al., 2015]) in terms of the weighted independence number since we can reduce it to bounding

$$\sum_{i \in V} \frac{1}{\sum_{j \in C(i)} p(j,i)} = \sum_{i \in V} \frac{w_i}{|C(i)|} .$$

We thus require a weighted version of Turán's theorem, which is formulated in the lemma below. This result has already been proved [Sakai et al., 2003], but we nevertheless provide a proof for completeness.

**Lemma 15.** *Let $G = (V,E)$ be an undirected graph with positive vertex weights $w_i$. Then,*

$$\sum_{i \in V} \frac{w_i}{|C(i)|} \leq \alpha_{\mathsf{w}}(G,w) .$$

*Proof.* Consider the following algorithm: as long as the graph is not empty, repeatedly choose a vertex $j$ that minimizes $|C(j)|/w_j$ among all remaining vertices and remove it from the graph along with its neighborhood. Let $i_1, \ldots, i_s$ be the sequence of $s$ vertices picked by this algorithm, which form an independent set by construction. Additionally, let $G_1, \ldots, G_{s+1}$ be the sequence of graphs generated by this iterative procedure, where $G_1 = G$ is the starting graph and $G_{s+1}$ is the empty graph. We also let $C_r(i)$ denote the extended neighborhood over $G_r$ of any $i \in V(G_r)$. Define

$$Q(H) = \sum_{i \in V(H)} \frac{w_i}{|C(i)|} \qquad \forall H \subseteq G ,$$

as the quantity we are trying to bound for $G$ and consider it over the graphs in the sequence generated by the procedure. It is strictly decreasing until reaching $Q(G_{s+1}) = 0$. In particular, at any step of the procedure it decreases by

$$Q(G_r) - Q(G_{r+1}) = \sum_{j \in C_r(i_r)} \frac{w_j}{|C(j)|} \leq \sum_{j \in C_r(i_r)} \frac{w_{i_r}}{|C(i_r)|} = \frac{|C_r(i_r)|}{|C(i_r)|} w_{i_r} \leq w_{i_r} ,$$

where the first inequality is due to the optimality of $|C(i_r)|/w_{i_r}$ at step $r$. We can use this inequality to bound $Q(G)$ by

$$Q(G) = \sum_{r=1}^{s} (Q(G_r) - Q(G_{r+1})) \leq \sum_{r=1}^{s} w_{i_r} \leq \max_{S \in \mathcal{I}(G)} w(S) = \alpha_{\mathsf{w}}(G,w) .$$

$\square$

## E.2   Directed Graph

Compared to the result in the previous section, we are more generally interested in directed graphs. We consider the case of directed, strongly observable support $\text{supp}(\mathcal{G}) = G = (V, E)$ with $V = [K]$ and $(i, i) \in E$ for all $i \in V$. In the directed case, we distinguish the in-neighborhood $N^{\text{in}}(i)$ over $G$ of a vertex $i \in V$ from its out-neighborhood $N^{\text{out}}(i)$. We use the convention that vertices with self-loops are not included in their neighborhoods, while all vertices are always included in their extended in-neighborhood $C^{\text{in}}(i) = N^{\text{in}}(i) \cup \{i\}$ and out-neighborhood $C^{\text{out}}(i) = N^{\text{out}}(i) \cup \{i\}$, respectively. We make this distinction to comply as much as possible with previous works providing analogous results [Alon et al., 2017], where the neighborhoods $N^{\text{in}}(i)$ and $N^{\text{out}}(i)$ did not include $i$ even in the presence of the self-loop $(i, i) \in E$.

The weighted independence number is defined in the same way as per undirected graphs, ignoring the direction of edges for the independence condition. Here we define in two slightly different manners the vertex weights: let

$$w_{\mathcal{G}}^{\text{in}}(i) = w_i^{\text{in}} = \left( \frac{1}{|C^{\text{in}}(i)|} \sum_{j \in C^{\text{in}}(i)} p(j, i) \right)^{-1} \tag{43}$$

be the inverse of the arithmetic mean of the incoming edge probabilities for $i$, and

$$w_{\mathcal{G}}^{\text{out}}(i) = w_i^{\text{out}} = \left( \frac{1}{|C^{\text{out}}(i)|} \sum_{j \in C^{\text{out}}(i)} p(i, j) \right)^{-1} \tag{44}$$

the analogous over outgoing edges. These two different assignments of vertex weights induce two weighted independence numbers $\alpha_{\mathsf{w}}(G, w^{\text{in}})$ and $\alpha_{\mathsf{w}}(G, w^{\text{out}})$, respectively.

Then, we prove a lemma similar to [Alon et al., 2017, Lemma 13] in the weighted case. Note, however, that in this case the lemma is tightly related to the specific definitions of vertex weights we are adopting.

**Lemma 16.** *Let $G = (V, E)$ be a directed graph with edge probabilities $p(i, j) \in [0, 1]$, and positive vertex weight vectors $w^{\text{in}}$ and $w^{\text{out}}$ as in Equations (43) and (44), respectively. Then,*

$$\sum_{i \in V} \frac{w_i^{\text{in}}}{|C^{\text{in}}(i)|} \le 3(\alpha_{\mathsf{w}}(G, w^{\text{in}}) + \alpha_{\mathsf{w}}(G, w^{\text{out}})) \ln(K + 1) \ .$$

*Proof.* We prove the statement by induction as in the proof of Alon et al. [2017, Lemma 13]. Consider the following algorithm: as long as the graph is not empty, repeatedly choose the vertex $j$ that maximizes $|C^{\text{in}}(j)|/w_j^{\text{in}}$ among all remaining vertices and remove it from the graph along with its incident edges. Let $i_1, \ldots, i_K$ be the vertices in the order the algorithm picks them. Additionally, let $G_1, \ldots, G_{K+1}$ be the sequence of graphs generated by this iterative procedure, where $G_1 = G$ is the original graph and $G_{K+1}$ is the empty graph. We also let $C_r^{\text{in}}(i)$ denote the extended in-neighborhood over $G_r$ of any $i \in V(G_r)$. Similarly to the proof of Lemma 15, define

$$Q(H) = \sum_{i \in V(H)} \frac{w_i^{\text{in}}}{|C^{\text{in}}(i)|} \qquad \forall H \subseteq G$$

as the quantity we want to bound for $G$, where the size of the in-neighborhood is always computed with respect to the starting graph $G$.

Define a new instance of the problem with graph $G' = (V, E')$ as the undirected version of $G$, where the edge probabilities are defined as $p'(i, j) = \frac{1}{2} p(i, j) + \frac{1}{2} p(j, i)$ for all $i, j \in V$ such that either $(i, j) \in E$ or $(j, i) \in E$. This new graph has $C(i) = C^{\text{in}}(i) \cup C^{\text{out}}(i)$. As a consequence, we can derive new vertex weights $w_i' = \left( \frac{1}{|C(i)|} \sum_{j \in C(i)} p'(j, i) \right)^{-1}$. This instance is such that

$$\sum_{i \in V} \frac{|C(i)|}{w_i'} = \sum_{i \in V} \sum_{j \in C(i)} p'(j, i) = \sum_{i \in V} \sum_{j \in C^{\text{in}}(i)} p(j, i) = \sum_{i \in V} \frac{|C^{\text{in}}(i)|}{w_i^{\text{in}}} \ . \tag{45}$$

Furthermore, notice that the newly defined vertex weights satisfy

$$w_i' = \frac{|C(i)|}{\sum_{j \in C(i)} p'(j,i)} \le \frac{|C^{\mathrm{in}}(i)|}{\sum_{j \in C(i)} p'(j,i)} + \frac{|C^{\mathrm{out}}(i)|}{\sum_{j \in C(i)} p'(j,i)}$$

$$\le \frac{2|C^{\mathrm{in}}(i)|}{\sum_{j \in C^{\mathrm{in}}(i)} p(j,i)} + \frac{2|C^{\mathrm{out}}(i)|}{\sum_{j \in C^{\mathrm{out}}(i)} p(i,j)}$$

$$= 2(w_i^{\mathrm{in}} + w_i^{\mathrm{out}}) \ . \tag{46}$$

Consider now the first vertex $i_1$ chosen by the procedure we introduced before. The value it maximizes is lower bounded by

$$\max_{i \in V} \frac{|C^{\mathrm{in}}(i)|}{w_i^{\mathrm{in}}} \ge \frac{1}{K} \sum_{i \in V} \frac{|C^{\mathrm{in}}(i)|}{w_i^{\mathrm{in}}}$$

$$= \frac{1}{K} \sum_{i \in V} \frac{|C(i)|}{w_i'} \qquad\qquad \text{by Equation (45)}$$

$$\ge \frac{K}{\sum_{i \in V} \frac{w_i'}{|C(i)|}} \qquad\qquad \text{by Jensen's inequality}$$

$$\ge \frac{K/2}{\sum_{i \in V} \frac{w_i^{\mathrm{in}}}{|C(i)|} + \sum_{i \in V} \frac{w_i^{\mathrm{out}}}{|C(i)|}} \qquad\qquad \text{by Equation (46)}$$

$$\ge \frac{K/2}{\alpha_{\mathsf{w}}(G, w^{\mathrm{in}}) + \alpha_{\mathsf{w}}(G, w^{\mathrm{out}})} \ . \qquad \text{by Lemma 15 over } G' \tag{47}$$

We can use this fact to show an upper bound for the sum $Q(G)$ as

$$Q(G) = \sum_{i \in V} \frac{w_i^{\mathrm{in}}}{|C^{\mathrm{in}}(i)|} = \frac{w_{i_1}^{\mathrm{in}}}{|C^{\mathrm{in}}(i_1)|} + \sum_{r=2}^{K} \frac{w_{i_r}^{\mathrm{in}}}{|C^{\mathrm{in}}(i_r)|}$$

$$\le \frac{2(\alpha_{\mathsf{w}}(G, w^{\mathrm{in}}) + \alpha_{\mathsf{w}}(G, w^{\mathrm{out}}))}{K} + Q(G_2) \ . \qquad \text{by Equation (47)}$$

As a last step, recursively repeat the same reasoning on $Q(G_2)$ and iterate it until reaching $G_K$ to conclude that

$$Q(G) \le 2 \sum_{r=1}^{K} \frac{\alpha_{\mathsf{w}}(G_r, w^{\mathrm{in}}) + \alpha_{\mathsf{w}}(G_r, w^{\mathrm{out}})}{K - r + 1} \le 3(\alpha_{\mathsf{w}}(G, w^{\mathrm{in}}) + \alpha_{\mathsf{w}}(G, w^{\mathrm{out}})) \ln(K+1) \ .$$

$\square$

We finally have all the tools required for demonstrating the next lemma. It essentially corresponds to Alon et al. [2015, Lemma 5] with the addition of edge probabilities. The main difference is that we show an upper bound in terms of two distinct independence numbers. They are both computed over the graph $G$ with vertex weights defined in terms of the worst-case edge probabilities. To be specific, we have a first weight assignment $w^-$ to vertices such that $w_{\mathcal{G}}^-(i) = w_i^- = \left( \min_{j \in C^{\mathrm{in}}(i)} p(j,i) \right)^{-1}$ is the reciprocal of the minimum incoming edge probability for vertex $i$. The second assignment $w^+$, instead, assigns weight $w_{\mathcal{G}}^+(i) = w_i^+ = \left( \min_{j \in C^{\mathrm{out}}(i)} p(i,j) \right)^{-1}$ equal to the inverse of the minimum outgoing edge probability for $i$.

**Lemma 17.** *Let $G = (V, E)$ be a directed graph with $|V| = K \ge 2$ and edge probabilities $p(i,j)$, and such that $(i,i) \in E$ for all $i \in V$. Let $z_i \in \mathbb{R}_+$ be a positive weight assigned to each $i \in V$. Assume that $\sum_{i \in V} z_i \le 1$ and that $z_i \ge \beta$ for all $i \in V$, given some constant $\beta \in (0, \frac{1}{2}]$. Then,*

$$\sum_{i \in V} \frac{z_i}{\sum_{j \in C^{\mathrm{in}}(i)} z_j p(j,i)} \le 6(\alpha_{\mathsf{w}}(G, w^-) + \alpha_{\mathsf{w}}(G, w^+)) \ln \left( \frac{2K^2}{\beta \rho} \right) \ ,$$

*where $\rho = \min_{i \in V} \sum_{j \in C^{\mathrm{in}}(i)} p(j,i) > 0$.*

*Proof.* The structure of this proof is similar to that of Alon et al. [2015, Lemma 5]. Define a discretization of $z_1, \ldots, z_K$ such that $(m_i - 1)/M \leq z_i \leq m_i/M$ for positive integers $m_1, \ldots, m_K$ and $M = \left\lceil \frac{2K}{\beta\rho} \right\rceil$. The discretized values are such that, for all $i \in V$,

$$\sum_{j \in C^{\mathrm{in}}(i)} m_j p(j,i) \geq M \sum_{j \in C^{\mathrm{in}}(i)} z_j p(j,i) \geq \frac{2K}{\beta\rho}\beta \sum_{j \in C^{\mathrm{in}}(i)} p(j,i) \geq 2K \geq 2|C^{\mathrm{in}}(i)| \ , \qquad (48)$$

where the first inequality holds because $z_j \leq m_j/M$, the second follows by definition of $M$ and by the assumption on $z_j$, whereas the third is due to the definition of $\rho$. Then, the sum of interest becomes

$$\begin{aligned}
\sum_{i \in V} \frac{z_i}{\sum_{j \in C^{\mathrm{in}}(i)} z_j p(j,i)} &\leq \sum_{i \in V} \frac{m_i}{M \sum_{j \in C^{\mathrm{in}}(i)} z_j p(j,i)} && \text{since } z_i \leq m_i/M \\
&\leq \sum_{i \in V} \frac{m_i}{\sum_{j \in C^{\mathrm{in}}(i)} m_j p(j,i) - |C^{\mathrm{in}}(i)|} && \text{since } Mz_j \geq m_j - 1 \\
&\leq 2 \sum_{i \in V} \frac{m_i}{\sum_{j \in C^{\mathrm{in}}(i)} m_j p(j,i)} && \text{by Equation (48).} \qquad (49)
\end{aligned}$$

Now build a new directed graph $G' = (V', E')$ derived (as in the proof of Alon et al. [2015, Lemma 5]) from graph $G$ by replacing each node $i \in V$ with a clique $K_i$ of size $m_i$ and all its edges having probability $p(i,i)$. Additionally add an edge from any $i' \in K_i$ to any $j' \in K_j$ having edge probability $p(i,j)$ if and only if $(i,j) \in E$. As a consequence, the right-hand side of Equation (49) is equal to

$$2 \sum_{i \in V'} \frac{1}{\sum_{j \in C^{\mathrm{in}}_{G'}(i)} p(j,i)} \ .$$

Observe that the independent sets in $G$ are preserved in $G'$: any independent set $S = \{i : i \in V'\} \in \mathcal{I}(G')$ in $G'$ has a corresponding one $\{i : i' \in S, i' \in K_i\}$ in $G$ with same cardinality and weight, assuming that the weight of $i' \in K_i$ in $G'$ is equal to the weight of $i \in V$ according to the weight assignment in $G$. We can reduce this latter sum to the same form as in Lemma 16 by assigning vertex weights

$$w_{i'}^{\mathrm{in}} = \left( \sum_{j \in C^{\mathrm{in}}(i)} \frac{m_j}{\sum_{k \in C^{\mathrm{in}}(i)} m_k} p(j,i) \right)^{-1}, \quad w_{i'}^{\mathrm{out}} = \left( \sum_{j \in C^{\mathrm{out}}(i)} \frac{m_j}{\sum_{k \in C^{\mathrm{out}}(i)} m_k} p(i,j) \right)^{-1},$$

to each vertex $i' \in K_i$, for all $i \in V$. Indeed, the previous sum becomes

$$\begin{aligned}
\sum_{i \in V'} \frac{1}{\sum_{j \in C^{\mathrm{in}}_{G'}(i)} p(j,i)} &= \sum_{i \in V'} \frac{w_i^{\mathrm{in}}}{|C^{\mathrm{in}}_{G'}(i)|} \\
&\leq 3(\alpha_{\mathsf{w}}(G', w^{\mathrm{in}}) + \alpha_{\mathsf{w}}(G', w^{\mathrm{out}})) \ln(|V'| + 1) && \text{by Lemma 16} \\
&\leq 3(\alpha_{\mathsf{w}}(G, w^-) + \alpha_{\mathsf{w}}(G, w^+)) \ln(|V'| + 1) \ ,
\end{aligned}$$

where the last inequality follows from the fact that $w_{i'}^{\mathrm{in}} \leq w_i^-$ and $w_{i'}^{\mathrm{out}} \leq w_i^+$ for all $i \in V$ and all $i' \in K_i$.

We conclude the proof by observing that this newly constructed graph also has

$$1 + |V'| = 1 + \sum_{i \in V} m_i \leq 1 + \sum_{i \in V}(Mz_i + 1) \leq K + M + 1 \leq 2K\left(1 + \frac{1}{\beta\rho}\right) \leq \frac{2K^2}{\beta\rho}$$

vertices, where the final inequality holds because $\beta\rho \leq K/2$ by definition, and we used the fact that $K \geq 2$. □