# OpenReview forum: "Learning on the Edge: Online Learning with Stochastic Feedback Graphs"
_NeurIPS.cc/2022/Conference — NeurIPS 2022 Accept_

### Official Review · Reviewer_e1KK · 2022-07-04

**Rating:** 7
**Confidence:** 3
**Soundness:** 3 good
**Presentation:** 4 excellent
**Contribution:** 3 good

**Summary:**

This paper considers the problem of bandits with stochastic graph feedback. Specifically, each edge of the feedback graph is realized with a fixed probability at each round instead of the deterministic setting. The authors propose a novel algorithm which achieves $O\left(\min\{\sqrt{\alpha T/\epsilon_1}, (\delta/\epsilon_2)^{1/3}T^{2/3}\}\right)$ regret bound where $\alpha$ is the independent number of the graph clipped on $\epsilon_1$ and $\delta$ is the weak domination number clipped on $\epsilon_2$. The algorithm contains three components, including one part exploring the structure of the graph,  one part exploiting the graph by picking one action over a period of rounds and one part that decides when to switch from the exploration part to the exploitation part. The authors also show the almost matching lower bound of this problem. Finally, based on a stronger assumption on the observations, the authors derive an algorithm with tighter regret bound.

**Questions:**

See suggestions in "Strengths And Weaknesses".

**Ethics Review Area:**

["I don’t know"]

**Limitations:**

Yes, the authors addressed the limitations of their work.

**Strengths And Weaknesses:**

Strength:
- The problem considered in this paper is well motivated and interesting.
- The paper is well written. The algorithms are intuitive and nicely explained.
- The proposed algorithm is novel to me and the authors show that the algorithm achieves the near-optimal regret bound. Although the first exploration part and the second exploitation (block-wise strategy) are very intuitive and classic in the stochastic graph feedback problem, the transition condition between the two parts require some novelty and the authors address this issue by carefully balancing the regret in the exploration phase and the exploitation phase. I checked the proofs for sec 3 and sec 4 and briefly checked the proofs in sec 5 and based on what I have checked, the proofs make sense to me.

Weakness:
- Overall I do not find any major weaknesses with the current work. There are some minor typos and issues:
-- line 206 and 208: I think here $\Delta=O((1/\epsilon)\ln(KT))$ should be $\Delta=\Omega((1/\epsilon)\ln(KT))$ or at least $\Delta=\Theta((1/\epsilon)\ln(KT))$
-- Algorithm 4 looks very dense to me. It would be better if there are more explanations in the some lines of the algorithm.

---

> ### Author Response · Authors · 2022-08-02
> **Response to reviewer e1KK**
>
> We thank the reviewer for pointing out the typos in lines 206 and 208. We will fix them. In the next revision, we will make an effort to clarify Algorithm 4 by adding some explanations in the pseudocode and will consider following a more modular approach to its presentation.

---

### Official Review · Reviewer_z5ND · 2022-07-08

**Rating:** 7
**Confidence:** 4
**Soundness:** 3 good
**Presentation:** 3 good
**Contribution:** 3 good

**Summary:**

The paper studies the online learning framework with stochastic feedback graphs. Unlike previous works, no assumptions are made on the feedback graphs. A regret-minimization algorithm, namely EdgeCatcher, is proposed and its regret bounds are analyzed in 3 cases where the feedback graphs are either strongly observable, observable or not observable. A lower bound is derived matching the upper-bounds guarantees of EdgeCatcher. Tighter bounds are also derived in the case where the entire feedback graph is observed at the end of each round.

**Questions:**

What is the complexity of computing $\varepsilon_s^*$ and $\varepsilon_w^*$ in (1) and (2)? Is there any better way to compute them  than considering each possible value of epsilon  in a brute-force manner? Similar question for the terms defined in Section 5.

The stopping criterion $\Phi$ defined in (5) depends on the constants $C_s, C_w$; so it also depends on the choice of algorithm A (so it should be written as $\Phi(\mathcal{G}, T, \mathcal{A})$. By (1) and (2), in the non-observable case, this $\Phi$ becomes infinite and then the RoundRobin will be played until $T/K$; then Blockreduction is called upon the remaining rounds. Will this lead to a linear regret?

The condition on $\varepsilon$ in Corollary 1 is rather counter-intuitive: as I understand, the smaller epsilon, the better approximation we have from RoundRobin, why do we need to have a lower-bound of $\varepsilon$ here, i.e., even if one can somehow obtain a better approximation, we cannot use it?

Any motivational examples for the cases considered in Section 5, or is this purely a theoretical question whether one can gets better guarantees? I might be mistaken but the notation $\hat{H}_t$  (Equation (6)) does not seem to be introduced anywhere.






**Limitations:**

 the authors adequately addressed the limitations and potential negative societal impact of their work

**Strengths And Weaknesses:**

Strengths: the paper is well-written and easy to follow. The presented results are interesting and relevant to the community. Although the proposed algorithms are based on previously known ideas, the combination of such elements in building up EdgeCatcher is quite elegant and interesting.

Weaknesses: certain parts in the main text requires more discussions (see below). Concrete examples and/or numerical experiments to show the performance of the proposed algorithms will be appreciated.

After the rebuttal phase: the authors answer the raised concerns. I am willing to increase the score to 7.

---

> ### Author Response · Authors · 2022-08-02
> **Response to reviewer z5ND**
>
> First of all, we thank the reviewer for the time invested in reviewing our work, and for the constructive comments. In this reply, we address all questions in hope of clarifying all uncertainties. Nevertheless, we invite the reviewer to ask further questions if needed.
>
> We begin by justifying the condition on the threshold $\varepsilon$ in Corollary 1. The lower bound on $\varepsilon$ is due to the fact that, with $T$ rounds, we can only obtain good estimates of edge probabilities that are larger than $1/T$. In any case, as the reviewer points out, even if we had an oracle telling us the true stochastic feedback graph, we could not use edges with probabilities smaller than $1/T$. This is the core of the tightness of our result: there is no gap between what we can explore (edges $e$ with $p_e \ge 1/T$) and what we can exploit (edges $e$ with $p_e \ge 1/T$).
>
> As for the computation of the optimal thresholds $\varepsilon_s^*$ and $\varepsilon_w^*$, defined in (1) and (2), we can actually bring down the number of thresholds that need to be considered. First, notice that it is immediate to see that we need only consider no more than $K^2$ possible thresholds: one for each edge probability. However, we can further reduce the number of computations if we aggregate edge probabilities within a constant factor of each other.
> Assume we check the stopping condition in RoundRobin only when $\tau$ is a power of $2$; i.e., $\tau=2^b$ for increasing integers $b \ge 1$. Then, this single check will cover all rounds $\tau'$ such that $\tau/2 = 2^{b-1} \le \tau' \le 2^b = \tau$ by considering the stochastic graph estimate $\hat{\mathcal{G}}$ achieved at round $\tau$. Using such a graph, we can compute $\alpha_{\varepsilon}/\varepsilon$ and $\delta_{\varepsilon}/\varepsilon$ with $\varepsilon = \varepsilon_\tau$, which are also $2$-approximations for the best respective ratios on any thresholded graph corresponding to rounds between $\tau/2$ and $\tau$ (conditioning on the high-probability event that we have $\varepsilon_{\tau'}$-good approximations simultaneously for all rounds $\tau'$). Observe that doing so is equivalent to restricting only to thresholds $\varepsilon_\tau$ in a decreasing geometric progression with common ratio $1/2$. Overall, the total number of computations is thus $O(\\min\\{\log(T), K^2\\})$ because we are interested in probability thresholds greater than $1/T$ (as already argued). This does not worsen the order of the regret bound while considering only a logarithmic number of thresholds.
>
> We then address the definition and behavior of the stopping criterion $\Phi$. In (5) we explicitly define $\Phi$ according to the regret bounds guaranteed by EXP3.G, so it indeed depends on our choice of $\mathcal{A}$. Other choices would change not only the constants $C_s$ and $C_w$, but also the rest of the formula. Once the algorithm is fixed, however, the input of $\Phi$ for different instances of the problem is just the graph and the time horizon.
> The observation is correct: looking at our $\Phi$, if the feedback graph is not observable, then the RoundRobin routine would run for $\Omega(T)$ time steps, thus yielding a linear worst-case regret, which is the best one could hope for in the non-observable case.
>
> Please, check the additional comment below for our response to the questions related to Section 5.

---

> > ### Author Response · Authors · 2022-08-02
> > **Response to reviewer z5ND (continued)**
> >
> > As per the utility of the improved bounds in Section 5, we provide some motivational examples as requested by the reviewer. The question of finding better guarantees for specific feedback graphs is interesting both from the theoretical and the practical perspective. There are indeed simple and natural stochastic feedback graphs for which the bounds provided in Section 5 improve significantly on the ones in Section 3. For instance, consider the scenario described in Example 1: there are $K$ actions and only self-loops with action-specific probabilities $\varepsilon_i$. Our general bound guarantees a $\sqrt{KT/(\min_i \varepsilon_i)}$ regret, while the refined bounds in Section 5 yield a $\sqrt{(\sum_i  1/\varepsilon_i)T}$ regret. The difference between the two bounds can be quite significant when, for example, only one of the $\varepsilon_i$ is very small, while all the others are close to $1$.
> > This example is not just a mathematical construction. Consider in fact the following scenario of faulty bandits: there are $K$ sensors $s_1, \dots, s_K$ that communicate with a central agent: every day, each sensor $s_i$ fails independently with some (unknown) probability $1 - \varepsilon_i$. Every day, the central agent sends a measurement request to one of the sensors. If the request is accepted, then the measurement is sent back to the server. Additionally, at the end of each day, each sensor sends the agent its daily status report (i.e., whether it was accepting requests or not during that day). Since each sensor always sends back a status report, it is possible for the central agent to reconstruct the realized feedback graph on top of receiving the information on the measurement. This problem is perfectly modeled by our example and can be tackled with improved bounds by OptimisticThenCommitGraph (otcG).
> >
> > Finally, regarding the comment on Equation (6), we thank the reviewer for pointing out this typo. We will fix it in the next revision.

---

> > > ### Comment · Reviewer_z5ND · 2022-08-08
> > > **response to authors**
> > >
> > > Thank you for answering the questions and clarifying the details. I am satisfied with most of the responses. A question/speculation still remains:
> > >
> > > *In (5) we explicitly define  according to the regret bounds guaranteed by EXP3.G, so it indeed depends on our choice of . Other choices would change not only the constants  and , but also the rest of the formula.*
> > >
> > > This makes me wonder if it might be better (and is it possible?) to write the formula of the stopping function in terms of the regret of the chosen algorithm A (as the choice of A changes the formulas)? In my personal opinions, the constants $C_s , C_w , C $ (that are not and cannot be explicitly defined) only add more confusions to the results statements.

---

> > > > ### Author Response · Authors · 2022-08-08
> > > > **Remarks on the stopping function $\Phi$**
> > > >
> > > > Thank you for your response, it is much appreciated. We are pleased that you are satisfied with most of the responses; if you have other specific concerns or questions we will be happy to provide further clarifications.
> > > >
> > > > Here we address the question/speculation about the stopping condition. It is indeed possible to state a “black-box” version of Theorem 4, valid for any learning algorithm $\mathcal{A}$, but we decided to specialize the result for EXP3.G for ease of exposition and analysis. We will add a comment on this in the revised version. Moreover, per our lower bounds this construction is minimax optimal (up to polylog terms), thus no algorithm can behave much better than our EdgeCatcher in terms of regret.
> > > > Even if part of the results are specific for EXP3.G, however, our current approach and presentation is highly modular: only Section 3.3 and Corollary 1 are stated in terms of EXP3.G and thus it is fairly easy to rephrase these results using other learning algorithms (and their regret bounds).
> > > >
> > > > Regarding $C_s$ and $C_w$ in our results, they are induced by our specific choice of the learning algorithm $\mathcal{A}$ and correspond to precise values, that can be easily derived from the analysis of EXP3.G in Alon et al. [2015]. We will report them in the revised version. $C$ is instead defined in the proof of Theorem 4 in the supplementary material (Appendix A) as a function of $C_s$ and $C_w$.
> > > >
> > > > Finally, we decided to express our results using the constants $C_s$ and $C_w$ for two reasons: first, we need to choose some constants for our stopping time to be well defined (the sole information about the asymptotic regime is not enough) and, second, we want to be able to say that the regret bound achievable in the commit phase is exactly the value of $\Phi$ that triggered the stopping condition.

---

### Official Review · Reviewer_FkW2 · 2022-07-11

**Rating:** 6
**Confidence:** 1
**Soundness:** 3 good
**Presentation:** 3 good
**Contribution:** 3 good

**Summary:**

This paper provides online learning with stochastic feedback graphs, which is a generalization of vanilla online learning with feedback graphs in the sense that the feedback is stochastic.
The authors provide regret upper bound, which is a natural extension of the regret bound in the vanilla setting under milder conditions than the one in many existing works.
A more efficient algorithm is provided under the assumption that the learner can observe the realization of the entire graph.
Their algorithm is based on the idea of reducing the considering problem to the vanilla one by introducing threshold $\epsilon$.

**Questions:**

- Is it in most cases that the $\sqrt{(\alpha_\epsilon/\epsilon) T}$ is much smaller than the other $(\delta_\epsilon/\epsilon)^{1/3} T^{2/3} \$?
- The considered setting can also be regarded as online learning with time-varying feedback graphs. Could the authors point out the relation between that setting?

**Strengths And Weaknesses:**

Strengths:
- Assumptions on the underlying graph are largely relaxed.
- The limitations of existing are well described.
- The paper is well organized and written.

Weaknesses:

(minor)
- The notation $\Delta$ around Eq.(3) is a bit confusing.
- The definition of $\Phi$ is given in page 7 while it is used in Algorithm 1 (page 5). I would like to see the definition or reference in page 5.

---

> ### Author Response · Authors · 2022-08-02
> **Response to reviewer FkW2**
>
> We want to thank the reviewer for the time invested in reading our work. In what follows, we try to provide answers to the questions in a clear way, with comparisons and examples that may help in conveying the relevance of our results.
>
> We first address the question about the comparison between regret bounds achievable with strongly observable and weakly observable thresholded supports, respectively.
> There are simple examples where the strongly observable term $\sqrt{(\alpha_{\varepsilon_s^*}/\varepsilon_s^*)T}$ dominates the weakly observable one $(\delta_{\varepsilon_w^*}/\varepsilon_w^*)^{1/3} T^{2/3}$, given the respective best thresholds $\varepsilon_s^*$ and $\varepsilon_w^*$ for the two terms. Consider for instance the following stochastic feedback graph on $K$ actions: one revealing action observes all the other actions and itself with probability $1$, while all the other edges of the complete graph on the $K$ actions have probability $\varepsilon$. The truncated graph $G_1$ is weakly observable, and the corresponding term in the regret is $T^{2/3}$ (as the dominating set consists of the sole revealing action); the best truncated graph that is strongly observable is $G_{\varepsilon}$, that is associated with $\sqrt{T/\varepsilon}$. It is clear that, for small $\varepsilon$, the latter bound becomes way weaker; for instance, for $\varepsilon = 1/T$, the weakly observable term is sublinear (i.e., $T^{2/3}$), while the strongly observable one is linear.
> This example sheds light on one of the main challenges of this problem: finding the right trade-off between keeping exploring, in the hope of finding better feedback graphs, and exploiting what has already been discovered. Although generally better, strong observability should not be pursued at all costs if weak observability is available with good enough probabilistic guarantees.
>
> Second, we clarify the differences between our setting and the time-varying feedback graph one, as requested by the reviewer in the last question. We could regard the setting as online learning with time-varying feedback graphs. However, current works in this setting assume that the entire sequence of graphs is either strongly observable or weakly observable. On the other hand, note that, due to the randomness in our setting, a realized graph may not even be observable, which means that at least one of the nodes does not have an incoming edge. As far as we can tell, this implies that the existing results do not apply to our setting.
> If we ignore the fact that the graphs might not be observable, we can make an interesting observation. Cohen et al. (2016) study a setting where the graphs are adversarially chosen, albeit guaranteeing self-loops at all nodes, and only the local structure of the feedback graph is observed. They show that if the losses are generated by an adversary, one cannot do better than $\sqrt{KT}$ regret and we might as well simply employ a standard bandit algorithm. Furthermore, removing the guarantee on the self-loops induces an $\Omega(T)$ regret. In Section 3 of our paper, we also observe only local information about the feedback graph and the losses are generated by an adversary. However, we show that if the graphs are stochastically generated, there is a $\sqrt{\alpha T/\varepsilon}$ regret bound. As a consequence, for $\varepsilon$ not too small, observing only the local information about the feedback graphs is in fact sufficient to obtain better results than in the bandit setting.

---

### Official Review · Reviewer_MMd7 · 2022-07-11

**Rating:** 6
**Confidence:** 3
**Soundness:** 3 good
**Presentation:** 3 good
**Contribution:** 3 good

**Summary:**

The paper considers the online learning problem with adversarial rewards and a stochastic feedback graph. There has been a line of research on this problem, most of them need some assumptions on the graph structure and knowledge about the parameter of the graph. The main contribution of the paper is a minimax optimal algorithm with a less stringent assumption on the graph structure and needs no knowledge about the graph parameter. The main idea of the algorithm is to estimate the graph structure properly and feed the estimated graph to the Exp3.G algorithm (cf. https://arxiv.org/abs/1502.07617). Furthermore, the paper shows that if we observe not only the realized reward but also the realized graph, we can achieve better regret in some problem instances.

**Questions:**

The related work section could be improved. This section looks like a simple enumeration of existing research, with little discussion about the relationship between this paper and prior work. For example, it says that “Following Mannor and Shamir [2011], we can consider a more general scenario where the feedback graph is not fixed but changes over time, resulting in a sequence G1, . . . , GT of feedback graphs [Alon et al., 2013, 2017, Kocák et al., 2014, 2016b, Cohen et al., 2016].” If the author could elaborate on the similarity and differences between the scenario of their paper and the scenario of this line of work, the related work section will be more informative.

**Limitations:**

I don't see potential negative societal impact.

**Strengths And Weaknesses:**

The contribution of the paper is solid. It generalizes existing research in an interesting way. The algorithm only assumes that all edges in the stochastic graph are independent, and needs no more knowledge about the graph structure.
Strength:
1. The paper is well written. The idea is clearly presented.
2. The paper generalizes existing research by relaxing the assumption significantly.
3. The upper bound has a matching minimax lower bound.
4. The author reduces the stochastic feedback graph problem to the deterministic feedback graph problem in a nontrivial way.
Weakness:
1. The main algorithm is a reduction to the well-known Exp3.G algorithm, thus the novelty is not significant.

---

> ### Author Response · Authors · 2022-08-02
> **Response to reviewer MMd7**
>
> We thank the reviewer for investing time in reading our work and writing the review. We are also grateful for the constructive comments.
>
> We begin by commenting on the weakness pointed out by the reviewer. We agree with the reviewer that one of our main results is the reduction of the stochastic feedback graph problem to the deterministic feedback graph problem, where one can use the EXP3.G algorithm, which is minimax optimal up to logarithmic factors. However, there are many aspects of our work that are novel, and we hope will be used in future work on online learning with feedback graphs. First, the reduction itself is original and, as the reviewers argued, also “nontrivial”, “quite elegant and interesting”. The careful design of the stopping time and of the exploration procedure results in a tight reduction: the near-minimax-optimal algorithms for the deterministic feedback graph problem are translated to near-minimax-optimal algorithms for the stochastic version. Second, we highlight that our second algorithm, OptimisticThenCommitGraph (otcG) (Section 5 and Appendix C), introduces many novel features, infusing new ideas into the EXP3.G approach. Just to cite a few of them: we use a notion of importance sampling adapted to the stochastic feedback graph setting, we introduce tighter and novel graph parameters and offer new results derived from the weighted version of Turán's theorem for the strongly observable case. Finally, we consider upper confidence estimates for the edge probabilities, bounding their divergence from the mean using an empirical Bernstein bound, and prove that using them in the importance-weighted estimate for the losses (in place of the true probabilities) suffices.
>
> As per the comments on the related works section, we are happy to elaborate further on how those results relate to ours; we will also make that section clearer in the next revision. Cohen et al. (2016) study a setting where the graphs are adversarially chosen and only the local structure of the feedback graph is observed. They show that, if the losses are generated by an adversary and all nodes always have a self-loop, one cannot do better than $\sqrt{KT}$ regret, and we might as well simply employ a standard bandit algorithm. Furthermore, removing the guarantee on the self-loops induces an $\Omega(T)$ regret. In Section 3 of our paper, we are in a similar situation, as we also observe only local information about the feedback graph and the losses are generated by an adversary. However, we show that if the graphs are stochastically generated with a strongly observable support for some threshold $\varepsilon$, there is a $\sqrt{\alpha T/\varepsilon}$ regret bound. As a consequence, for $\varepsilon$ not too small, observing only the local information about the feedback graphs is in fact sufficient to obtain better results than in the bandit setting. Similarly, if there are no self-loops in the support but the support is weakly observable, then our regret bounds are sublinear rather than linear in $T$.
>
> Alon et al. (2013, 2017) and Kocák et al. (2014) also consider adversarially generated sequences $G_1,G_2,\ldots$ of deterministic feedback graphs. In the case of directed feedback graphs, Alon et al. (2013) investigate a model in which $G_t$ is revealed to the learner at the beginning of each round $t$. Alon et al. (2017) and Kocák et al. (2014) extend this analysis to the case when $G_t$ is strongly observable and made available only at the end of each round $t$. In comparison, in our setting the graphs (or the local information about the graph) revealed to the learner (at the end of each round) may not even be observable, let alone strongly observable. Despite this seemingly challenging setting for previous works, we nevertheless obtain sublinear regret bounds.
>
> Finally, Kocák et al. (2016b) study a feedback model where the losses of other actions in the out-neighborhood of the action played are observed with an edge-dependent noise. In their setting, the feedback graphs $G_t$ are weighted and revealed at the beginning of each round. They introduce edge weights $s_t(i,j) \in [0,1]$ that determine the feedback according to the following additive noise model: $s_t(I_t,j)\ell_t(j) + \bigl(1-s_t(I_t,j)\bigr)\xi_t(j)$, where $\xi_t(j)$ is a zero-mean bounded random variable. Hence, if $s_t(i,j) = 1$, then $I_t=i$ allows to observe the loss of action $j$ without any noise. If $s_t(i,j) = 0$, then only noise is observed. Note that they assume $s_t(i,i)=1$ for each $i$, implying strong observability. Although similar in spirit to our feedback model, our results do not directly compare with theirs.

---

> > ### Comment · Reviewer_MMd7 · 2022-08-09
> > **Reply to Rebuttal**
> >
> > Thanks for the careful response.
> > After reading authors' response and comments of other reviewers, I tend to keep my original score.

---

### Meta-Review · Area_Chair_e1Ny · 2022-08-24

**Recommendation:** Accept
**Confidence:** Certain

**Metareview:**

The reviewers came to consensus that this paper makes a good progress on online learning with stochastic feedback graphs. I agree with these opinions and please polish the manuscript by addressing the raised minor concerns such as the presentation issues in the final version.

**Award:**

No

---

### Decision · Program_Chairs · 2022-09-14

Accept